# Loss-Aware Distributionally Robust Optimization
# via Trainable Optimal Transport Ambiguity Sets

**Jonas Ohnemus** [1]   **Marta Fochesato** [1]   **Riccardo Zuliani** [1]   **John Lygeros** [1]

## Abstract

Optimal-transport distributionally robust optimization (OT-DRO) robustifies data-driven decision-making under uncertainty by capturing the sampling-induced statistical error via optimal transport ambiguity sets. The standard OT-DRO pipeline consists of a two-step procedure, where the ambiguity set is first designed and subsequently embedded into the downstream OT-DRO problem. However, this separation between uncertainty quantification and optimization may lead to excessive conservatism. We introduce an end-to-end pipeline to automatically learn decision-focused ambiguity sets for OT-DRO problems, where the loss function informs the shape of the ambiguity set, leading to less conservative decisions whose distributional robustness is enforced via data-driven bootstrapping. We formulate the learning problem as a bilevel optimization program and solve it via a hypergradient-based method. By leveraging the recently introduced nonsmooth conservative implicit function theorem, we establish convergence to a critical point of the bilevel problem. We present experiments validating our method on standard portfolio optimization and linear regression tasks.

## 1. Introduction

Optimal-transport distributionally robust optimization (OT-DRO) has emerged as a principled framework for decision-making under uncertainty due to its ability to capture distributional uncertainty arising, for example, from sampling. The OT-DRO problem can be thought of as the zero-sum game

$$\inf_{w \in \mathcal{W}} \sup_{\mathbb{Q} \in \mathcal{A}} \mathbb{E}_{\xi \sim \mathbb{Q}} \left[ \ell(w, \xi) \right], \qquad (1)$$

[1]Automatic Control Laboratory (IfA), ETH Zürich, Zürich, Switzerland. Correspondence to: Jonas Ohnemus <johnemus@ethz.ch>.

*Proceedings of the 43$^{rd}$ International Conference on Machine Learning*, Seoul, South Korea. PMLR 306, 2026. Copyright 2026 by the author(s).

where the decision-maker chooses a decision $w \in \mathcal{W} \subseteq \mathbb{R}^k$ to minimize the expectation of the loss function $\ell$, and the adversary chooses a distribution $\mathbb{Q}$ from a so-called ambiguity set $\mathcal{A} \subseteq \mathcal{P}(\Xi)$ with $\mathcal{P}(\Xi)$ denoting the set of probability distributions supported on $\Xi \subseteq \mathbb{R}^d$. The OT-DRO problem in (1) has gained attention over the past years thanks to applications in machine learning (Shafieezadeh Abadeh et al., 2015; Blanchet et al., 2019a; Ho-Nguyen & Wright, 2023), portfolio optimization (Blanchet et al., 2022), control (Taskesen et al., 2023; Aolaritei et al., 2023), and power systems (Poolla et al., 2021), among others.

The traditional deployment of OT-DRO (1) relies on two steps carried out sequentially. In the first step, the ambiguity set is designed on the basis of the empirical observations, typically in the form of independent samples $\{\hat{\xi}_1, \ldots, \hat{\xi}_J\}$ extracted from an unknown probability distribution $\mathbb{P}$. A natural way to integrate this information is by constructing an ambiguity set

$$\mathcal{A} := \mathcal{B}_\varepsilon(\hat{\mathbb{P}}) = \left\{ \mathbb{Q} \in \mathcal{P}(\Xi) \mid d(\mathbb{Q}, \hat{\mathbb{P}}) \le \varepsilon \right\}, \qquad (2)$$

defined as a ball of radius $\varepsilon \in \mathbb{R}_{\ge 0}$ centered around the empirical distribution $\hat{\mathbb{P}} = \frac{1}{J} \sum_{j=1}^{J} \delta_{\hat{\xi}_j}$. Here, $d : \mathcal{P}(\Xi) \times \mathcal{P}(\Xi) \to [0, +\infty)$ is an optimal transport based discrepancy defined as $d(\mathbb{P}_1, \mathbb{P}_2) := \inf_{\pi \in \Pi(\mathbb{P}_1, \mathbb{P}_2)} \mathbb{E}_{\xi_1, \xi_2 \sim \pi}[\kappa(\xi_1, \xi_2)]$, where $\kappa : \Xi \times \Xi \to [0, +\infty)$ is a prescribed transportation cost function satisfying the identity of indiscernibles, and $\Pi(\mathbb{P}_1, \mathbb{P}_2)$ represents the set of all joint probability distributions of $\xi_1, \xi_2$ with marginals $\mathbb{P}_1, \mathbb{P}_2$, respectively. In the second step, $\mathcal{A}$ is embedded into (1) and the resulting problem is solved either by resorting to finite-dimensional reformulations grounded on duality theory (Gao & Kleywegt, 2023; Blanchet & Murthy, 2019), or on stochastic gradient methods (Yu et al., 2022; Li et al., 2019; 2020). If the ambiguity set is designed to contain the true distribution $\mathbb{P}$ with high probability, DRO theory offers an out-of-sample certificate ensuring that when deploying the obtained minimizer on new, unseen data, *post-decision disappointment* does not occur with high probability (unlike standard empirical risk minimization methods, such as sample average approximation (Smith & Winkler, 2006)).

This traditional OT-DRO pipeline can lead to overly conservative decisions, as the downstream optimization problem

does not inform the geometry of the ambiguity set. Consider, for example, the choice of the transportation cost as $\kappa(\xi_1, \xi_2) = \|\xi_1 - \xi_2\|_2^p$, resulting in the celebrated type-p Wasserstein distance (Mohajerin Esfahani & Kuhn, 2018). Intuitively, this choice requires that the true distribution deviates little from the empirical estimate in *all* directions. In reality, however, our primary concern is to exclude only those distributions that actively increase the worst-case cost, rather than imposing uniform constraints in all directions. This intuition motivates the introduction of a new OT-DRO methodology, where the geometry of the ambiguity set — which represents a degree of freedom that has not been exploited thus far — is informed by the loss function $\ell$ to ensure coverage of the true distribution while selectively excluding adversarial distributions that inflate the worst-case cost. To address the looseness of standard DRO bounds based on measure concentration inequalities, we replace them with a bootstrap-based constraint (Summers & Kamgarpour, 2021).

Our contributions are as follows.

- **End-to-end OT-DRO pipeline.** We introduce a novel *end-to-end* OT-DRO pipeline to automatically learn decision-focused ambiguity sets $\mathcal{A}_\theta$, defined by some parameter $\theta \in \mathbb{R}^{n_\theta}$, leading to less conservative decisions whose distributional robustness is enforced via data-driven bootstrapping. Specifically, among all ambiguity sets leading to the same out-of-sample disappointment $\beta$, we want to determine the one ensuring minimum out-of-sample risk.

- **Algorithmic solution.** We encode the problem as a bilevel optimization program where the upper level chooses a geometry for $\mathcal{A}_\theta$, for example via a parametrized transportation cost $\kappa(\cdot, \cdot; \theta)$, while the lower level solves (1) for the chosen ambiguity set $\mathcal{A}_\theta$. To solve the bilevel problem, we use a hypergradient-based method based on the recently introduced nonsmooth conservative implicit function theorem, and we show that, under mild conditions, our numerical scheme provably converges to the set of critical points of the bilevel program (or to a neighbourhood of them) despite the nonsmoothness and nonconvexity of the solution map of the lower level.

- **Software.** We present an open-source implementation of our algorithm in Python at: `https://github.com/JonasOhn/trainable-ot-dro`.

## 2. Related Works

**Metric learning.** Metric learning (Bellet et al., 2013) (also referred to as "smart predict-and-optimize" in operations research (Elmachtoub & Grigas, 2022)) refers to the paradigm of training a predictive model to minimize the loss on a downstream optimization task and has gained increasing attention across several domains (Donti et al., 2017), (Demirović et al., 2019), (Wilder et al., 2019), (Cameron et al., 2022) (see also the survey papers (Mandi et al., 2024; Sadana et al., 2025)). Recently, it has also been applied to design decision-focused uncertainty sets (Chenreddy et al., 2022; Sun et al., 2023; Wang et al., 2025; Chenreddy & Delage, 2024). In particular, (Wang et al., 2025) addresses contextual stochastic optimization, where the goal is to learn an uncertainty set (in $\mathbb{R}^n$) that maximizes expected performance across a family of contextual problems. By contrast, we consider distributional ambiguity and robustify in the space of probability distributions. Within OT-based DRO, (Blanchet et al., 2019b) calibrates a Mahalanobis distance to penalize directions with high performance impact. However, their approach is limited to linear regression and requires a separate calibration step, whereas our end-to-end framework uses the DRO loss to directly inform the set design, reducing conservatism. (Behzadian et al., 2021) learns norm weights in robust Markov decision processes with finite state-action spaces, while (Schuurmans & Patrinos, 2023) selectively enlarges ambiguity sets of discrete distributions in directions with limited effect on the worst-case cost. However, neither approach extends to continuous distributions or supports flexible parameterizations of the transport cost. Further, (Costa & Iyengar, 2023) leverages residual structure of the uncertainty distribution, limiting general applicability, and (Ma et al., 2024) differentiates through conic programs to learn conic-representable sets for mixed-integer DRO, but overlooks the nondifferentiability of the solution map. Finally, compared to (Chaouach et al., 2023), our approach reduces conservatism without relying on independence assumptions among features.

**Conformal risk training.** A related line of work investigates end-to-end calibration and risk control via conformal methods. Conformal risk training (CRT) (Yeh et al., 2025) is an end-to-end framework for conformal risk control that differentiates through the calibration procedure, thereby aligning calibration with the downstream application. In this sense, CRT shares our end-to-end perspective, but differs in its robustness mechanism. Specifically, CRT provides finite-sample guarantees for a scalar predictive risk under exchangeability. Moreover, although CRT supports risk functionals such as conditional value-at-risk (CVaR) (Yeh et al., 2025), CVaR can be interpreted, at the level of the empirical loss distribution, as protecting against tail losses by allowing an adversary to reweight observed losses (Shapiro et al., 2021, Chapter 6). In contrast, the OT-DRO framework considered here allows robustness in both probability weights and support: the adversary may transport probability mass from observed samples to potentially unobserved points, with the transport cost encoding plausibility.

**Differentiable optimization.** Differentiable optimization refers to the practice of differentiating the solution map of optimization problems, generally by applying the implicit function theorem to their optimality conditions (Dontchev & Rockafellar, 2009). This idea has been applied to differentiate the solution map of quadratic programs (Amos & Kolter, 2017), linear programs with a regularizing term (Mandi & Guns, 2020), and linear conic programs (Busseti et al., 2019; Agrawal et al., 2019). As a direct consequence of the implicit function theorem, all these methods implicitly assume that the solution map is continuously differentiable with respect to the problem parameters. However, the solution map of an (even convex) optimization program is generally not everywhere differentiable. To relax the continuous differentiability assumption, one can utilize the concept of conservative Jacobians (Bolte & Pauwels, 2021), which extend traditional gradients to almost everywhere differentiable functions, and, most notably, admit a nonsmooth implicit function theorem (Bolte et al., 2021). Leveraging this concept, (Zuliani et al., 2025) develops a first-order method with convergence guarantees to solve a bilevel problem with a quadratic lower level. In (Bolte et al., 2021, Proposition 4), the authors apply the nonsmooth implicit function theorem in the context of conic programs. This work is a fundamental building block for the algorithm used in this paper.

## 3. Preliminaries

**Notation.** We assume an underlying probability space $(\Omega, \mathcal{F}, P)$ and define the distribution of any random vector $\xi : \Omega \to \mathbb{R}^d$ by the pushforward distribution $\mathbb{P} = P \circ \xi^{-1}$ of $P$ with respect to $\xi$. $\mathcal{P}(\Xi)$ denotes the set of probability distributions on $\Xi \subseteq \mathbb{R}^d$ and $\mathcal{P}_g(\Xi)$ its restriction to the set of Gaussians. We use $\mathcal{N}(\mu, \Sigma)$ to denote a Gaussian distribution with mean $\mu \in \mathbb{R}^d$ and covariance $\Sigma \in \mathbb{R}^{d \times d}$, and $\mathcal{U}(a, b)$ to denote a uniform distribution in the interval $[a, b]$. For $n \in \mathbb{Z}_+$, we set $[n] = \{1, \dots, n\}$. Given a probability distribution $\mathbb{P}$ and a set $\mathcal{X}$, we use $\mathbb{P}^n := \mathbb{P} \times \dots \times \mathbb{P}$ and $\mathcal{X}^{\otimes n} := \mathcal{X} \otimes \dots \otimes \mathcal{X}$ to denote the product distribution and the product set, respectively. We denote the Euclidean norm with $\| \cdot \|$ and use $\text{dist}(x, \mathcal{X}) := \inf\{\|x - z\| \mid z \in \mathcal{X}\}$ to denote the point-to-set distance. Lastly, $\mathbb{L}_{++}^n$ denotes the set of positive definite lower triangular matrices.

**Path differentiability.** *Conservative Jacobians* can be used to generalize the notion of Jacobian to functions that are almost everywhere differentiable (Bolte & Pauwels, 2021). Specifically, given a locally Lipschitz function $f : \mathbb{R}^n \to \mathbb{R}^m$, we say that the outer semicontinuous, compact-valued map $\mathcal{J}_f : \mathbb{R}^n \rightrightarrows \mathbb{R}^{m \times n}$ is a conservative Jacobian of $f$ if, given any absolutely continuous function $\varphi : [0, 1] \to \mathbb{R}^n$, $\frac{d}{dt} f(\varphi(t)) = V \dot{\varphi}(t)$ for any $V \in \mathcal{J}_f(\varphi(t))$ and almost every $t \in [0, 1]$. By Rademacher's theorem, $\nabla f(x)$ exists for

almost every $x \in \mathbb{R}^n$, in which case, by (Bolte & Pauwels, 2021, Theorem 1), $\mathcal{J}_f(x) = \{\nabla f(x)\}$, meaning that $\mathcal{J}_f$ coincides almost everywhere with the standard Jacobian. We say that a function $f$ is *path-differentiable* if it admits a conservative Jacobian. Here, we focus on the class of locally Lipschitz functions that are *definable in an o-minimal structure*, or simply *definable* (Coste, 1999) (see also Appendix B for a concise explanation), which always admit a conservative Jacobian (Bolte & Pauwels, 2021, Proposition 2). This class of functions contains most functions commonly found in the fields of control and optimization, including semialgebraic functions and analytic functions restricted to a definable domain.

Given a locally Lipschitz definable function $f : \mathbb{R}^n \to \mathbb{R}$ and a sequence of positive step sizes $\{\alpha_i\}_{i \in \mathbb{N}}$, the update rule

$$x_{i+1} = x_i - \alpha_i d_i, \quad d_i \in \mathcal{J}_f(x_i), \tag{3}$$

is guaranteed to converge to a critical point $\bar{x}$ for which $0 \in \mathcal{J}_f(\bar{x})$ if $\alpha_i > 0$ is square summable but not summable (Davis et al., 2020, Theorem 3.2). If $\alpha_i \equiv \bar{\alpha}$, then for a small enough $\bar{\alpha}$, $\limsup_{i \to \infty} \text{dist}(0, \mathcal{J}_f(x_i)) \leq \epsilon$, where $\epsilon > 0$ can be made arbitrarily small by reducing $\bar{\alpha}$ (Bolte et al., 2025, Theorem 2.2).

**Differentiating through conic programs.** Conic programs are a broad class of optimization problems of the form

$$\begin{aligned} \min_{x,s} \quad & c^\top x \\ \text{subject to} \quad & Ax + s = b \\ & (x, s) \in \mathbb{R}^n \times \mathcal{K}, \end{aligned} \tag{4}$$

where $\mathcal{K}$ is a closed convex cone such as the nonnegative orthant or the second-order cone. Assuming $(A, b, c) = (A(\theta), b(\theta), c(\theta))$ depend on a parameter $\theta \in \mathbb{R}^{n_\theta}$, one can treat the primal-dual solution $(x^*(\theta), y^*(\theta), s^*(\theta))$ of (4) (assuming its existence) as a function of $\theta$ and write the solution map as $\mathcal{S} : \mathbb{R}^{n_\theta} \to \mathbb{R}^{n+2m}$, where $\mathcal{S}(\theta) = (x^*(\theta), y^*(\theta), s^*(\theta))$. By implicitly differentiating the KKT conditions of (4), it is possible to obtain the conservative Jacobian $\mathcal{J}_\mathcal{S}(\theta)$ of $\mathcal{S}$ (Bolte et al., 2021). A more thorough description of the differentiation procedure is provided in Appendix C.

## 4. Learning the Uncertainty in OT-DRO

### 4.1. Problem Formulation

We modify the OT-DRO problem (1) by replacing $\mathcal{A}$ with a parametrized ambiguity set $\mathcal{A}_\theta$ given by

$$\mathcal{A}_\theta := \mathcal{B}_\varepsilon(\hat{\mathbb{P}}; \theta) := \{\mathbb{Q} \in \mathcal{P}(\Xi) \mid d(\mathbb{Q}, \hat{\mathbb{P}}; \theta) \leq \varepsilon\}, \tag{5}$$

where the parameter $\theta \in \Theta \subseteq \mathbb{R}^{n_\theta}$ affects the optimal transport based discrepancy, given by

$$d(\mathbb{Q}, \hat{\mathbb{P}}; \theta) := \inf_{\pi \in \Pi(\mathbb{Q}, \hat{\mathbb{P}})} \mathbb{E}_{\xi_1, \xi_2 \sim \pi}[\kappa(\xi_1, \xi_2; \theta)],$$

through the transportation cost $\kappa(\cdot, \cdot; \theta)$, thus defining the geometry of the set, and $\varepsilon$ is a fixed parameter ensuring that the set contains the true probability distribution with high confidence. While $\varepsilon$ is typically tuned via cross-validation or calibrated using measure concentration bounds to guarantee statistical coverage, in this paper, we set $\varepsilon$ to a fixed value and only focus on modifying the geometry of the set through $\theta$. This leads to the parameterized OT-DRO problem

$$\inf_{w \in \mathcal{W}} \sup_{\mathbb{Q} \in \mathcal{A}_\theta} \mathbb{E}_{\xi \sim \mathbb{Q}} \left[ \ell(w, \xi) \right]. \tag{6}$$

In the remainder, we always assume that (6) is well-posed for all $\theta \in \Theta$, that is, it admits a finite minimizer $\hat{w}_\theta$. Conditions for well-posedness of problem (6) have been established in (Yue et al., 2022). Let $\hat{\ell}(\hat{w}_\theta)$ be the corresponding optimal solution, and let $\ell^\star := \inf_{w \in \mathcal{W}} \mathbb{E}_{\xi \sim \mathbb{P}}[\ell(w, \xi)]$ be the optimal loss under complete knowledge of the distribution $\mathbb{P}$. Our goal is to determine a parameter vector $\theta^\star$ that attains the lowest in-sample loss $\hat{\ell}(\hat{w}_{\theta^\star})$ satisfying

$$\Pr(\ell^\star \leq \mathbb{E}_{\xi \sim \mathbb{P}}[\ell(\hat{w}_{\theta^\star}, \xi)] \leq \hat{\ell}(\hat{w}_{\theta^\star})) \geq 1 - \beta, \tag{7}$$

for a given user-defined reliability parameter $\beta \in (0, 1)$. Equation (7) represents an out-of-sample performance certificate on the data-driven decision $\hat{w}_{\theta^\star}$. By maximally reducing $\hat{\ell}(\hat{w}_{\theta^\star})$ through a careful design of the geometry of $\mathcal{A}_\theta$, the out-of-sample loss $\mathbb{E}_{\xi \sim \mathbb{P}}[\ell(\hat{w}_{\theta^\star}, \xi)]$ gradually approaches the true loss $\ell^\star$, which represents a fixed problem-specific global lower bound.

A sufficient condition for (7) to hold is $\Pr(\mathbb{P} \in \mathcal{B}_\varepsilon(\hat{\mathbb{P}}; \theta^\star)) \geq 1 - \beta$, since whenever $\mathbb{P}$ lies in the ambiguity set, the worst-case loss over the set is an upper bound of the out-of-sample loss. We therefore optimize the following surrogate:

$$\hat{\ell}(w_\theta) := \inf_{w \in \mathcal{W}, \theta \in \Theta} \sup_{\mathbb{Q} \in \mathcal{B}_\varepsilon(\hat{\mathbb{P}}; \theta)} \mathbb{E}_{\xi \sim \mathbb{Q}} \left[ \ell(w, \xi) \right] \tag{8a}$$

$$\text{subject to} \quad \Pr(\mathbb{P} \in \mathcal{B}_\varepsilon(\hat{\mathbb{P}}; \theta)) \geq 1 - \beta. \tag{8b}$$

The interpretation is that among all the ambiguity sets containing the true probability distribution $\mathbb{P}$ with high likelihood, we select the one resulting in the lowest value of $\hat{\ell}$.

As an example, consider the simplified problem in Figure 1, where the loss increases in the northeast direction. If the orange and purple ambiguity sets both contain $\mathbb{P}$ with probability $1 - \beta$, then choosing the orange set is more convenient, as this set produces a less

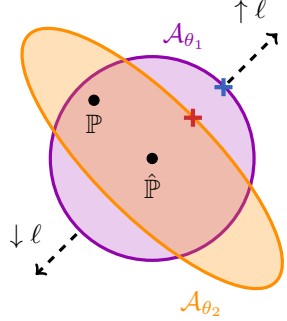

*Figure 1.* Simplified problem.

conservative solution (red cross) compared to the purple set (blue cross) while maintaining the same out-of-sample guarantees.

We propose an automated pipeline to solve (8) that solely relies on the structure of the loss function $\ell$ and on the availability of $J$ samples $\{\hat{\xi}_1, \ldots, \hat{\xi}_J\}$ without requiring knowledge of $\mathbb{P}$.

### 4.2. A Bilevel Formulation

We consider a bilevel surrogate of (8) that exploits the structural properties of the problem

$$\inf_{\theta \in \Theta} \sup_{\mathbb{Q} \in \mathcal{B}_\varepsilon(\hat{\mathbb{P}}; \theta)} \mathbb{E}_{\xi \sim \mathbb{Q}} [\ell(w_\theta^\star, \xi)]$$

$$\text{s.t.} \quad \Pr(d(\mathbb{P}, \hat{\mathbb{P}}; \theta) \leq \varepsilon) \geq 1 - \beta \tag{9}$$

$$w_\theta^\star = \arg \inf_{w \in \mathcal{W}} \sup_{\mathbb{Q} \in \mathcal{B}_\varepsilon(\hat{\mathbb{P}}; \theta)} \mathbb{E}_{\xi \sim \mathbb{Q}} [\ell(w, \xi)].$$

Compared to (8), the bilevel formulation in (9) separates the joint minimization over $w$ and $\theta$ across two different levels, with the shape-inducing variable $\theta$ being optimized at the upper level and the task-related variable $w$ being optimized at the lower level. We can think of (9) as the problem of *tuning* an optimal transport-based ambiguity set.

**Lower level.** For a fixed $\theta \in \Theta$, the lower-level constitutes a standard OT-DRO problem of the form (1). For the problem classes considered in this paper, and under the regularity conditions listed in Appendix A, this lower-level OT-DRO problem admits the finite-dimensional convex reformulation

$$\mathcal{S}(\theta) = \begin{cases} \arg \min & c(\theta)^\top x \\ \text{s.t.} & A(\theta)\, x + s = b(\theta) \\ & s \in \mathcal{K}. \end{cases} \tag{10}$$

where $\mathcal{S}(\theta) = (x^\star(\theta), y^\star(\theta), s^\star(\theta))$ is the solution map of (10) consisting of the primal, dual, and the slack variables, respectively, and $\mathcal{K}$ is a convex and closed cone. Problem (10) is a parametrized conic program, where the functional representation of $(A(\theta), b(\theta), c(\theta))$ depends on the specific parametrization of the transportation cost $\kappa(\cdot, \cdot; \theta)$. We rely on the following standing assumption.

**Assumption 4.1.** *The minimizer of the lower level of* (9) *is unique for all $\theta \in \Theta$.*

Assumption 4.1 is typically employed in the context of implicit differentiation, see e.g. (Bolte et al., 2021).

Examples of suitable parameterizations $\kappa(\cdot, \cdot; \theta)$ of the transportation cost that lead to formulations of the form (10) include:

1. *Mahalanobis distance*: Let $L \in \mathbb{L}_{++}^d$ be a positive definite lower triangular matrix, and $p \in [0, \infty)$. Then,

for $\theta = L$, the transportation cost $\kappa(\xi_1, \xi_2; \theta) = \left\| L^\top(\xi_1 - \xi_2) \right\|_2^p$ encodes anisotropic features in the form of different sensitivities along different directions in $\mathbb{R}^d$. More specifically, using the singular value decomposition of $L = U\Sigma V^\top$, with $\{\sigma_i\}_{i=1}^d$ being the singular values of $L$ and $\{u_i\}_{i=1}^d$ the orthonormal columns of $U$, the transportation cost becomes $\|L^\top(\xi_1 - \xi_2)\|_2^p = \left( \sum_{i=1}^d \sigma_i^2 |u_i^\top(\xi_1 - \xi_2)|^2 \right)^{p/2}$. Thus, moving a unit of probability mass from the center distribution in the direction $u_i$ costs $\sigma_i^p$: the higher the value of $\sigma_i$, the less probability mass is moved in the direction $u_i$ and vice versa.

2. *Functional composition of norms*: Let $f : \mathbb{R}_{\geq 0} \times \mathbb{R}^d \to \mathbb{R}_{\geq 0}$ be a continuous and strictly convex function parametrized for any $\theta \in \mathbb{R}^d$ such that (i) $x_1 \leq x_2 \implies f(x_1; \theta) \leq f(x_2; \theta)$, (ii) $\lim_{x \to \infty} \frac{f(x;\theta)}{x} \to \infty$, (iii) $f(0; \theta) = 0$. Then, $f(\|\xi_1 - \xi_2\|; \theta)$ is a valid parametrization. Possible choices of $f$ include convex combinations of norms, the (scaled) exponential and logarithmic functions, and the maximum of quadratic functions.

3. *Conic combination of transportation costs*: Let $\kappa_1(\xi_1, \xi_2), \ldots, \kappa_m(\xi_1, \xi_2)$ be valid transportation costs; then $\sum_{i=1}^m \theta_i \kappa_i(\xi_1, \xi_2)$, with $\theta_i \geq 0$, is a valid parametrization. If the dictionary $\{\kappa_i\}_{i=1}^m$ is complete[1], this parametrization universally determines all possible geometries of the ambiguity set.

**Upper level.** Since the optimal value of the lower-level conic reformulation equals the DRO worst-case value, the upper-level objective can be written as the linear conic objective $c(\theta)^\top x^\star(\theta)$; any nonlinearities of the original loss are encoded in the auxiliary variables and constraints collected in $x$. The upper-level problem is thus given by

$$\min_{\theta \in \Theta} \quad c(\theta)^\top x^\star(\theta) \tag{11a}$$

$$\text{subject to} \quad \Pr(d(\mathbb{P}, \hat{\mathbb{P}}; \theta) \leq \varepsilon) \geq 1 - \beta. \tag{11b}$$

The probability in (11b) is taken with respect to the random sampling process that generates the dataset $\hat{\mathcal{D}}_J := \{\hat{\xi}_1, \ldots, \hat{\xi}_J\}$ used to construct the reference distribution $\hat{\mathbb{P}} := \hat{\mathbb{P}}(\hat{\mathcal{D}}_J) = \frac{1}{J} \sum_{j=1}^J \delta_{\hat{\xi}_j}$. The set $\hat{\mathcal{D}}_J$ is a realization of the random multi-sample $\mathcal{D}_J$ distributed according to $\mathbb{P}^J$ and supported on $(\Xi)^{\otimes J}$.

[1] By complete we mean that the dictionary spans the entire space of admissible transportation costs defined as $\mathcal{T} := \{\kappa(x, y) \text{ symmetric}, \kappa(x, x) = 0, \kappa(x, y) > 0 \text{ iff } x \neq y\}$. For example, if the dictionary only includes Euclidean distances possibly raised to some powers, their conic hull cannot approximate non-metric costs. Nonetheless, even incomplete dictionaries are often "rich" enough in practice.

Note that (11b) depends on the true distribution $\mathbb{P}$, which is not known in our setting. We can approximate the probability in (11b) by bootstrapping samples of $\mathcal{D}_J$ from the set $\hat{\mathcal{D}}_J$ with replacement, obtaining

$$\begin{aligned} \Pr(d(\mathbb{P}, \hat{\mathbb{P}}; \theta) \leq \varepsilon) &= \mathop{\mathbb{E}}_{\mathcal{D}_J \sim \mathbb{P}^J} \left[ \mathbb{1} \left\{ d(\mathbb{P}, \hat{\mathbb{P}}(\mathcal{D}_J); \theta) \leq \varepsilon \right\} \right] \\ &\approx \frac{1}{n_{\mathrm{b}}} \sum_{k=1}^{n_{\mathrm{b}}} \mathbb{1} \left\{ d(\hat{\mathbb{P}}, \hat{\mathbb{P}}(\hat{\mathcal{D}}_J^k); \theta) \leq \varepsilon \right\}. \end{aligned} \tag{12}$$

where $n_{\mathrm{b}}$ denotes the number of multi-samples $\hat{\mathcal{D}}_J^k$ extracted from $\hat{\mathbb{P}}$. We can then rewrite (11b) as

$$\frac{1}{n_{\mathrm{b}}} \sum_{k=1}^{n_{\mathrm{b}}} \mathbb{1} \left\{ d(\hat{\mathbb{P}}, \hat{\mathbb{P}}(\hat{\mathcal{D}}_J^k); \theta) \leq \varepsilon \right\} \geq 1 - \beta, \tag{13}$$

where we require the distance between the nominal distribution and at least a $1 - \beta$ fraction of the bootstrapped distributions to not exceed $\varepsilon$. Notice that (13) can effectively be implemented with the available information.

## 5. Algorithm Design

In this section, we devise a hypergradient-based algorithm with convergence guarantees to solve the problem in (9). The main challenges lie in (i) ensuring that $\theta$ belongs to the feasible set $\Theta := \{\theta \in \mathbb{R}^{n_\theta} : (13) \text{ holds}\}$, and (ii) obtaining the hypergradient, i.e., the gradient of the upper-level objective function $c(\theta)^\top x^\star(\theta)$ with respect to $\theta$.

Generally, $\Theta$ is a nonconvex set as the condition in (13) is not convex in $\theta$ (see Appendix D.1.1 for a proof in the case of the Mahalanobis distance). To avoid computationally expensive projection operations, we solve instead the following unconstrained optimization problem, where the constraints are replaced with a penalty function

$$\min_\theta \quad \varphi(\theta) := \underbrace{c(\theta)^\top x^\star(\theta)}_{:=\varphi_{\mathrm{o}}(\theta, x^\star(\theta))} + \underbrace{\lambda_{\mathrm{p}} \max\{0, e(\theta)\}^2}_{:=\varphi_{\mathrm{p}}(\theta)}. \tag{14}$$

where

$$e(\theta) = \left( \frac{1}{n_{\mathrm{b}}} \sum_{k=1}^{n_{\mathrm{b}}} \sigma \left( d(\hat{\mathbb{P}}, \hat{\mathbb{P}}(\hat{\mathcal{D}}_J^k); \theta)/\varepsilon - 1 \right) \right) - \beta, \tag{15}$$

and $\sigma(x) = [1 + \exp(-\eta_{\mathrm{p}} x)]^{-1}$ is the sigmoid function. In (14), $\varphi_{\mathrm{p}}(\theta)$ penalizes positive constraint violations $e(\theta)$ by a large coefficient $\lambda_p \in \mathbb{R}_{>0}$. The expression in (15) is a smooth approximation of the complement of the indicator function in (13), with $\eta_{\mathrm{p}} \in \mathbb{R}_{>0}$ regulating the approximation accuracy, where larger values yield a better approximation.

To obtain the hypergradient of the objective in (14), we need the following assumption.

**Assumption 5.1.** *The functions $c$ and $x^\star$ are locally Lipschitz and definable in $\theta$. For any $\mathbb{P}_1$ and $\mathbb{P}_2$, $d(\mathbb{P}_1, \mathbb{P}_2; \theta)$ is locally Lipschitz and definable in $\theta$.*

Assumption 5.1 is mild and has already been proposed and studied in the context of conic programming (Bolte et al., 2021; 2024). In Appendix C we provide more details outlining sufficient conditions under which Assumption 5.1 holds in our setting. Under Assumption 5.1, the hypergradient of (14) can be obtained by applying the chain rule

$$\mathcal{J}_\varphi(\theta) = \{ J_{\varphi_o} + 2\lambda_{\mathrm{p}} \max\{0, e(\theta)\} J_e : \\ J_{\varphi_o} \in \mathcal{J}_{\varphi_o}(\theta), \ J_e \in \mathcal{J}_e(\theta) \}, \quad (16)$$

where

$$\mathcal{J}_{\varphi_o}(\theta) = \{ J_{\varphi_o, \theta} + J_{x^\star}^\top J_{\varphi_o, x} : \\ [J_{\varphi_o, \theta} \ J_{\varphi_o, x}] \in \mathcal{J}_{\varphi_o}(\theta, x^\star(\theta)), \\ J_{x^\star} \in \mathcal{J}_{x^\star}(\theta) \}. \quad (17)$$

Computing the Jacobian of the constraint violation $e(\theta)$ requires differentiating the value function of an optimal transport problem with respect to $\theta$, as explained in Appendix D. The conservative Jacobian $\mathcal{J}_{x^\star}$ of $x^\star$ can be obtained by differentiating the solution of the conic program (10), as explained in Appendix C.

Given an element $J_\varphi(\theta_i)$ of $\mathcal{J}_\varphi(\theta_i)$, the update step for $\theta$ follows the hypergradient descent dynamics

$$\theta_{i+1} = \theta_i - \alpha_i J_\varphi(\theta_i), \qquad \alpha_i \in \mathbb{R}_{>0}.$$

We summarize the proposed learning procedure in Algorithm 1, whose computational complexity is discussed in Appendix H.

---

**Algorithm 1** Loss-aware DRO

---

**Input:** $\theta_0, \hat{\mathcal{D}}_J = \{\hat{\xi}_j\}_{j=1}^J, \{\alpha_i\}_{i \in \mathbb{N}}, \alpha_i > 0$
**Output:** $\theta^\star, \hat{w}_{\theta^\star}$
Bootstrap $n_{\mathrm{b}}$ reference distributions $\{\hat{\mathbb{P}}_k\}_{k=1}^{n_{\mathrm{b}}}$ by resampling $\hat{\mathcal{D}}_J$ with replacement
Find $\varepsilon$ once as the $1 - \beta$ quantile of $\{d(\hat{\mathbb{P}}_k, \hat{\mathbb{P}}; \theta_0)\}_{k=1}^{n_{\mathrm{b}}}$
**for** $i = 1$ **to** maxiter **do**
   Solve conic program with $A(\theta_i), b(\theta_i), c(\theta_i)$ for $x^\star(\theta_i)$
   Differentiate through conic program for $J_{x^\star}(\theta_i)^\top c(\theta_i)$
   Calculate $d(\hat{\mathbb{P}}_k, \hat{\mathbb{P}}; \theta_i)$ for all $k \in [n_{\mathrm{b}}]$ and obtain $J_e(\theta_i) \in \mathcal{J}_e(\theta_i)$
   Compute $J_\varphi(\theta_i) = J_{x^\star}(\theta_i)^\top c(\theta_i) + x^\star(\theta_i)^\top J_c(\theta_i) + 2\lambda_{\mathrm{p}} \max\{0, e(\theta_i)\} J_e(\theta_i)$
   Update $\theta_{i+1} = \theta_i - \alpha_i J_\varphi(\theta_i)$
   Update the conic program parameters $A(\theta_{i+1})$, $b(\theta_{i+1}), c(\theta_{i+1})$
**end for**

---

### 5.1. Convergence

The gradient descent procedure in Algorithm 1 is guaranteed to converge to a critical point under mild conditions.

**Theorem 5.1.** *Under Assumption 5.1, if the step sizes $\alpha_i \geq 0$ are square summable but not summable, and $\sup_i \|\theta_i\| < +\infty$, then $\theta_\infty := \lim_{i \to \infty} \theta_i \in \mathrm{crit}\,\varphi$, where $\mathrm{crit}\,\varphi = \{\theta : 0 \in \mathcal{J}_\varphi(\theta)\}$. If $\alpha_i \equiv \bar{\alpha}$, then for every $\epsilon > 0$ there exists an $\bar{\alpha} > 0$ such that $\limsup_{i \to \infty} \mathrm{dist}(\theta_i, \mathrm{crit}\,\varphi) \leq \epsilon$.*

*Proof.* Definability of $e(\theta)$ follows immediately from the definability of $d(\mathbb{P}_1, \mathbb{P}_2; \theta)$, since all functions involved are definable, and inversion preserves definability (Kurdyka, 1998, Remark 2). The result then follows from (Davis et al., 2020, Theorem 3.2) for vanishing step sizes, and from (Bolte et al., 2025, Theorem 2.2) for constant step sizes. □

As a by-product of our problem formulation, we can also embed an OT-DRO problem as a general-purpose differentiable layer. Our analysis is general and extends beyond OT-based ambiguity set to *any* ambiguity set that can be represented as a conic set. This class is broad and encompasses many relevant uncertainty descriptions (Ben-Tal & Nemirovski, 2001), including support-, moment-, and entropic-based ambiguity sets. The differentiable layer can be used in any machine learning framework (e.g., PyTorch, TensorFlow, JAX) to learn the ambiguity set parameters and can be integrated into a larger layered architecture.

## 6. Numerical Experiments

All experiments are synthetic: the true distribution $\mathbb{P}$ is used only for evaluation, not by the learning algorithm. This allows us to compute out-of-sample performance and empirically assess the coverage condition $\mathbb{P} \in \mathcal{B}_\varepsilon(\hat{\mathbb{P}}; \theta)$, which would not be directly observable on real data.

### 6.1. Portfolio Optimization

We consider a portfolio optimization problem where the goal is to find the optimal allocation of $k$ financial assets to maximize the profit from the investment. Mathematically, $\ell(w, \xi) = -w^\top \xi$, where $w \in \mathbb{R}^k$ is the vector of allocation weights, $\xi \in \mathbb{R}^k$ are the returns, and $\mathcal{W} = \{ w \in \mathbb{R}^k \mid \mathbf{1}^\top w = 1, \ w \succeq 0 \}$. We assume that the returns $\xi$ are distributed according to an unknown Gaussian probability distribution $\mathbb{P}$. The decision maker considers a nominal distribution $\hat{\mathbb{P}} \triangleq \mathcal{N}(\hat{\mu}, \hat{\Sigma})$ formed via sample mean $\hat{\mu}$ and covariance $\hat{\Sigma}$. We use a parametric Wasserstein ambiguity set with $\kappa(\xi_1, \xi_2; \theta) = \|L^\top(\xi_1 - \xi_2)\|_2^2$ where $\theta = L \in \mathbb{L}_{++}^k$ is the tuning parameter initialized at $L_0 = I_k$. Formally, the transportation cost corresponds to the Mahalanobis distance with weight $L$. To hedge against the tail-risk, we consider the Conditional Value at Risk

(CVaR) of the loss $\ell$ (Rockafellar & Uryasev, 2000). For any given geometry parameter $L$, the lower-level OT-DRO portfolio problem reads

$$\min_{w \in \mathcal{W}} \max_{\mathbb{Q} \in \mathcal{B}_\varepsilon(\hat{\mathbb{P}}; L)} \text{CVaR}_\gamma^{\mathbb{Q}} \left( -w^\top \xi \right). \tag{18}$$

where

$$\text{CVaR}_\gamma^{\mathbb{Q}}(X) = \min_{\tau \in \mathbb{R}} \left\{ \tau + \frac{1}{\gamma} \mathbb{E}_{X \sim \mathbb{Q}}[\max(0, X - \tau)] \right\}$$

and $\mathcal{B}_\varepsilon(\hat{\mathbb{P}}; L) = \{\mathbb{Q} \in \mathcal{P}_g(\mathbb{R}^k) \mid d(\mathbb{Q}, \hat{\mathbb{P}}; L) \leq \varepsilon\}$ is a parametrized OT ambiguity set restricted to contain only Gaussians.[2] In the bilevel procedure, (18) is solved for the current value of $L$, while $L$ itself is updated by the upper-level problem. We defer the reformulation of (18) to Appendix A.2.2 and the experimental details to Appendix E.1. Further, we report results on portfolio optimization using a discrete reference distribution in Appendix F.2. We first illustrate how our procedure works on a specific problem instance with $k = 2, J = 30, n_b = 20, \gamma = 0.05, \beta = 0.1$. We focus on this interpretable low-dimensional setting to visually verify that our method correctly aligns the transportation cost with the loss landscape. Results for $k = 10$ are presented in Appendix F.1. As the iterations progress, the $L$ matrix changes (cf. Figure 3, top left), increasing the weight of the probability mass in the lower-left corner corresponding to adversarial returns. This effectively decreases the worst-case $\text{CVaR}_\gamma^{\mathbb{Q}^\star}$, as distributions that assign high probability mass in this direction are excluded from the ambiguity set. At the same time, this leads to an improved out-of-sample performance $\text{CVaR}_\gamma^{\mathbb{P}}$, indicating a reduction in the conservatism of the solution (top right). The gap between the blue and the red curves in Figure 3 (top right) reflects the "price of robustness" due to only knowing the distribution through samples, as $\mathcal{B}_\varepsilon(\hat{\mathbb{P}}; L)$ also contains different, and possibly more adversarial, distributions. In addition to the single example, we validate our procedure on 50 independent experiments with $k = 3$, each using a different true Gaussian distribution $\mathbb{P}$. For each distribution, 10 distinct datasets are sampled. To evaluate the effectiveness of our procedure, we monitor the relative improvement of the worst-case objective

$$f_0 = \text{CVaR}_\gamma^{\xi \sim \mathbb{Q}^\star(L_0)} \left[ -(w^\star(L_0))^\top \xi \right]$$
$$\text{and } f^\star = \text{CVaR}_\gamma^{\xi \sim \mathbb{Q}^\star(L^\star)} \left[ -(w^\star(L^\star))^\top \xi \right]$$

and of the out-of-sample performance

$$\ell_0 = \text{CVaR}_\gamma^{\xi \sim \mathbb{P}} \left[ -(w^\star(L_0))^\top \xi \right]$$
$$\text{and } \ell^\star = \text{CVaR}_\gamma^{\xi \sim \mathbb{P}} \left[ -(w^\star(L^\star))^\top \xi \right].$$

---

[2]This restriction is solely done to simplify the implementation of the bootstrapping procedure, as it suffices to bootstrap the first two moments of the nominal distribution to fully characterize the bootstrapped ones. Appendix F.2 provides a fully data-driven implementation that does not rely on such a restriction.

Figure 3 (bottom) confirms that a reduction of conservatism is observed on average across all problem instances, with the relative improvement being larger for a smaller number of samples. We assess the coverage of the true distribution by tracking the parametrized, normalized distance $d(\mathbb{P}, \hat{\mathbb{P}}; L)$ across all experiments in Figure 2. The results show that the proposed penalty prevents the bilevel update from worsening the empirical coverage behavior of the initial bootstrap-calibrated ambiguity set. In contrast, omitting the coverage penalty in (13) generally shrinks the ambiguity set in directions needed for coverage, preventing it from containing the true distribution.

## 6.2. Linear Regression

Next, we consider the distributionally robust linear regression task (Kuhn et al., 2024; Shafieezadeh-Abadeh et al., 2019; Blanchet et al., 2019a; Chen & Paschalidis, 2020), where the goal is learning a linear regressor that performs well under uncertainty in the data distribution. Let $x \in \mathbb{R}^k$ and $y \in \mathbb{R}$ be the independent and dependent variables, respectively, and let $\xi = (x, y) \in \mathbb{R}^{k+1}$. We parametrize the transportation cost as $\kappa(\xi_1, \xi_2; \theta) = \left\| L^\top (\xi_1 - \xi_2) \right\|_2$ with $\theta = L \in \mathbb{L}_{++}^{k+1}$ initialized at $L_0 = I_{k+1}$, and consider the loss function $\ell_1(w, \xi) = \left| (-w, 1)^\top \xi \right| = \left| \bar{w}^\top \xi \right|$, where the decision $w \in \mathbb{R}^k$ represents the weights of the linear model. The distributionally robust linear regression problem reads

$$\min_{w \in \mathbb{R}^k} \max_{\mathbb{Q} \in \mathcal{B}_\varepsilon(\hat{\mathbb{P}}; L)} \mathbb{E}_{\xi \sim \mathbb{Q}} \left( \ell_1(w, \xi) \right), \tag{19}$$

where $\hat{\mathbb{P}} = \frac{1}{J} \sum_{j=1}^J \delta_{\hat{\xi}_j}$ with $\hat{\xi}_j = \{(\hat{x}_j, \hat{y}_j)\}$. We defer the reformulation of (19) to Appendix A.3.1 and the experimental details to Appendix E.2. Further, we report the analogous case with a squared loss function $\ell_2(w, \xi) = \left( (-w, 1)^\top \xi \right)^2$ in Appendix F.3.

As before, we begin by showing the results of a single problem instance, where $\xi = (x, y) \in \mathbb{R}^2$ is generated with the following linear model corrupted by white noise

$$y = w \, x + e, \tag{20}$$

where $e \sim \mathcal{N}(0, \sigma)$ and $x \sim \mathcal{U}(-10.0, 10.0)$. The weight $w$ is deterministic and set to 1, and the standard deviation of the noise is $\sigma = 10$.

Figure 4 shows the true distribution of $\xi$ and the $J = 20$ samples drawn from it (top left), the unit transportation cost ellipses across iterations (top right), and the effect of our procedure on the linear model (bottom). We set $n_b = 20$. Figure 5 (top) shows the worst-case expected absolute error $e_{wc}(L) = \mathbb{E}_{\xi \sim \mathbb{Q}^\star(L)} \left( \ell_1(w^\star(L), \xi) \right)$, corresponding to the upper level objective, and the expected absolute error on the true data generating process $e_{oos}(L) = \mathbb{E}_{\xi \sim \mathbb{P}} \left( \ell_1(w^\star(L), \xi) \right)$, approximated with $10^7$ independent

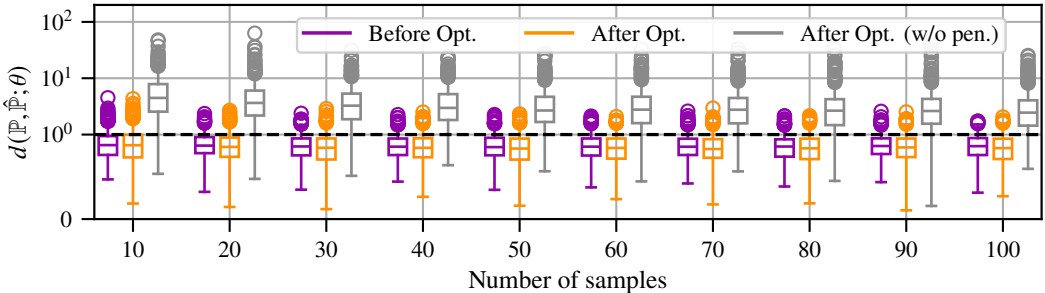

*Figure 2.* Normalized distance $d(\mathbb{P}, \hat{\mathbb{P}}; L)$ between reference and true distributions before (violet) and after (orange) the bilevel procedure is applied (i.e., for $L_0$ and $L^\star$, respectively). Values above the dashed line indicate that the true distribution is outside the calibrated ambiguity set. The bootstrap-coverage penalty preserves the empirical coverage behavior of the initial geometry $L_0$, whereas removing the penalty with $\lambda_p = 0$ leads to pronounced undercoverage (grey).

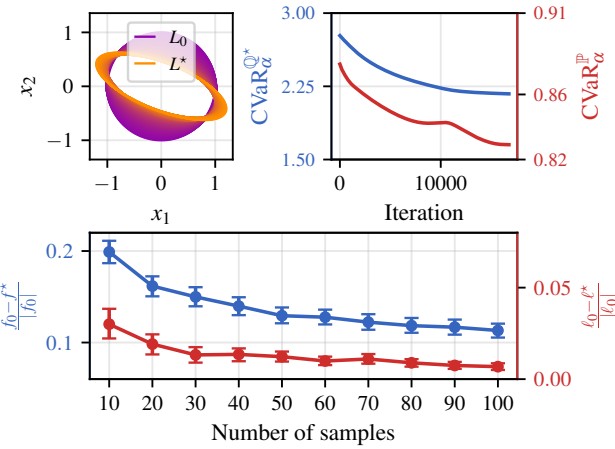

*Figure 3.* Results of the bilevel optimization. Change in transportation cost parametrization (top left), convergence plot over iterations (top right), average improvement over multiple experiments w.r.t. number of samples (bottom)

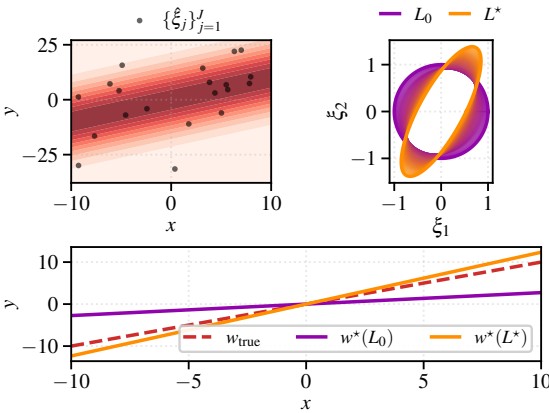

*Figure 4.* Underlying true distribution $\mathbb{P}$ (red contour) and samples (black), unit-cost ellipses defined by the matrices $L_i$ in the upper-level optimization, and optimal decisions before ($w^\star(L_0)$) and after ($w^\star(L^\star)$) optimization.

samples. Both metrics decrease across iterations, indicating a reduction in the decision's conservatism. We corroborate our results on 10 independent experiments, each using a different true distribution $\mathbb{P}$ from which 10 datasets are generated. To evaluate the proposed method, we consider the same relative improvement metrics used before, adapted to the linear regression loss function. Results, shown in Figure 5 (bottom), suggest once again that the proposed method reduces, on average, the conservatism of the decision.

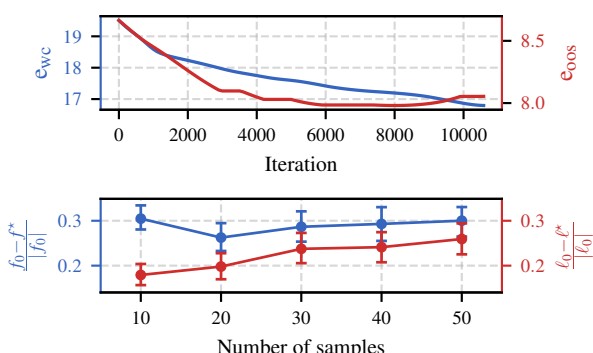

*Figure 5.* Expected absolute error on the worst-case distribution $e_{wc}$ and expected absolute error on the true data generating process $e_{oos}$ approximated with 10 million samples (top). Average improvement over multiple experiments w.r.t. the number of samples (bottom).

## 7. Concluding Remarks and Limitations

**Discussion.** The proposed method is most beneficial when a standard isotropic transport cost allocates robustness budget to directions that are weakly relevant for the downstream decision. In such cases, learning the transportation geometry allows the ambiguity set to stretch along nuisance directions while remaining tighter along decision-relevant directions, thereby reducing conservatism without changing the initially calibrated radius. Appendix G provides a pedagogical nuisance-feature example illustrating this regime.

**Limitations.** A key limitation of our work lies in the absence of theoretical guarantees for the coverage constraint (Eq. 11b), which we currently enforce heuristically through a bootstrapping procedure. Although our experiments highlight the practical effectiveness and robustness of the proposed algorithm, providing a rigorous lower bound on the probability of covering the true distribution under resampling-based methods remains an open challenge. Addressing this requires a deeper understanding of the interplay between $J$ and $n_{\mathrm{b}}$, as preliminarily discussed in (Summers & Kamgarpour, 2021), and represents an important direction for future research.

**Outlook.** In view of the growing popularity of the OT-DRO model, the results in this work carry important practical implications: we expect our end-to-end pipeline to result in less conservative, yet reliable decisions across several domains, including finance and machine learning, as demonstrated in Section 6.

## Acknowledgements

The authors thank the anonymous reviewers for their constructive feedback. This work was supported as a part of NCCR Automation, a National Centre of Competence in Research, funded by the Swiss National Science Foundation (grant number 51NF40_225155).

## Impact Statement

This paper presents work aimed at advancing the field of machine learning, particularly distributionally robust optimization. There are not many potential societal consequences of our work that we feel must be highlighted here.

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

# Appendix Contents

# A. Finite-Dimensional DRO Reformulation as a Conic Program

## A.1. General Reformulation

We provide a finite-dimensional convex reformulation for a general instance of the OT-DRO problem (1) (and of its dual) under mild regularity conditions.

### A.1.1. PRIMAL PROBLEM

Consider the primal OT-DRO problem

$$\inf_{w \in \mathcal{W}} \sup_{\mathbb{Q} \in \mathcal{B}_\varepsilon(\hat{\mathbb{P}})} \mathbb{E}_{\xi \sim \mathbb{Q}} \left[ \ell(w, \xi) \right]. \tag{21}$$

with empirical nominal distribution $\hat{\mathbb{P}} = \frac{1}{J} \sum_{j=1}^{J} \delta_{\hat{\xi}_j}$, and OT ambiguity set defined as

$$\mathcal{B}_\varepsilon(\hat{\mathbb{P}}) = \left\{ \mathbb{Q} \in \mathcal{P}(\Xi) \mid d(\mathbb{Q}, \hat{\mathbb{P}}) \leq \varepsilon \right\}, \tag{22}$$

where $d(\mathbb{Q}, \hat{\mathbb{P}}) : \mathcal{P}(\Xi) \times \mathcal{P}(\Xi) \to [0, +\infty)$ is defined as

$$d(\mathbb{Q}, \hat{\mathbb{P}}) = \inf_{\pi \in \Pi(\mathbb{Q}, \hat{\mathbb{P}})} \int_{\Xi \times \Xi} \kappa(\xi, \hat{\xi}) \, d\pi(\xi, \hat{\xi}),$$

for a certain transportation cost $\kappa(\xi_1, \xi_2) : \Xi \times \Xi \to [0, +\infty)$. To provide a convex finite-dimensional reformulation of (21), some regularity conditions need to be satisfied (Shafiee et al., 2025).

**Assumption A.1** (Transportation cost). *The transportation cost has to satisfy the properties:*

   *i*  $\kappa(\xi_1, \xi_2) \geq 0$ *always, and* $\kappa(\xi_1, \xi_2) = 0 \iff \xi_1 = \xi_2$.

  *ii.*  $\kappa(\xi_1, \xi_2)$ *is lower semicontinuous in* $(\xi_1, \xi_2)$ *and convex in its first argument.*

 *iii.*  *There exists a reference point* $\hat{\xi} \in \mathbb{R}^d$ *such that* $\mathbb{E}_{\xi \sim \hat{\mathbb{P}}}[\kappa(\xi, \hat{\xi})] < +\infty$.

 *iv.*  *There exists a metric* $d(\xi_1, \xi_2)$ *on* $\mathbb{R}^d$ *with compact sublevel sets such that* $\kappa(\xi_1, \xi_2) \geq d^p(\xi_1, \xi_2)$ *for some* $p \in \mathbb{N}$.

Specifically, Assumptions A.1(iii-iv) allows to prove that the ambiguity set $\mathcal{B}_\varepsilon(\hat{\mathbb{P}})$ is weakly compact. Assumptions A.1(i-ii) automatically imply that $\kappa(\xi_1, \xi_2)$ is proper, convex, and lower semicontinuous in $\xi_1$ for any fixed $\xi_2$ and that it is proper, convex, and lower semicontinuous in $\xi_2$ for any fixed $\xi_1$.

Next, we assume the following properties for the loss function.

**Assumption A.2** (Loss function). *The loss function is representable as a point-wise maximum of finitely many functions*

$$\ell(w, \xi) = \max_{i \in I} \ell_i(w, \xi),$$

*where each* $\ell_i$ *is proper, convex, and lower semicontinuous in* $w$, *and* $-\ell_i$ *are proper, convex, and lower semicontinuous in* $\xi$.

Moreover, we enforce the following for the support set.

**Assumption A.3** (Support of the uncertainty). *The support set is representable as*

$$\Xi = \left\{ \xi \in \mathbb{R}^d \mid f_r(\xi) \leq 0 \ \forall r \in [R] \right\},$$

*where each $f_r$ is proper, convex, and lower semicontinuous.*

The finite-dimensional convex reformulation of (21) relies on the following technical condition.

**Assumption A.4** (Slater conditions). *It holds:*

i. *For every $j \in [J], \hat{\xi}_j \in \mathrm{relint}(\mathrm{dom}(\kappa(\cdot, \hat{\xi}_j)))$ is a Slater point for the support set $\Xi$.*

ii. *The feasible set $\mathcal{W}$ admits a Slater point.*

Let Assumptions A.1, A.2, A.3, and A.4 hold. Further, let $\varepsilon > 0$ and assume $\mathcal{W}$ is compact and convex. Then, the primal OT-DRO problem (21) has the same infimum as the following finite-dimensional convex optimization problem (Shafiee et al., 2025, Proposition 2.13):

$$
\begin{aligned}
\inf \quad & \lambda \varepsilon + \frac{1}{J} \sum_{j \in [J]} s_j \\
\text{subject to} \quad & w \in \mathcal{W}, \lambda \in \mathbb{R}_+, \tau_{ijr} \in \mathbb{R}_+, s_j \in \mathbb{R}, a_{ij}^\ell, a_{ij}^c, a_{ijr}^f \in \mathbb{R}^d \quad \forall i \in [I], \ j \in [J], \ r \in [R] \\
& (-\ell_i)^{*2}(w, a_{ij}^l) + \lambda \kappa^{*1}\left( \frac{a_{ij}^c}{\lambda}, \hat{\xi}_j \right) + \sum_{r \in [R]} \tau_{ijr} f_r^*\left( \frac{a_{ijr}^f}{\tau_{ijr}} \right) \leq s_j \quad \forall i \in [I], \ j \in [J] \\
& a_{ij}^\ell + a_{ij}^c + \sum_{r \in [R]} a_{ijr}^f = 0 \quad \forall i \in [I], \ j \in [J].
\end{aligned}
\tag{23}
$$

Here, for any function of two arguments $f(\cdot, \cdot)$, we use $f^{*1}$ and $f^{*2}$ to denote the convex conjugate of the function with respect to its first and second argument while keeping the other argument fixed. A superscript $f^*$ on a function in a single argument denotes the convex conjugate.

A.1.2. DUAL PROBLEM

Similarly, the dual OT-DRO problem

$$
\sup_{\mathbb{Q} \in \mathcal{B}_\varepsilon(\hat{\mathbb{P}})} \inf_{w \in \mathcal{W}} \mathbb{E}_{\xi \sim \mathbb{Q}} [\ell(w, \xi)]
\tag{24}
$$

admits a tractable reformulation under certain regularity conditions. Specifically, in addition to Assumptions A.1, A.2, A.3, and A.4, we require the following.

**Assumption A.5** (Dual regularity conditions). *One of the following three conditions has to be satisfied: (i) $\mathbb{E}_{\xi \sim \mathbb{Q}}[\ell(w, \xi)]$ is inf-compact in $w \in \mathcal{W}$ for some $\mathbb{Q} \in \mathcal{B}_\varepsilon(\hat{\mathbb{P}})$, (ii) $\Xi$ is compact, or (iii) $\kappa(\cdot, \hat{\xi})$ grows superlinearly in its first argument.*

**Assumption A.6** (Feasible decision set). *The feasible set is representable as*

$$\mathcal{W} = \{ w \in \mathbb{R}^k \mid g_l(w) \leq 0 \ \forall l \in [L]\},$$

*where each $g_l$ is proper, convex, and lower semicontinuous.*

Let all the above assumptions hold and let $\varepsilon > 0$. Then, the dual DRO problem (24) has the same supremum as the following

finite-dimensional convex optimization problem (Shafiee et al., 2025, Proposition 2.15):

$$
\begin{aligned}
\max \quad & -\sum_{i \in [I]} \sum_{j \in [J]} q_{ij} \ell_i^{*1}\left(\alpha_{ij}/q_{ij}, \ \hat{\xi}_j + b_{ij}/q_{ij}\right) - \sum_{l \in [L]} \nu_l g_l^*(\beta_l/\nu_l) \\
\text{subject to} \quad & q_{ij}, \ \nu_l \in \mathbb{R}_+, b_{ij}, \alpha_{ij}, \beta_l \in \mathbb{R}^k \quad \forall i \in [I], j \in [J], l \in [L] \\
& f_r\left(\hat{\xi}_j + b_{ij}/q_{ij}\right) \leq 0 \quad \forall i \in [I], j \in [J], r \in [R] \\
& \sum_{i \in [I]} q_{ij} = p_j \quad \forall j \in [J] \\
& \sum_{i \in [I]} \sum_{j \in [J]} \alpha_{ij} + \sum_{l \in [L]} \beta_l = 0 \\
& \sum_{i \in [I]} \sum_{j \in [J]} q_{ij} \kappa(\hat{\xi}_j + b_{ij}/q_{ij}, \hat{\xi}_j) \leq \varepsilon.
\end{aligned}
\tag{25}
$$

The optimal variables of this problem $\left\{q_{ij}^\star, \ b_{ij}^\star, \ \alpha_{ij}^\star\right\}_{i,j}$ and $\{\nu_l^\star, \ \beta_l^\star\}_l$ can then be used to construct $\mathbb{Q}^\star$ as follows. Consider $\mathcal{I}_j^+ = \{i \in [I] \mid q_{ij}^\star > 0\}, \mathcal{I}_j^0 = \{i \in [I] \mid q_{ij}^\star = 0, \ b_{ij}^\star = 0\}$, and $\mathcal{I}_j^\infty = \{i \in [I] \mid q_{ij}^\star = 0, \ b_{ij}^\star \neq 0\}$. Then

$$
\mathbb{Q}^\star = \sum_{j \in [J]} \sum_{i \in \mathcal{I}_j^+} q_{ij}^\star \delta_{\hat{\xi}_j + b_{ij}^\star/q_{ij}^\star} \quad \text{if } \mathcal{I}_j^\infty = \emptyset \ \forall j \in [J]
$$

If there is a $j$ for which $\mathcal{I}_j^\infty$ is not empty, i.e., there exist $i, j$ such that $q_{ij}^\star = 0$ but $b_{ij}^\star \neq 0$, then only an asymptotic sequence of probability distributions can be generated that converges to the optimal solution of the dual DRO problem (24). For $n \to \infty$, this construction reads (Shafiee et al., 2025):

$$
\mathbb{Q}^\star \underset{n \to \infty}{\longleftarrow} \mathbb{Q}(n) = \sum_{j \in [J]} \sum_{i \in \mathcal{I}_j^+ \cup \mathcal{I}_j^\infty} \begin{cases} q_{ij}^\star (1 - |\mathcal{I}_j^\infty|/n) \ \delta_{\hat{\xi}_j + b_{ij}^\star/q_{ij}^\star} & \text{if } i \in \mathcal{I}_j^+ \\ \frac{p_j}{n} \ \delta_{\hat{\xi}_j + n \, b_{ij}^\star/p_j} & \text{if } i \in \mathcal{I}_j^\infty \end{cases}
$$

## A.2. Reformulation of the Portfolio Optimization Problem

This section provides convex reformulations for distributionally robust portfolio optimization with parametrized transportation costs across different problem settings.

### A.2.1. DISCRETE REFERENCE DISTRIBUTION

In case of discrete nominal distributions, we can directly invoke the results from Appendix A.1. The conjugate function of the objective in the case of a bilinear loss function, as in the portfolio optimization example, i.e.,

$$
-\ell(x, y) = x^\top y
$$

evaluates as

$$
(-\ell)^{*2}(x, y) = \sup_{\zeta \in \mathbb{R}^k} \left\{\zeta^\top(y - x)\right\} = \begin{cases} 0 & \text{if } y = x \\ \infty & \text{otherwise} \end{cases}.
\tag{26}
$$

**type-1 Mahalanobis distance.** We consider a parametrized transportation cost of the form $\kappa(\xi_1, \xi_2; \theta) = \|L^\top(\xi_1 - \xi_2)\|_2$ with $\theta = L \in \mathbb{L}_{++}^k$, and let $\Xi = \mathbb{R}^d$ for $d = k$. Then, the conjugate function of

$$
\kappa(y, a; \theta) = \|L^\top(y - a)\|_2,
$$

is given by

$$\kappa^{*1}(y, a) = \sup_{x \in \mathbb{R}^d} \left\{ x^\top y - \kappa(x, a) \right\} \tag{27}$$

$$= \sup_{x \in \mathbb{R}^d} \left\{ x^\top y - \left\| L^\top (x - a) \right\|_2 \right\} \tag{28}$$

$$= a^\top y + \sup_{z \in \mathbb{R}^d} \left\{ z^\top y - \left\| L^\top z \right\|_2 \right\} \tag{29}$$

$$= a^\top y + \sup_{w \in \mathbb{R}^d} \left\{ y^\top L^{-\top} w - \left\| w \right\|_2 \right\} \tag{30}$$

$$= \begin{cases} a^\top y & \text{if } \left\| L^{-1} y \right\|_2 \leq 1 \\ \infty & \text{otherwise} \end{cases}. \tag{31}$$

where the first and second equality follow from the definition of convex conjugate and from the parametrized transportation cost, respectively; the third one is obtained by introducing the change of variables $z = x - a$; and the fourth one by letting $w = L^\top z$. As for the last equality, we proceed as follows. Let $w = \alpha \cdot \left( L^{-1} y \right) + \beta \cdot (u)$, where the vector $u$ is orthogonal to $L^{-1} y$ and $u$ has length one, the objective function inside the supremum of (30) follows as

$$y^\top L^{-\top} w - \left\| w \right\|_2 = \alpha y^\top L^{-\top} L^{-1} y + \beta \left( u \right)^\top L^{-1} y - \left\| \alpha L^{-1} y + \beta u \right\|_2 = f_{\sup}(\alpha, \beta).$$

However, as $u$ is orthogonal to $L^{-1} y$ and thus $(u)^\top L^{-1} y = 0$, we can simply choose $\beta^\star = 0$, i.e., $w^\star$ has to be aligned with $L^{-1} y$. At this point, we turn our attention to $\alpha$:

$$f_{\sup}(\alpha) = \alpha \left\| L^{-1} y \right\|_2^2 - |\alpha| \left\| L^{-1} y \right\|_2$$
$$= \left\| L^{-1} y \right\|_2 \left( \alpha \left\| L^{-1} y \right\|_2 - |\alpha| \right)$$

We distinguish two cases:

- $\alpha \geq 0$: $f_{\sup}(\alpha) = \alpha \left\| L^{-1} y \right\|_2 \left( \left\| L^{-1} y \right\|_2 - 1 \right)$
  The objective function $f_{\sup}(\alpha)$ in this region is unbounded above if $\left\| L^{-1} y \right\|_2 - 1 > 0$.
  If $\left\| L^{-1} y \right\|_2 \leq 1$, then the supremum is zero (as then $f_{\sup}(\alpha) \leq 0$ for all $\alpha \geq 0$).

- $\alpha < 0$: $f_{\sup}(\alpha) = \alpha \left\| L^{-1} y \right\|_2 \left( \left\| L^{-1} y \right\|_2 + 1 \right)$
  The objective function is a linear function in $\alpha$ with a positive slope. Its value is always negative (as $\alpha < 0$), which means that the optimal value of $\alpha$ is $\alpha^\star = 0$, and thus $f_{\sup}^\star = 0$.

Combining both results directly leads to (31). The resulting reformulation for the distributionally robust portfolio optimization problem in this setting reads

$$\inf \quad \lambda \varepsilon + \frac{1}{J} \sum_{j \in [J]} s_j$$

$$\text{subject to} \quad w \in \mathcal{W}, \lambda \in \mathbb{R}_+, s_j \in \mathbb{R} \quad \forall j \in [J]$$

$$- w^\top \hat{\xi}_j \leq s_j \quad \forall j \in [J]$$

$$\left\| L^{-1} w \right\|_2 \leq \lambda.$$

Next, we introduce the auxiliary variable $u$ such that $Lu = w \implies u = L^{-1} w$. Further, we introduce the variables $z_j = s_j + w^\top \hat{\xi}_j \geq 0$ and recognize that $z_j = 0 \; \forall j$ at optimality. This leads to

$$\min_{\lambda, w, u} \quad \lambda \varepsilon - w^\top \frac{1}{J} \sum_{j \in [J]} \hat{\xi}_j$$

$$\text{subject to} \quad w \in \mathbb{R}_+^k, \lambda \in \mathbb{R}_+$$

$$\mathbf{1}^\top w = 1$$

$$Lu = w \tag{32}$$

$$\left\| u \right\|_2 \leq \lambda.$$

In turn, (32) can be cast as conic program in standard form (4) by using the following definitions:

$$s = (s_1, s_2, s_3, s_4) \in \mathcal{K} := \{0\} \times \{0\}^k \times \mathbb{R}_+^k \times \text{SOC}(k+1) \tag{33}$$

$$x = (w, \lambda, u) \in \mathbb{R}^{k+1+k} \tag{34}$$

$$c = (-\frac{1}{J} \sum_{j \in [J]} \hat{\xi}_j, \, \varepsilon, \, 0, \, ..., \, 0) \tag{35}$$

$$b = (1, \, 0, \, ..., \, 0) \tag{36}$$

$$A = \begin{bmatrix} \mathbf{1}_k^\top & 0 & 0 \\ -I_k & 0 & L \\ -I_k & 0 & 0 \\ 0 & -1 & 0 \\ 0 & 0 & -I_k \end{bmatrix}. \tag{37}$$

**type-2 Mahalanobis distance.** We consider a different parametrization for the transportation cost given by $\kappa(x, y; \theta) = \left\| L^\top (x - y) \right\|_2^2$, which admits the conjugate

$$\kappa^{*1}(y, a) = \sup_{x \in \mathbb{R}^d} \left\{ x^\top y - \left\| L^\top (x - a) \right\|_2^2 \right\}$$
$$= a^\top y + \frac{1}{4} y^\top (LL^\top)^{-1} y.$$

Again, we invoke the results of Appendix A.1, letting $\Xi = \mathbb{R}^d$ and using the conjugate of the bilinear loss function from (26). Now, however, we deliberately square the inequality in the definition of the ambiguity set (22). We obtain the following convex problem:

$$\min \quad \lambda \varepsilon^2 + \frac{1}{J} \sum_{j \in [J]} s_j$$
$$\text{subject to} \quad w \in \mathcal{W}, \lambda \in \mathbb{R}_+, s_j \in \mathbb{R} \quad \forall j \in [J]$$
$$\frac{1}{4\lambda} w^\top L^{-\top} L^{-1} w - \hat{\xi}_j^\top w \le s_j \quad \forall j \in [J].$$

Towards reformulating it as a conic program in standard form (4), we introduce the variables $t_j = \hat{\xi}_j^\top w + s_j$ as well as the vector $z$ such that $w = Lz \implies z = L^{-1}w$, yielding

$$\min \quad \lambda \varepsilon^2 + \frac{1}{J} \sum_{j \in [J]} s_j$$
$$\text{subject to} \quad w \in \mathcal{W}, \lambda \in \mathbb{R}_+, t_j \in \mathbb{R}, z \in \mathbb{R}^k, s_j \in \mathbb{R} \quad \forall j \in [J]$$
$$t_j = s_j + \hat{\xi}_j^\top w \quad \forall j \in [J]$$
$$w = Lz$$
$$z^\top z \le 4\lambda t_j \quad \forall j \in [J].$$

Using the second-order cone representation (Ben-Tal & Nemirovski, 2001):

$$z^\top z \le t \cdot a, t \ge 0, a \ge 0 \quad \Longleftrightarrow \quad \left\| (2z, t - a) \right\|_2 \le t + a, t \ge 0, a \ge 0,$$

we finally get

$$
\begin{aligned}
\min \quad & \lambda \varepsilon^2 + \frac{1}{J} \sum_{j \in [J]} (t_j - \hat{\xi}_j^\top w) \\
\text{subject to} \quad & w \in \mathcal{W}, \lambda \in \mathbb{R}_+, u_j \in \mathbb{R}^{k+1}, t_j \in \mathbb{R}, v_j \in \mathbb{R} \quad \forall j \in [J] \\
& v_j = 4\lambda + t_j \quad \forall j \in [J] \\
& \|u_j\|_2 \le v_j \quad \forall j \in [J] \\
& w = Lz \\
& u_j = \begin{bmatrix} 2z \\ 4\lambda - t_j \end{bmatrix} \quad \forall j \in [J] .
\end{aligned}
\tag{38}
$$

In turn, (38) can be readily written in standard form (4) by using the following definitions:

$$
\begin{aligned}
s &= (s_1, s_2, v_1, u_1, ..., v_J, u_J) \in \mathcal{K} := \{0\}^{1+k} \times \mathbb{R}_+^{k+1} \times \{\mathrm{SOC}(k+2)\}^J \\
x &= (t_1, ..., t_J, w, \lambda, z) \in \mathbb{R}^{J+k+1+k} \\
c &= \left( \tfrac{1}{J}, ..., \tfrac{1}{J}, -\tfrac{1}{J} \sum_{j \in [J]} \hat{\xi}_j, \varepsilon^2, 0, ..., 0 \right) \\
b &= (1, 0, ..., 0)
\end{aligned}
$$

$$
A = \left[ \begin{array}{cccc}
A_{11} & A_{12} & A_{13} & A_{14} \\
A_{21} & A_{22} & A_{23} & A_{24} \\
A_{31} & A_{32} & A_{33} & A_{34} \\
\hdashline
A_{41} & A_{42} & A_{43} & A_{44} \\
A_{51} & A_{52} & A_{53} & A_{54} \\
\vdots & \vdots & \vdots & \vdots \\
A_{(5+J-1)1} & A_{(5+J-1)2} & A_{(5+J-1)3} & A_{(5+J-1)4}
\end{array} \right] \in \mathbb{R}^{(1+k+k+1+J[k+2]) \times (J+k+1+k)}
$$

where

$$
A_{12} = \mathbf{1}^\top, \; A_{22} = -I_{k \times k}, \; A_{24} = L, \; A_{32} = -I_{k \times k}, \; A_{43} = -1,
$$

and the part below the dashed line corresponds to the $J$ second-order cone constraints, where

$$
\begin{aligned}
A_{(5+j-1)1}[:,j] &= (-1, 0, ..., 0, +1) \in \mathbb{R}^{(2+k) \times 1} \quad \forall j \in \{1, ..., J\} \\
A_{(5+j-1)3} &= (-4, 0, ..., 0, -4) \in \mathbb{R}^{(2+k) \times 1} \quad \forall j \in \{1, ..., J\}
\end{aligned}
$$

$$
A_{(5+j-1)4} = \begin{bmatrix}
0 & \cdots & 0 \\
-2 & & 0 \\
& \ddots & \\
0 & & -2 \\
0 & \cdots & 0
\end{bmatrix} \in \mathbb{R}^{(2+k) \times k} \quad \forall j \in \{1, ..., J\}
$$

and $X[:,j]$ denotes the $j$-th column of $X$. The rest of $A$ are zeros.

### A.2.2. GAUSSIAN REFERENCE DISTRIBUTION

Next, we turn attention to the setting where the nominal distribution is Gaussian rather than empirical, i.e., $\hat{\mathbb{P}} = \mathcal{N}(\hat{\mu}, \hat{\Sigma})$ for $\hat{\mu}, \hat{\Sigma}$ known. In this setting, by extending (Nguyen et al., 2025, Theorem 5) to the parametrized transportation cost $\kappa(\xi_1, \xi_2; \theta) = \|L^\top (\xi_1 - \xi_2)\|_2^2$ with $\theta = L \in \mathbb{L}_{++}^{d=k}$, we obtain

$$
\begin{aligned}
& \min_{w \in \mathcal{W}} \max_{\mathbb{Q} \in \mathcal{B}_\varepsilon(\hat{\mathbb{P}})} \mathcal{R}_{\xi \sim \mathbb{Q}} \left( -w^\top \xi \right) \\
&= \min_{w \in \mathcal{W}} -\hat{\mu}^\top w + \alpha \sqrt{w^\top \hat{\Sigma} w} + \varepsilon \sqrt{1 + \alpha^2} \sqrt{w^\top (LL^\top)^{-1} w},
\end{aligned}
\tag{39}
$$

where $\alpha$ is a risk coefficient that only depends on $\gamma$, thus it is a constant. To bring (39) into the conic standard form (44), we introduce auxiliary scalar variables $u$, $v$, and vector variables $z$ and $q$. We obtain:

$$
\begin{aligned}
\min_{w,u,v,q,z} \quad & -\hat{\mu}^\top w + \alpha u + \varepsilon \sqrt{1+\alpha^2}\, v \\
\text{subject to} \quad & \|z\|_2 \le u \\
& \|q\|_2 \le v \\
& w = Lq \\
& \sqrt{\hat{\Sigma}}\, w = z \\
& \mathbf{1}^\top w = 1 \\
& w \succeq 0,
\end{aligned}
$$

which can now be readily cast into the standard form (4) using the conic slack vector

$$
s = (s_1, \bar{w}, (\bar{u}, z), (\bar{v}, \bar{q})) \in \mathcal{K} := \{0\}^{k+1} \times \mathbb{R}^k_+ \times \mathrm{SOC}(k+1) \times \mathrm{SOC}(k+1)
$$

and the primal variable vector $x = (w, u, v, q) \in \mathbb{R}^{k+1+1+k}$. The corresponding problem data matrix and vectors are given by

$$
c = \left( -\hat{\mu},\ \alpha,\ \varepsilon\sqrt{1+\alpha^2},\ 0 \dots 0 \right) \tag{40}
$$

$$
b = (1,\ 0 \dots 0) \tag{41}
$$

$$
A = \begin{bmatrix}
\mathbf{1}^\top & 0 & 0 & 0 \\
-I_k & 0 & 0 & L \\
-I_k & 0 & 0 & 0 \\
0 & -1 & 0 & 0 \\
-\sqrt{\hat{\Sigma}} & 0 & 0 & 0 \\
0 & 0 & -1 & 0 \\
0 & 0 & 0 & -I_k
\end{bmatrix}. \tag{42}
$$

**Worst-case moments.** Analogously to (Nguyen et al., 2025, Theorem 5), we can then compute the worst-case moments as follows:

$$
\mu_L^\star = \hat{\mu} - \frac{\rho}{\sqrt{1+\alpha^2}\|w\|_{(LL^\top)^{-1}}} (LL^\top)^{-1} w,
$$

$$
\Sigma_L^\star = \left( I + \frac{\lambda^\star (LL^\top)^{-1} w w^\top}{\gamma^\star - \lambda^\star w^\top (LL^\top)^{-1} w} \right) \hat{\Sigma} \left( I + \frac{\lambda^\star w w^\top (LL^\top)^{-1}}{\gamma^\star - \lambda^\star w^\top (LL^\top)^{-1} w} \right),
$$

where

$$
\gamma^\star = \frac{1}{2\rho} \sqrt{1+\alpha^2} \|w\|_{(LL^\top)^{-1}}
$$

and

$$
\lambda^\star = \left( \frac{w^\top (LL^\top)^{-1} w}{\gamma^\star} + \frac{2}{\alpha} \sqrt{w^\top \hat{\Sigma} w} \right)^{-1}.
$$

### A.3. Reformulation of the Linear Regression Problem

In this section, we provide convex reformulations for the distributionally robust linear regression problem with parametrized transportation cost in different settings.

A.3.1. TYPE-1 MAHALANOBIS DISTANCE AND ABSOLUTE ERROR

We provide here a reformulation of the distributionally robust linear regression problem under an $\ell_1$-loss function (see definition in Section 6.2) and show that it can be cast as a conic program. This loss is generally more forgiving (hence, robust) to large residuals than the squared loss, and therefore makes an interesting case study. We further consider a parametrized transportation cost of the form $\kappa(\xi_1, \xi_2; \theta) = \|L^\top(\xi_1 - \xi_2)\|_2$ with $\theta = L \in \mathbb{L}^d_{++}$. By following a similar reasoning as in (Chen & Paschalidis, 2020, Theorem 4.2.1 (p.73)), we obtain the following equivalence:

$$\min_{w \in \mathbb{R}^k} \max_{\mathbb{Q} \in \mathcal{B}_\varepsilon(\hat{\mathbb{P}})} \mathbb{E}_{\xi \sim \mathbb{Q}}(\ell_1(w, \xi)) = \min_{w \in \mathbb{R}^k} \frac{1}{J} \sum_{j=1}^J \left| (-w, 1)^\top \hat{\xi}_j \right| + \varepsilon \|(-w, 1)\|_{(LL^\top)^{-1}}, \tag{43}$$

with $w \in \mathbb{R}^k$ and $\hat{\xi}_j \in \mathbb{R}^{d=k+1}$. We can write (43) equivalently as:

$$
\begin{aligned}
\min \quad & \frac{1}{J} \mathbf{1}^\top v + \varepsilon u \\
\text{subject to} \quad & a_j = v_j - y_j + w^\top x_j \\
& b_j = v_j + y_j - w^\top x_j \\
& \|z\|_2 \leq u \\
& a \geq 0, b \geq 0 \\
& Lz + (w, 0) = (0, 1).
\end{aligned}
$$

We can now readily cast the above formulation in the standard form (4) using the conic slack vector

$$s = (0, a, b, (\bar{u}, \bar{z})) \in \mathcal{K} := \{0\}^d \times \mathbb{R}^J_+ \times \mathbb{R}^J_+ \times \mathrm{SOC}(d+1)$$

and the primal variable vector $x = (v, u, z, w) \in \mathbb{R}^{J+1+d+k}$. The corresponding problem data matrix and vectors are

$$
\begin{aligned}
c &= (1/J, \, ..., \, 1/J, \, \varepsilon, \, 0 \,...\, 0) \\
b &= (0, \, ..., \, 0, \, 1, \, -y, \, y, \, 0 \,...\, 0) \\
A &= \begin{bmatrix} 0 & 0 & L & [I_k; 0] \\ -I_J & 0 & 0 & -X \\ -I_J & 0 & 0 & X \\ 0 & -e_1 & [0; -I_d] & 0 \end{bmatrix}
\end{aligned}
$$

where $X = [x_1, \ldots, x_J]^\top \in \mathbb{R}^{J \times k}$ is the data matrix and $e_1 = [1, 0, \ldots, 0]^\top$ is the first standard basis vector in $\mathbb{R}^d$.

Observe that (43) can be interpreted as a regularized linear regression, where the regression coefficients are penalized via $\varepsilon$ and $L$. Specifically, the regularizer could be seen as a control over the amount of ambiguity in the data and provides a rigorous theoretical foundation on why the $\ell_2$-regularizer prevents overfitting of the training data. The connection between robustness and regularization has been established in several works (Chen & Paschalidis, 2020).

A.3.2. TYPE-2 MAHALANOBIS DISTANCE AND SQUARED ERROR

In this section, we provide the reader with the reformulation of the distributionally robust linear regression when using the squared error $\ell_2(w, \xi) = \left((-w, 1)^\top \xi\right)^2$ and the parametrized transportation cost $\kappa(\xi_1, \xi_2; \theta) = \|L^\top(\xi_1 - \xi_2)\|_2^2$ with $\theta = L \in \mathbb{L}^d_{++}$. By invoking (Blanchet et al., 2019a, Proposition 2), we have the following equivalence:

$$\min_{w \in \mathbb{R}^k} \max_{\mathbb{Q} \in \mathcal{B}_\varepsilon(\hat{\mathbb{P}})} \mathbb{E}_{\xi \sim \mathbb{Q}}(\ell_2(w, \xi)) = \min_{w \in \mathbb{R}^k} \left( \sqrt{\frac{1}{J} \sum_{j=1}^J \left[(-w, 1)^\top \hat{\xi}_j\right]^2} + \varepsilon \|(-w, 1)\|_{(LL^\top)^{-1}} \right)^2$$

The final reformulation, after recognizing nonnegativity of the terms in the square and thus optimizing its square root, follows as

$$
\begin{aligned}
\min \quad & \lambda/\sqrt{J} + \varepsilon z \\
\text{subject to} \quad & \|q\|_2 \le z, \|a\|_2 \le \lambda \\
& Lq = v \\
& [X, y]\, v = a \\
& v_d = 1,
\end{aligned}
$$

where $v_d$ is the $d$-th entry of the vector $v$. This can now be readily cast into the standard form (4) using the conic slack vector

$$
s = (0, (\bar{\lambda}, \bar{a}), (\bar{z}, \bar{q})) \in \mathcal{K} := \{0\}^{d+1+J} \times \mathrm{SOC}(J+1) \times \mathrm{SOC}(d+1)
$$

and the primal variable vector $x = (v, \lambda, a, z, q) \in \mathbb{R}^{d+1+J+1+d}$. The corresponding problem data matrix and vectors are

$$
c = \left(0, 1/\sqrt{J}, 0 \dots 0,\ \varepsilon,\ 0 \dots 0\right)
$$

$$
b = e_{d+1}
$$

$$
A = \begin{bmatrix}
-I_d & 0 & 0 & 0 & L \\
[0 \dots 0\, 1] & 0 & 0 & 0 & 0 \\
[X, y] & 0 & -I_J & 0 & 0 \\
0 & -1 & 0 & 0 & 0 \\
0 & 0 & -I_J & 0 & 0 \\
0 & 0 & 0 & -1 & 0 \\
0 & 0 & 0 & 0 & -I_d
\end{bmatrix},
$$

where $e_{d+1}$ is the unit vector with 1 in entry $d + 1$. The resulting objective value must then be squared to recover the worst-case expectation.

## B. Functions Definable in an o-Minimal Structure

We provide here a brief definition of definable functions and sets, and refer the reader to the monograph (Coste, 1999) for additional information.

**Definition B.1** ((Coste, 1999, Definition 1.4-1.5)). *An o-minimal structure expanding the real closed field $\mathbb{R}$ is a collection $\mathcal{S} = (\mathcal{S}^n)_{n \in \mathbb{N}}$, where each $\mathcal{S}^n \subset \mathbb{R}^n$ satisfies the following*

1. *all algebraic subsets of $\mathbb{R}^n$ are contained in $\mathcal{S}^n$.*

2. *$\mathcal{S}^n$ is a Boolean subalgebra of $\mathbb{R}^n$.*

3. *If $A \in \mathcal{S}^m$ and $B \in \mathcal{S}^m$, then $A \times B \in \mathcal{S}^{n+m}$.*

4. *The projection onto the first $n$ coordinates of any $A \in \mathcal{S}^{n+1}$ belongs to $\mathcal{S}^n$.*

5. *The elements of $\mathcal{S}^1$ are precisely the finite unions of points and intervals.*

*The elements of $\mathcal{S}^n$ are called definable subsets of $\mathbb{R}^n$. A function is called definable if its graph is a definable set. Definable functions comprise the vast majority of functions commonly found in the field of optimization.*

The class of definable sets is large. In particular, all cones that are generally considered in the context of distributionally robust optimization (i.e., the exponential cone, the second-order cone, and the positive semidefinite cone) are all definable. Similarly, definable functions include almost all functions that are commonly found in the field of optimization (Bolte & Pauwels, 2021).

## C. Differentiating Through Conic Programs

Conic programming generally refers to problems where the feasible set is the intersection of an affine subspace and a nonempty closed convex cone (Agrawal et al., 2019; O'Donoghue et al., 2016; Busseti et al., 2019). In primal form, a conic program can be written as

$$
\begin{aligned}
\min_{x,s} \quad & c^\top x \\
\text{subject to} \quad & Ax + s = b \\
& s \in \mathcal{K}
\end{aligned}
\tag{44}
$$

where $x \in \mathbb{R}^n$, $s \in \mathbb{R}^m$ is a primal slack variable, and $\mathcal{K} \subseteq \mathbb{R}^m$ is a nonempty, closed, convex cone with dual cone $\mathcal{K}^* = \{y \in \mathbb{R}^m \mid \inf_{z \in \mathcal{K}} y^\top z \geq 0\} \subseteq \mathbb{R}^m$.

The dual of (44) is a conic program of the form

$$
\begin{aligned}
\min_{y} \quad & b^\top y \\
\text{subject to} \quad & A^\top y + c = 0 \\
& y \in \mathcal{K}^*
\end{aligned}
\tag{45}
$$

where $y \in \mathbb{R}^m$ is the dual variable.

In this section, we define the solution map $\mathcal{S}$ mapping the problem definition $(A, b, c)$ to the primal-dual solution $(x, y, s)$. Moreover, following (Bolte et al., 2021), we provide sufficient conditions under which $\mathcal{S}$ admits a conservative Jacobian and provide an expression for it.

### C.1. Necessary Optimality Conditions

Any primal-dual optimizers $(x, s, y)$ must satisfy the KKT conditions

$$
\begin{aligned}
0 &= A^\top y + c, & \text{(46a)} \\
s &= -Ax + b, & \text{(46b)} \\
0 &= s^\top y. & \text{(46c)}
\end{aligned}
$$

Following (Bolte et al., 2021), we can equivalently express (46) via

$$
\begin{aligned}
A^\top \Pi_{\mathcal{K}^*}(v) + c &= 0, & \text{(47a)} \\
-Au + v - \Pi_{\mathcal{K}^*}(v) + b &= 0, & \text{(47b)}
\end{aligned}
$$

where $v = y - s$, $u = x$, and $\Pi_{\mathcal{K}^*}$ denotes the projector to the closed convex cone $\mathcal{K}^*$. Letting $z = (u, v)$, condition (47) is equivalent to

$$
\mathcal{N}(z, A, b, c) := (Q(A, b, c) - I)\Pi(z) + V(b, c) + z = 0,
$$

where $\Pi$ denotes the projection onto $\mathbb{R}^n \times \mathcal{K}^*$, and

$$
Q(A, b, c) = \begin{bmatrix} 0 & A^\top \\ -A & 0 \end{bmatrix} \in \mathbb{R}^{N \times N}, \quad V(b, c) = \begin{bmatrix} c \\ b \end{bmatrix} \in \mathbb{R}^N,
$$

with $N = m + n$.

### C.2. The Solution Map

The solution map $\mathcal{S} : (A, b, c) \mapsto (x, y, s)$ is defined as the mapping from the optimization problem data $(A, b, c)$ to the vectors $(x, y, s)$ that satisfy the KKT conditions of the conic optimization problem, assuming such vectors are unique.

One way to define $\mathcal{S}$ is through the following composition of functions (Bolte et al., 2021):

$$
\mathcal{S}(A, b, c) = [\phi \circ \nu](A, b, c),
\tag{48}
$$

where

- $\nu : \mathbb{R}^{m \times n} \times \mathbb{R}^m \times \mathbb{R}^n \to \mathbb{R}^N$ is implicitly defined as $\mathcal{N}(\nu(A, b, c), A, b, c) = 0$;

- $\phi : \mathbb{R}^N \to \mathbb{R}^n \times \mathbb{R}^m \times \mathbb{R}^m$ is defined as $\phi(u, v) = (u, \Pi_{\mathcal{K}^*}(v), \Pi_{\mathcal{K}^*}(v) - v)$.

## C.3. Derivative of the Solution Map

The Jacobian of $\mathcal{S}$ can be obtained by computing the Jacobians of $\nu$ and $\phi$, and applying the chain rule of differentiation to (48).

**Proposition C.1** ((Bolte et al., 2021, Proposition 4)). *Assume $\Pi_{\mathcal{K}^*}$ is locally Lipschitz and definable with convex conservative Jacobian $\mathcal{J}_{\Pi_{\mathcal{K}^*}}$. Let $\mathcal{J}_{\mathcal{N}}$ be the convex conservative Jacobian of the residual $\mathcal{N}$. Assume that given any $z = \nu(A, b, c)$, and $[J_z, J_A, J_b, J_c] \in \mathcal{J}_{\mathcal{N}}(z, A, b, c)$, all the matrices $J_z$ are invertible. Then $\mathcal{S}$ is locally Lipschitz and definable with conservative Jacobian $\mathcal{J}_{\mathcal{S}}(A, b, c) = \mathcal{J}_\phi(\nu(A, b, c))\mathcal{J}_\nu(A, b, c)$, where*

$$\mathcal{J}_\nu(A, b, c) = \{-U^{-1}V : [U \ V] \in \mathcal{J}_{\mathcal{N}}(\nu(A, b, c), A, b, c)\}, \quad \mathcal{J}_\phi(z) = \begin{bmatrix} I & 0 \\ 0 & \mathcal{J}_{\Pi_{\mathcal{K}^*}}(v) \\ 0 & \mathcal{J}_{\Pi_{\mathcal{K}^*}}(v) - I \end{bmatrix}.$$

The residual map $\mathcal{N}$ can be obtained by simple addition and product operations starting from $(A, b, c)$ and $\Pi_{\mathcal{K}^*}$; therefore, it is always locally Lipschitz and definable in $(A, b, c)$ if the same holds for the projector $\Pi_{\mathcal{K}^*}$. We now prove that $\Pi$ is also locally Lipschitz and definable under the following, mild conditions.

**Assumption C.1.** *The set $\mathcal{K}$ is definable in an o-minimal structure.*

**Assumption C.2.** *The primal (44) and the dual (45) satisfy the metric regularity condition*

$$b \in \text{int}\{\text{span } A + \mathcal{K}\}, \quad c \in \text{int}\{A^\top \mathcal{K}^*\}. \tag{49}$$

Metric regularity ensures zero duality gap (Bonnans & Shapiro, 2000, Theorem 3.6), meaning that the KKT conditions are necessary and sufficient conditions of optimality. If $\text{int } \mathcal{K}$ is nonempty, then metric regularity of the primal problem is equivalent to *Slater's constraint qualification*, i.e., to the existence of some $\bar{x} \in \mathcal{Q}$ such that $G\bar{x} \in \text{int } \mathcal{K}$ (Bonnans & Shapiro, 2000, Proposition 2.106). Slater's constraint qualification is a mild assumption which is *generic* in linear conic programs like (44), that is, it holds for almost all combinations of problem parameters (Dür et al., 2012).

**Lemma C.1.** *Under Assumptions C.1 and C.2, let $(x, s, y)$ be a primal-dual optimizer of (44), and let $z = (x, y - s)$. Then $\Pi$ is locally Lipschitz and definable at $z$ and $\Pi(z) = (x, y)$.*

*Proof.* Under metric regularity, $z$ solves the primal-dual embedding, meaning that $y \in \mathcal{K}^*$. Since $\Pi(z) = \Pi(x, y - s) = (x, \Pi_{\mathcal{K}^*}(y - s))$, and the map $z \mapsto x$ is linear, we only have to prove that the projector $\Pi_{\mathcal{K}^*}$ is path-differentiable at $y - s$. The point $v = \Pi_{\mathcal{K}^*}(y - s)$ uniquely satisfies $y - s - v \in N_{\mathcal{K}^*}(v), v \in \mathcal{K}^*$. The set $N_{\mathcal{K}^*}(v) = \{w : \langle w, z - v \rangle \leq 0, \forall z \in \mathcal{K}^*\}$ is definable, since it is defined by a first order condition (Coste, 1999, Condition 4, page 12), and $\mathcal{K}^*$ is definable as an immediate consequence of Assumption C.1. Therefore, $v$ is the unique element of a definable set, and the set $\{(y, \Pi_{\mathcal{K}^*}(y)) : y \in \mathcal{K}^*\}$ is definable. This proves that $\Pi_{\mathcal{K}^*}$ is definable. Since projections to convex sets are Lipschitz, this concludes the proof. □

**Remark C.1.** *Proposition C.1 provides sufficient conditions under which $\mathcal{S}$ is locally Lipschitz and definable in $(A, b, c)$. Since $(A, b, c)$ are parameterized in $\theta$, to fulfill the assumption on $x^*(\theta)$ and $c$ in Assumption 5.1 we simply require that $(A, b, c)$ are Lipschitz continuous and definable functions of $\theta$, which is not a restrictive assumption.*

### C.3.1. COMPUTING THE FORWARD DERIVATIVE

Computing the forward derivative of the solution map amounts to evaluating the product $J_{\mathcal{S}}(A, b, c)(dA, db, dc)$ between an element $J_{\mathcal{S}}(A, b, c)$ of the conservative Jacobian $\mathcal{J}_{\mathcal{S}}(A, b, c)$ of $\mathcal{S}$ and a direction $(dA, db, dc)$. Using Proposition C.1 we have

$$(dx, dy, ds) = J_{\mathcal{S}}(A, b, c)(dA, db, dc) = J_\phi(z)J_\nu(A, b, c)(dA, db, dc)$$

where $J_\phi(z) \in \mathcal{J}_\phi(z)$, and $J_\nu(A, b, c) \in \mathcal{J}_\nu(A, b, c)$ can be computed by solving

$$J_\nu(A, b, c) = \underset{z}{\text{argmin}} \ \|J_1 x + J_2(dA, db, dc)\|^2, \quad [J_1, J_2] \in \mathcal{J}_{\mathcal{N}}(z, A, b, c). \tag{50}$$

Letting $J_\nu(A, b, c) = (\mathrm{d}u, \mathrm{d}v)$, the forward derivative $(\mathrm{d}x, \mathrm{d}y, \mathrm{d}s)$ of the primal-dual solution map can be obtained as follows

$$
\begin{bmatrix} \mathrm{d}x \\ \mathrm{d}y \\ \mathrm{d}s \end{bmatrix} = \begin{bmatrix} \mathrm{d}u \\ J_{\Pi_{\mathcal{K}^*}}(v)\mathrm{d}v \\ J_{\Pi_{\mathcal{K}^*}}(v)\mathrm{d}v - \mathrm{d}v \end{bmatrix}.
$$

### C.3.2. Computing the Adjoint Derivative

For reverse auto-differentiation, we are interested in computing the product between $J_{\mathcal{S}}(A, b, c)^\top$, with $J_{\mathcal{S}}(A, b, c) \in \mathcal{J}_{\mathcal{S}}(A, b, c)$ and a direction $(\mathrm{d}x, \mathrm{d}y, \mathrm{d}s)$

$$
(\mathrm{d}A, \mathrm{d}b, \mathrm{d}c) = J_{\mathcal{S}}(A, b, c)^\top (\mathrm{d}x, \mathrm{d}y, \mathrm{d}s) = J_\nu^\top (A, b, c)^\top J_\phi(z)^\top (\mathrm{d}x, \mathrm{d}y, \mathrm{d}s). \tag{51}
$$

To evaluate (51), we first compute

$$
\mathrm{d}z = J_\phi(z)^\top (\mathrm{d}x, \mathrm{d}y, \mathrm{d}s) = \begin{bmatrix} \mathrm{d}x \\ J_{\Pi_{\mathcal{K}^*}}^\top (v)(\mathrm{d}y + \mathrm{d}s) - \mathrm{d}s \end{bmatrix}.
$$

Next, we can obtain $\mathrm{d}s = J_{\mathcal{S}}(A, b, c)^\top \mathrm{d}z$ by solving

$$
\mathrm{d}s = \underset{ds}{\mathrm{argmin}} \ \|J_1 ds + J_2 \mathrm{d}z\|^2, \quad [J_1 \ J_2] \in \mathcal{J}_{\mathcal{N}}(z, A, b, c). \tag{52}
$$

## D. Differentiating Through Optimal Transport Problems

This appendix shows how a parametrized optimal transport problem can be computed and differentiated with respect to the transportation cost parameter. To this end, we focus as an example on the type-$p$ Mahalanobis distance with transportation cost $\kappa(\xi_1, \xi_2; \theta) = \|L^\top(\xi_1 - \xi_2)\|_2^p$ with $\theta = L \in \mathbb{L}_{++}^d$. Other possible transportation cost parametrizations follow a similar pattern. In particular, we will consider distance calculations between two discrete distributions (Appendix D.1) and between two Gaussian distributions (Appendix D.2).

### D.1. Parametrized Mahalanobis Distance Between Discrete Distributions

Given two discrete distributions, the parametrized Mahalanobis distance can be efficiently computed by solving a linear program.

**Fact D.1** (Calculation of the discrete Mahalanobis distance)**.** *The parametrized type-$p$ Mahalanobis distance $d(\mathbb{P}, \mathbb{Q}; L)$ between two discrete distributions $\mathbb{P} \triangleq \sum_{i=1}^I p_i \delta_{x_i}$ and $\mathbb{Q} \triangleq \sum_{j=1}^J q_j \delta_{y_j}$ can be calculated as the $p$-th root of the optimal value of the following linear program:*

$$
\begin{aligned}
d(\mathbb{P}, \mathbb{Q}; L)^p = \min_{\pi_{ij}} \quad & \sum_{i=1}^I \sum_{j=1}^J \|L^\top(x_i - y_j)\|_2^p \pi_{ij} \\
\textit{subject to} \quad & \sum_{i=1}^I \pi_{ij} = q_j \quad \forall j \in [J] \\
& \sum_{j=1}^J \pi_{ij} = p_i \quad \forall i \in [I] \\
& \pi_{ij} \geq 0 \quad \forall i \in [I], \ j \in [J],
\end{aligned}
\tag{53}
$$

*and $d(\mathbb{P}, \mathbb{Q}; L)^p = \sum_{i=1}^I \sum_{j=1}^J \tilde{c}_{ij}(L)\pi_{ij}^\star(L)$, where $\tilde{c}_{ij} : \mathbb{L}_{++}^d \to \mathbb{R}_+$ is defined as $\tilde{c}_{ij}(L) = \|L^\top(x_i - y_j)\|_2^p$, and $\pi_{ij}^\star(L) \in \mathbb{R}_+$ is the optimal probability mass associated to the indices $i, j$.*

The derivative of the parametrized Wasserstein distance with respect to the cost matrix $L$ for two discrete distributions is also efficiently computed as follows.

**Fact D.2** (Differentiation of the discrete Mahalanobis distance). *Suppose the solution map of Fact D.1 is differentiable at* $(\mathbb{P}, \mathbb{Q})$. *By the envelope theorem (Milgrom & Segal, 2002), the gradient of the type-$p$ Mahalanobis distance between two discrete distributions* $\mathbb{P} \triangleq \sum_{i=1}^{I} p_i \delta_{x_i}$ *and* $\mathbb{Q} \triangleq \sum_{j=1}^{J} q_j \delta_{y_j}$ *with respect to the parameter matrix* $L \in \mathbb{L}_{++}^d$ *is given by*

$$\frac{\partial d(\mathbb{P}, \mathbb{Q}; L)^p}{\partial L_{kl}} = \sum_{i=1}^{I} \sum_{j=1}^{J} \frac{\partial \tilde{c}_{ij}(L)}{\partial L_{kl}} \pi_{ij}^\star(L). \tag{54}$$

*By defining* $\Delta_{ij} = (x_i - y_j)$ *for all combinations of* $i \in [I]$ *and* $j \in [J]$ *it is possible to write*

$$\frac{\partial \tilde{c}_{ij}(L)}{\partial L} = \frac{\Delta_{ij} \Delta_{ij}^\top L}{\tilde{c}_{ij}(L)} \tag{55}$$

*for* $p = 1$ *(where at the singularity* $x_i = y_j$, *we set the gradient to zero, consistent with the vanishing limit of this ratio), and*

$$\frac{\partial \tilde{c}_{ij}(L)}{\partial L} = 2 \Delta_{ij} \Delta_{ij}^\top L \tag{56}$$

*for* $p = 2$.

The path-differentiability assumption in Assumption 5.1 is fulfilled under the assumptions of Proposition C.1. Practically, $d$ is almost everywhere differentiable and its conservative Jacobian $\mathcal{J}_d$ of $d(\cdot, \cdot; L)$ is almost everywhere equal to the gradient of $d$, which can be computed through Fact D.2.

### D.1.1. NONCONVEXITY OF THE PARAMETRIZED DISCRETE MAHALANOBIS DISTANCE

We show that the parametrized discrete Mahalanobis distance is generally nonconvex with respect to the parameter matrix $L$ by providing counterexamples for the case $p = 1$ and $p = 2$. For simplicity, let $d_p(\mathbb{P}, \mathbb{Q}; L)$ be the type-$p$ Wasserstein distance between the two discrete distributions $\mathbb{P}$ and $\mathbb{Q}$ with respect to the cost matrix $L$.

**type-1 discrete Mahalanobis distance**   Consider the following two discrete distributions:

$$\mathbb{P} \triangleq 0.4 \delta_{x_1} + 0.6 \delta_{x_2} \text{ with } x_1 = (0.7, 0.4) \text{ and } x_2 = (1.7, 1.0)$$
$$\mathbb{Q} \triangleq 0.5 \delta_{y_1} + 0.5 \delta_{y_2} \text{ with } y_1 = (1.8, 0.1) \text{ and } y_2 = (0.5, 1.4).$$

Next, consider

$$L_1 = \begin{bmatrix} 1 & 0 \\ 0.5 & 0.5 \end{bmatrix} \text{ and } L_2 = \begin{bmatrix} 0.5 & 0 \\ 1.0 & 1.0 \end{bmatrix}.$$

Solving the linear program in Fact D.1 yields

$$d_1(\mathbb{P}, \mathbb{Q}; L_1) = 0.6203,$$
$$d_1(\mathbb{P}, \mathbb{Q}; L_2) = 0.5036,$$
$$d_1(\mathbb{P}, \mathbb{Q}; 0.5 \cdot L_1 + 0.5 \cdot L_2) = 0.6820,$$

which indicates that

$$d_1(\mathbb{P}, \mathbb{Q}; 0.5 \cdot L_2 + 0.5 \cdot L_1) > 0.5 \cdot d_1(\mathbb{P}, \mathbb{Q}; L_1) + 0.5 \cdot d_1(\mathbb{P}, \mathbb{Q}; L_2),$$

thus proving the nonconvexity of the type-1 Mahalanobis distance.

**type-2 discrete Mahalanobis distance**   Consider the following two discrete distributions:

$$\mathbb{P} \triangleq 0.6 \delta_{x_1} + 0.4 \delta_{x_2} \text{ with } x_1 = (1.2, 1.9) \text{ and } x_2 = (0.1, 0.1)$$
$$\mathbb{Q} \triangleq 0.6 \delta_{y_1} + 0.4 \delta_{y_2} \text{ with } y_1 = (0.2, 1.4) \text{ and } y_2 = (1.4, 0.3).$$

Next, consider

$$L_1 = \begin{bmatrix} 1 & 0 \\ 0.5 & 0.5 \end{bmatrix} \text{ and } L_2 = \begin{bmatrix} 0.5 & 0 \\ 1.0 & 0.5 \end{bmatrix}.$$

Solving the linear program in Fact D.1 yields

$$d_2(\mathbb{P}, \mathbb{Q}; L_1) = 1.0578,$$
$$d_2(\mathbb{P}, \mathbb{Q}; L_2) = 0.9646,$$
$$d_2(\mathbb{P}, \mathbb{Q}; 0.5 \cdot L_1 + 0.5 \cdot L_2) = 1.1433,$$

which indicates that

$$d_2(\mathbb{P}, \mathbb{Q}; 0.5 \cdot L_2 + 0.5 \cdot L_1) > 0.5 \cdot d_2(\mathbb{P}, \mathbb{Q}; L_1) + 0.5 \cdot d_2(\mathbb{P}, \mathbb{Q}; L_2),$$

thus proving the nonconvexity of the type-2 Mahalanobis distance.

### D.2. Parametrized Gelbrich Distance

We define the parametrized Gelbrich distance as follows.

**Definition D.1** (Parametrized Gelbrich distance). *The parameterized Gelbrich distance between two mean-covariance pairs $(\mu_{\mathbb{Q}},\ \Sigma_{\mathbb{Q}})$ and $(\mu_{\mathbb{P}},\ \Sigma_{\mathbb{P}})$ in $\mathbb{R}^d \times \mathbb{S}_+^d$, where $\mathbb{S}_+^d$ denotes the set of positive semidefinite symmetric matrices, is defined as*

$$g((\mu_{\mathbb{Q}},\ \Sigma_{\mathbb{Q}}), (\mu_{\mathbb{P}},\ \Sigma_{\mathbb{P}}); L) := \left( \left\| L^\top (\mu_{\mathbb{Q}} - \mu_{\mathbb{P}}) \right\|^2 + \operatorname{Tr} \left[ (\Sigma_{\mathbb{Q}} + \Sigma_{\mathbb{P}}) LL^\top \right] \right.$$
$$\left. - 2 \operatorname{Tr} \left[ \left( \Sigma_{\mathbb{P}}^{\frac{1}{2}} LL^\top \Sigma_{\mathbb{Q}} LL^\top \Sigma_{\mathbb{P}}^{\frac{1}{2}} \right)^{\frac{1}{2}} \right] \right)^{\frac{1}{2}}, \tag{57}$$

*for the parameter matrix $L \in \mathbb{L}_{++}^d$.*

Note that for $\mathbb{P} \triangleq \mathcal{N}(\mu_{\mathbb{P}}, \Sigma_{\mathbb{P}})$ and $\mathbb{Q} \triangleq \mathcal{N}(\mu_{\mathbb{Q}}, \Sigma_{\mathbb{Q}})$, one has

$$g((\mu_{\mathbb{Q}},\ \Sigma_{\mathbb{Q}}), (\mu_{\mathbb{P}},\ \Sigma_{\mathbb{P}}); L) = d_2(\mathbb{Q}, \mathbb{P}; L). \tag{58}$$

The local Lipschitz continuity and definability assumption in Assumption 5.1 is immediately verified by the Gelbrich distance since the definition in Definition D.1 involves compositions of definable functions (square root, trace, transposition, product). Moreover, since the Gelbrich distance is continuously differentiable everywhere, its conservative Jacobian coincides with its gradient (Bolte & Pauwels, 2021, Theorem 1).

We can efficiently differentiate the parametrized Gelbrich distance with respect to the parameter matrix $L$ as follows. To shorten notation, we now write $C = (\mu_{\mathbb{Q}} - \mu_{\mathbb{P}})(\mu_{\mathbb{Q}} - \mu_{\mathbb{P}})^\top + \Sigma_{\mathbb{Q}} + \Sigma_{\mathbb{P}}$, $A = \sqrt{\Sigma_{\mathbb{Q}}}$, and $B = \sqrt{\Sigma_{\mathbb{P}}}$. As the derivative of the square root and the chain rule in the scalar case are well-known, we focus on the derivative of the following function in $L$

$$h(L) = \operatorname{Tr} \left[ CLL^\top \right] - 2 \operatorname{Tr} \left[ \sqrt{BLL^\top AALL^\top B} \right].$$

The derivative of the first term is $\frac{\partial}{\partial L} \operatorname{Tr} \left[ CLL^\top \right] = CL + C^\top L$ (Petersen & Pedersen, 2008). To compute the derivative of the second term, let $f(L) = \operatorname{Tr} \left[ \sqrt{BLL^\top AALL^\top B} \right]$ and

$$
\begin{array}{lll}
\mathcal{H}: \mathbb{R}^{d \times d} \to \mathbb{S}_{++}^d, & \mathcal{H}(X) = XX^\top, & \partial H = (\partial X) X^\top + X (\partial X)^\top \\
\mathcal{P}: \mathbb{S}_{++}^d \to \mathbb{R}^{d \times d}, & \mathcal{P}(X) = BXA, & \partial P = B (\partial X) A \\
\mathcal{S}: \mathbb{S}_{++}^d \to \mathbb{S}_{++}^d, & \mathcal{S}(X) = \sqrt{X}, & \partial S \text{ satisfies:} \quad (\partial S) S + S (\partial S) = \partial X \\
\mathcal{Y}: \mathbb{S}_{++}^d \to \mathbb{R}_+, & \mathcal{Y}(X) = \operatorname{Tr} [X], & \partial y = \operatorname{Tr} [\partial X].
\end{array}
$$

We then have

$$f(L) = \mathcal{Y}\Big( \mathcal{S}\big[ \mathcal{H}(\mathcal{P}[\mathcal{H}(L)]) \big] \Big) \quad \text{or} \quad f = \mathcal{Y} \circ \mathcal{S} \circ \mathcal{H} \circ \mathcal{P} \circ \mathcal{H}.$$

Working backwards, we define the following functions

$$
\begin{aligned}
g_1 &= \mathcal{Y}: & S &\mapsto y \\
g_2 &= \mathcal{Y} \circ \mathcal{S}: & H_2 &\mapsto y \\
g_3 &= \mathcal{Y} \circ \mathcal{S} \circ \mathcal{H}: & P &\mapsto y \\
g_4 &= \mathcal{Y} \circ \mathcal{S} \circ \mathcal{H} \circ \mathcal{P}: & H_1 &\mapsto y \\
g_5 &= \mathcal{Y} \circ \mathcal{S} \circ \mathcal{H} \circ \mathcal{P} \circ \mathcal{H} = f: & L &\mapsto y
\end{aligned}
$$

Applying the chain rule yields

$$
\frac{\partial g_2(H_2)}{\partial [H_2]_{ij}} = \frac{\partial \mathcal{Y}(S)}{\partial [H_2]_{ij}} = \frac{\partial \mathcal{Y}(\mathcal{S}(H_2))}{\partial [H_2]_{ij}} = \mathrm{Tr}\left[ \left( \frac{\partial \mathcal{Y}(S)}{\partial S} \right)^\top \frac{\partial \mathcal{S}(H_2)}{\partial [H_2]_{ij}} \right] = \mathrm{Tr}\left[ \frac{\partial \mathcal{S}(H_2)}{\partial [H_2]_{ij}} \right],
$$

where we used $\frac{\partial \mathcal{Y}(S)}{\partial S} = I$ in the last equality. We then obtain

$$
\frac{\partial g_3(P)}{\partial [P]_{ij}} = \frac{\partial g_2(H_2)}{\partial [P]_{ij}} = \frac{\partial g_2(\mathcal{H}(P))}{\partial [P]_{ij}} = \mathrm{Tr}\left[ \left( \frac{\partial g_2(H_2)}{\partial H_2} \right)^\top \frac{\partial \mathcal{H}(P)}{\partial [P]_{ij}} \right]
$$

$$
\frac{\partial g_4(H_1)}{\partial [H_1]_{ij}} = \frac{\partial g_3(P)}{\partial [H_1]_{ij}} = \frac{\partial g_3(\mathcal{P}(H_1))}{\partial [H_1]_{ij}} = \mathrm{Tr}\left[ \left( \frac{\partial g_3(P)}{\partial P} \right)^\top \frac{\partial \mathcal{P}(H_1)}{\partial [H_1]_{ij}} \right]
$$

$$
\frac{\partial g_5(L)}{\partial [L]_{ij}} = \frac{\partial g_4(H_1)}{\partial [L]_{ij}} = \frac{\partial g_4(\mathcal{H}(L))}{\partial [L]_{ij}} = \mathrm{Tr}\left[ \left( \frac{\partial g_4(H_1)}{\partial H_1} \right)^\top \frac{\partial \mathcal{H}(L)}{\partial [L]_{ij}} \right]
$$

At this point, we are left with the task of computing the elementary derivative matrices (in black). We can do so by recognizing the following identities, where $E^{ij}$ is a matrix such that $E^{ij}(i,j) = 1$ at index $(i,j)$ and zero everywhere else:

$$
\begin{aligned}
\frac{\partial \mathcal{H}(L)}{\partial [L]_{ij}} &= (E^{ij})L^\top + L\left(E^{ij}\right)^\top \\
\frac{\partial \mathcal{P}(H_1)}{\partial [H_1]_{ij}} &= B E^{ij} A \\
\frac{\partial \mathcal{H}(P)}{\partial [P]_{ij}} &= \left(E^{ij}\right)P^\top + P(E^{ij})^\top \\
\left( \frac{\partial \mathcal{S}(H_2)}{\partial [H_2]_{ij}} \right) S + S \left( \frac{\partial \mathcal{S}(H_2)}{\partial [H_2]_{ij}} \right) &= E^{ij}.
\end{aligned}
$$

The matrices on the right-hand side above are available from the forward pass, and the partial derivative of $\mathcal{S}$ with respect to $[H_2]_{ij}$ is found by solving the corresponding Lyapunov equation in the last row. Finally, the gradient matrix is obtained by arranging the entries at their respective indices, as done in the backward step:

$$
\frac{\partial f(L)}{\partial L} = \left[ \frac{\partial g_5(L)}{\partial [L]_{ij}} \right].
$$

The overall gradient of the Gelbrich distance results from elementary calculations.

### D.2.1. NONCONVEXITY OF THE PARAMETRIZED GELBRICH DISTANCE

We provide examples showing the nonconvexity of the parametrized (squared) Gelbrich distance between two mean-covariance pairs, as defined in Definition D.1, with respect to the parameter $L$. Again, we proceed by offering a counterexample.

**Gelbrich Distance.**  Consider the two mean-covariance pairs

$$\mu_{\mathbb{P}} = (1.0, 0.6) \text{ and } \Sigma_{\mathbb{P}} = \begin{bmatrix} 0.1 & 0.0 \\ 0.0 & 1.0 \end{bmatrix}$$

$$\mu_{\mathbb{Q}} = (0.8, 0.6) \text{ and } \Sigma_{\mathbb{Q}} = \begin{bmatrix} 10.0 & 0.0 \\ 0.0 & 1.0 \end{bmatrix},$$

and the two $L$ matrices

$$L_1 = \begin{bmatrix} 0.2 & 0 \\ 0.2 & 1.9 \end{bmatrix} \text{ and } L_2 = \begin{bmatrix} 0.6 & 0 \\ 0.8 & 0.5 \end{bmatrix}.$$

To shorten the notation, let $N_{\mathbb{P}} = (\mu_{\mathbb{P}}, \Sigma_{\mathbb{P}})$ and $N_{\mathbb{Q}} = (\mu_{\mathbb{Q}}, \Sigma_{\mathbb{Q}})$. Applying the formula in Definition D.1 yields

$$g(N_{\mathbb{P}}, N_{\mathbb{Q}}; L_1) = 0.5675,$$
$$g(N_{\mathbb{P}}, N_{\mathbb{Q}}; L_2) = 1.3142,$$
$$g(N_{\mathbb{P}}, N_{\mathbb{Q}}; 0.5 \cdot L_1 + 0.5 \cdot L_2) = 1.0636,$$

indicating that

$$g(N_{\mathbb{P}}, N_{\mathbb{Q}}; 0.5 \cdot L_2 + 0.5 \cdot L_1) > 0.5 \cdot g(N_{\mathbb{P}}, N_{\mathbb{Q}}; L_1) + 0.5 \cdot g(N_{\mathbb{P}}, N_{\mathbb{Q}}; L_2),$$

and proving that the Gelbrich distance is not convex.

**Squared Gelbrich Distance.**  Similarly, consider the two mean-covariance pairs

$$\mu_{\mathbb{P}} = (0.4, 0.6) \text{ and } \Sigma_{\mathbb{P}} = \begin{bmatrix} 0.1 & 0.0 \\ 0.0 & 1.0 \end{bmatrix}$$

$$\mu_{\mathbb{Q}} = (0.4, 0.4) \text{ and } \Sigma_{\mathbb{Q}} = \begin{bmatrix} 10.0 & 0.0 \\ 0.0 & 1.0 \end{bmatrix},$$

and the two $L$ matrices

$$L_1 = \begin{bmatrix} 0.7 & 0 \\ 0.4 & 1.9 \end{bmatrix} \text{ and } L_2 = \begin{bmatrix} 0.9 & 0 \\ 0.9 & 0.6 \end{bmatrix}.$$

Again, let $N_{\mathbb{P}} = (\mu_{\mathbb{P}}, \Sigma_{\mathbb{P}})$ and $N_{\mathbb{Q}} = (\mu_{\mathbb{Q}}, \Sigma_{\mathbb{Q}})$. Proceeding as before, we obtain

$$g(N_{\mathbb{P}}, N_{\mathbb{Q}}; L_1)^2 = 3.9720,$$
$$g(N_{\mathbb{P}}, N_{\mathbb{Q}}; L_2)^2 = 4.4900,$$
$$g(N_{\mathbb{P}}, N_{\mathbb{Q}}; 0.5 \cdot L_1 + 0.5 \cdot L_2)^2 = 4.4865,$$

indicating that

$$g(N_{\mathbb{P}}, N_{\mathbb{Q}}; 0.5 \cdot L_2 + 0.5 \cdot L_1)^2 > 0.5 \cdot g(N_{\mathbb{P}}, N_{\mathbb{Q}}; L_1)^2 + 0.5 \cdot g(N_{\mathbb{P}}, N_{\mathbb{Q}}; L_2)^2,$$

and proving that the squared Gelbrich distance is not convex.

## E. Numerical Experiment Details

This appendix provides further details concerning the numerical experiments presented in Section 6.

The complete pseudocode used in the numerical experiments is provided in Algorithm 2. Compared to Algorithm 1 reported in Section 5, Algorithm 2 enhances numerical stability by clipping the values of $J_\phi(\theta_i)$ between a lower bound $\underline{\nabla}_J$ and an upper bound $\overline{\nabla}_J$; similarly, it clips the eigenvalues of $\theta'_{i+1}(\theta'_{i+1})^\top$ between $\underline{\lambda}_M$ and $\overline{\lambda}_M$.

Table 1 reports the values of the hyperparameters used for all numerical procedures.

---

**Algorithm 2** Loss-aware Distributionally Robust Optimization (complete)

---

**Input:** initial guess $\theta_0 = I_d$, samples $\hat{\mathcal{D}}_J = \{\hat{\xi}_j\}_{j=1}^J$, step sizes $\{\alpha_i\}_{i \in \mathbb{N}} > 0$
**Output:** $\theta^\star$, $\hat{w}_{\theta^\star}$
Bootstrap $n_\mathrm{b}$ reference distributions $\{\hat{\mathbb{P}}_k\}_{k=1}^{n_\mathrm{b}}$ by resampling $\hat{\mathcal{D}}_J$ with replacement
Find $\varepsilon$ as the $1 - \beta$ quantile of $\{d(\hat{\mathbb{P}}_k, \hat{\mathbb{P}}; \theta_0)\}_{k=1}^{n_\mathrm{b}}$
**for** $i = 1$ **to** `maxiter` **do**
  Solve conic program with $A(\theta_i), b(\theta_i), c(\theta_i)$ for $x^\star(\theta_i)$
  Differentiate through conic program for $J_{x^\star}(\theta_i)^\top c(\theta_i)$
  Calculate $d(\hat{\mathbb{P}}_k, \hat{\mathbb{P}}; \theta_i)$ for all $k \in [n_\mathrm{b}]$ and obtain $J_e(\theta_i) \in \mathcal{J}_e(\theta_i)$
  Compute $J_\varphi(\theta_i) = J_{x^\star}(\theta_i)^\top c(\theta_i) + x^\star(\theta_i)^\top J_c(\theta_i) + 2\lambda_\mathrm{p} \max\{0, e(\theta_i)\} J_e(\theta_i)$
  Clip the values of $J_\varphi(\theta_i)$ between $\left[\underline{\nabla}_J, \overline{\nabla}_J\right]$
  Update $\theta'_{i+1} = \theta_i - \bar{\alpha} \nabla J_\varphi(\theta_i)_\mathrm{clip}$
  Clip the eigenvalues of $M = \theta'_{i+1}(\theta'_{i+1})^\top$ between $\left[\underline{\lambda}_M, \overline{\lambda}_M\right]$
  Find $\theta_{i+1}$ as the lower triangular Cholesky factorization of $M$
  Update the conic program parameters $A(\theta_{i+1}), b(\theta_{i+1}), c(\theta_{i+1})$
  **if** $(\varphi(\theta_i) - \varphi(\theta_{i+1})) / \varphi(\theta_i) < $ `tol` **then**
    **break**
  **end if**
**end for**

---

| Parameter | Value |
|-----------|-------|
| $\alpha_i \equiv \bar{\alpha}$ | $1 \times 10^{-4}$ |
| `tol` | $1 \times 10^{-6}$ |
| $\lambda_\mathrm{p}$ | 10 |
| $\eta_\mathrm{p}$ | 100.0 |
| `maxiter` | $1 \times 10^6$ |
| $\overline{\nabla}_J$ | 1000 |
| $\underline{\nabla}_J$ | $-1000$ |
| $\overline{\lambda}_M$ | $1 \times 10^6$ |
| $\underline{\lambda}_M$ | $1 \times 10^{-6}$ |

*Table 1.* List of Hyperparameters used for all numerical examples.

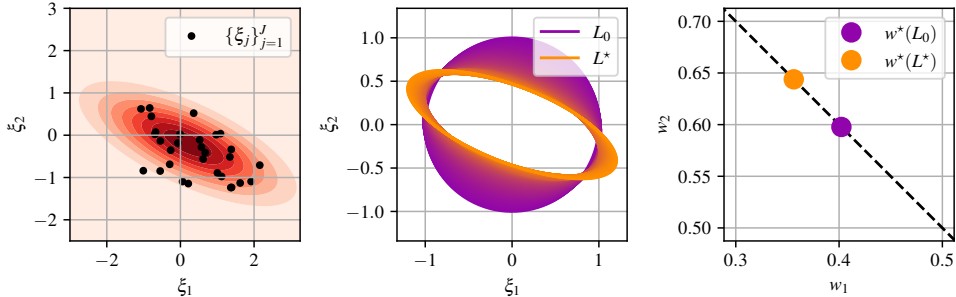

*Figure 6.* Underlying true distribution $\mathbb{P}$ as a red contour and samples in black (left), unit-cost ellipses defined by the matrices $L_i$ in the upper-level optimization (center), and optimal decisions before ($w^\star(L_0)$) and after ($w^\star(L^\star)$) optimization (right).

### E.1. Portfolio Optimization Example

In addition to the visualizations of the single experiment example in Section 6.1, Figure 6 shows the underlying true distribution $\mathbb{P}$ and the samples drawn from it (left panel). Further, the corresponding change in portfolio allocations is shown in the right panel. The center panel again visualizes how the parametrization of the ambiguity set changes over the iterations.

The bootstrapping procedure of Algorithm 2 for this portfolio optimization example consists of resampling $\hat{\mathcal{D}}_J$ with replacement and calculating for each bootstrapped dataset of size $J$ a sample mean and covariance to estimate the Gaussian reference distributions $\{\hat{\mathbb{P}}_k\}_{k=1}^{n_\mathrm{b}}$.

Moreover, we state the generation process for the Gaussian distributions $\mathbb{P}$ used in Subsection 6.1, along with the means and covariances obtained for the multi-experiment analysis. Note that for the latter, each Gaussian distribution is resampled 10 times to generate distinct datasets, thereby strengthening the robustness of the results. For $k = 3$, $\overline{\mu} = 1.0$, $\underline{\mu} = -1.0$, $\overline{\sigma} = 0.1$, and $\underline{\sigma} = 0.01$, the Gaussian distributions are generated as

$$\mu \sim \mathcal{U}(\underline{\mu}, \overline{\mu})^k \tag{59}$$

and

$$\tilde{\Sigma} \sim \mathcal{U}(\underline{\sigma}, \overline{\sigma})^{k \times k} \tag{60}$$

$$\Sigma = \tilde{\Sigma}\tilde{\Sigma}^\top + 10^{-6} I_k. \tag{61}$$

For $i = 1, \dots, 50$, the $i$-th Gaussian distribution is defined as $\mathcal{N}(\mu_i, \Sigma_i)$, where $\mu_i$ and $\Sigma_i$ are reported in Table 2.

### E.2. Linear Regression Example

We state the generation process used for the multi-experiments in Subsection 6.2: $x$ in (20) is uniformly sampled in the interval $[-10, 10]$, $w$ is uniformly sampled the interval $[-10, 10]$ and $\sigma^2$ is uniformly sampled in the interval $[500, 1000]$. This procedure leads to the 10 underlying true models stated in Table 3.

Figure 7 presents the average worst-case and out-of-sample errors across all experiments in Subsection 6.2. Notably, both error measures decrease after bilevel optimization (right) compared to before (left).

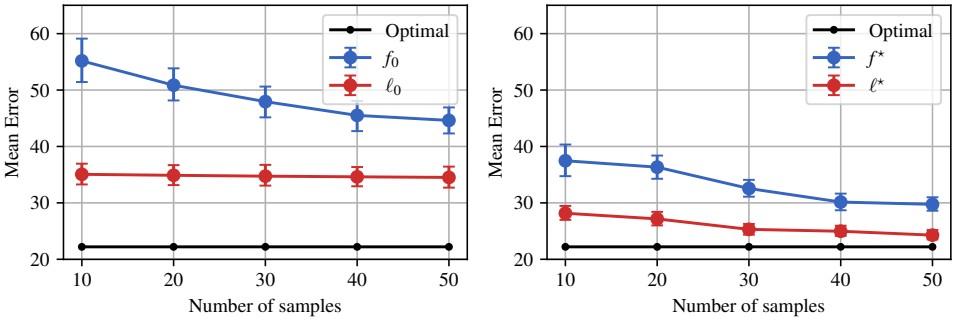

*Figure 7.* Average initial (left) and final (right) objectives (worst-case objective in blue and out-of-sample performance in red) of the distributionally robust linear regression model with absolute error over multiple experiments. The black line shows the average best possible expected error over all experiments.

## F. Further Experimental Results

This appendix provides additional results for different settings of the portfolio optimization and linear regression tasks.

### F.1. Portfolio Optimization with Gaussian Reference Distribution (higher-dimensional case)

We investigate the distributionally robust portfolio optimization task with Gaussian reference distribution from Subsection 6.1, but now consider a higher number of assets, i.e., $k = 10$. The means and covariances are generated using the same distributions as shown in Appendix E.1. We generate 10 independent experiments, each using a different true Gaussian distribution $\mathbb{P}$. For each distribution, 10 distinct datasets are sampled, resulting in 100 different trials for each sample size

*Table 2.* List of Gaussian distributions (means and covariance matrices) used in the experiments.

| Dist. | Mean $\mu_i$ | Covariance Matrix $\Sigma_i$ | | | Dist. | Mean $\mu_i$ | Covariance Matrix $\Sigma_i$ | | |
|---|---|---|---|---|---|---|---|---|---|
| 1 | $\begin{pmatrix}-0.99\\0.74\\-0.51\end{pmatrix}$ | 0.010 | 0.014 | 0.010
0.014 0.020 0.015
0.010 0.015 0.012 | 26 | $\begin{pmatrix}0.00\\0.08\\-0.73\end{pmatrix}$ | 0.003 | 0.006 | 0.005
0.006 0.017 0.011
0.005 0.011 0.016 |
| 2 | $\begin{pmatrix}-0.98\\0.00\\-0.01\end{pmatrix}$ | 0.002 0.003 0.005
0.003 0.004 0.007
0.005 0.007 0.011 | | | 27 | $\begin{pmatrix}0.10\\0.43\\0.21\end{pmatrix}$ | 0.016 0.011 0.009
0.011 0.009 0.006
0.009 0.006 0.005 | | |
| 3 | $\begin{pmatrix}-0.90\\0.01\\0.04\end{pmatrix}$ | 0.006 0.006 0.007
0.006 0.006 0.007
0.007 0.007 0.009 | | | 28 | $\begin{pmatrix}0.10\\0.42\\-0.42\end{pmatrix}$ | 0.017 0.006 0.010
0.006 0.002 0.003
0.010 0.003 0.012 | | |
| 4 | $\begin{pmatrix}-0.85\\0.56\\-0.12\end{pmatrix}$ | 0.020 0.005 0.009
0.005 0.001 0.002
0.009 0.002 0.005 | | | 29 | $\begin{pmatrix}0.13\\-0.14\\-0.81\end{pmatrix}$ | 0.002 0.005 0.005
0.005 0.020 0.016
0.005 0.016 0.015 | | |
| 5 | $\begin{pmatrix}-0.75\\-0.48\\-0.19\end{pmatrix}$ | 0.012 0.009 0.010
0.009 0.011 0.007
0.010 0.007 0.010 | | | 30 | $\begin{pmatrix}0.15\\0.06\\0.53\end{pmatrix}$ | 0.002 0.003 0.004
0.003 0.004 0.006
0.004 0.006 0.012 | | |
| 6 | $\begin{pmatrix}-0.74\\-0.00\\0.20\end{pmatrix}$ | 0.016 0.013 0.008
0.013 0.011 0.006
0.008 0.006 0.007 | | | 31 | $\begin{pmatrix}0.30\\-0.91\\-0.96\end{pmatrix}$ | 0.018 0.008 0.013
0.008 0.004 0.006
0.013 0.006 0.010 | | |
| 7 | $\begin{pmatrix}-0.68\\0.14\\-0.25\end{pmatrix}$ | 0.005 0.004 0.007
0.004 0.007 0.009
0.007 0.009 0.013 | | | 32 | $\begin{pmatrix}0.39\\0.63\\-0.31\end{pmatrix}$ | 0.008 0.004 0.008
0.004 0.011 0.012
0.008 0.012 0.015 | | |
| 8 | $\begin{pmatrix}-0.68\\-1.00\\-0.57\end{pmatrix}$ | 0.019 0.008 0.016
0.008 0.004 0.007
0.016 0.007 0.015 | | | 33 | $\begin{pmatrix}0.39\\0.28\\-0.74\end{pmatrix}$ | 0.016 0.017 0.011
0.017 0.022 0.013
0.011 0.013 0.008 | | |
| 9 | $\begin{pmatrix}-0.64\\-0.20\\0.79\end{pmatrix}$ | 0.004 0.005 0.006
0.005 0.008 0.009
0.006 0.009 0.010 | | | 34 | $\begin{pmatrix}0.40\\-0.37\\-0.76\end{pmatrix}$ | 0.007 0.007 0.004
0.007 0.010 0.007
0.004 0.007 0.004 | | |
| 10 | $\begin{pmatrix}-0.56\\0.74\\-0.59\end{pmatrix}$ | 0.002 0.002 0.004
0.002 0.004 0.004
0.004 0.004 0.010 | | | 35 | $\begin{pmatrix}0.41\\0.32\\-0.86\end{pmatrix}$ | 0.014 0.016 0.017
0.016 0.019 0.021
0.017 0.021 0.023 | | |
| 11 | $\begin{pmatrix}-0.53\\-0.14\\-0.82\end{pmatrix}$ | 0.004 0.008 0.008
0.008 0.015 0.016
0.008 0.016 0.018 | | | 36 | $\begin{pmatrix}0.46\\0.39\\0.88\end{pmatrix}$ | 0.010 0.007 0.011
0.007 0.018 0.014
0.011 0.014 0.016 | | |
| 12 | $\begin{pmatrix}-0.50\\0.89\\-0.62\end{pmatrix}$ | 0.009 0.010 0.007
0.010 0.010 0.007
0.007 0.007 0.009 | | | 37 | $\begin{pmatrix}0.48\\0.51\\-0.07\end{pmatrix}$ | 0.010 0.011 0.007
0.011 0.012 0.006
0.007 0.006 0.004 | | |
| 13 | $\begin{pmatrix}-0.44\\-0.08\\-0.76\end{pmatrix}$ | 0.003 0.005 0.005
0.005 0.014 0.012
0.005 0.012 0.011 | | | 38 | $\begin{pmatrix}0.55\\-0.12\\0.72\end{pmatrix}$ | 0.009 0.009 0.008
0.009 0.011 0.005
0.008 0.005 0.013 | | |
| 14 | $\begin{pmatrix}-0.34\\-0.19\\0.15\end{pmatrix}$ | 0.010 0.011 0.012
0.011 0.013 0.014
0.012 0.014 0.018 | | | 39 | $\begin{pmatrix}0.56\\0.21\\0.42\end{pmatrix}$ | 0.014 0.012 0.006
0.012 0.013 0.007
0.006 0.007 0.004 | | |
| 15 | $\begin{pmatrix}-0.32\\-0.08\\0.87\end{pmatrix}$ | 0.016 0.015 0.012
0.015 0.014 0.011
0.012 0.011 0.010 | | | 40 | $\begin{pmatrix}0.66\\-0.28\\0.41\end{pmatrix}$ | 0.012 0.010 0.013
0.010 0.010 0.011
0.013 0.011 0.016 | | |
| 16 | $\begin{pmatrix}-0.27\\0.19\\-0.22\end{pmatrix}$ | 0.005 0.005 0.008
0.005 0.010 0.015
0.008 0.015 0.023 | | | 41 | $\begin{pmatrix}0.69\\-0.68\\0.12\end{pmatrix}$ | 0.014 0.010 0.004
0.010 0.009 0.006
0.004 0.006 0.007 | | |
| 17 | $\begin{pmatrix}-0.27\\-0.60\\-0.82\end{pmatrix}$ | 0.006 0.010 0.006
0.010 0.016 0.011
0.006 0.011 0.013 | | | 42 | $\begin{pmatrix}0.70\\0.77\\0.53\end{pmatrix}$ | 0.016 0.012 0.016
0.012 0.009 0.012
0.016 0.012 0.018 | | |
| 18 | $\begin{pmatrix}-0.22\\0.19\\0.03\end{pmatrix}$ | 0.005 0.007 0.004
0.007 0.012 0.008
0.004 0.008 0.005 | | | 43 | $\begin{pmatrix}0.73\\0.71\\0.62\end{pmatrix}$ | 0.015 0.009 0.014
0.009 0.013 0.009
0.014 0.009 0.013 | | |
| 19 | $\begin{pmatrix}-0.20\\0.43\\-0.44\end{pmatrix}$ | 0.013 0.010 0.009
0.010 0.011 0.008
0.009 0.008 0.008 | | | 44 | $\begin{pmatrix}0.75\\0.94\\0.74\end{pmatrix}$ | 0.012 0.010 0.013
0.010 0.009 0.011
0.013 0.011 0.020 | | |
| 20 | $\begin{pmatrix}-0.17\\0.44\\-1.00\end{pmatrix}$ | 0.007 0.006 0.009
0.006 0.011 0.015
0.009 0.015 0.020 | | | 45 | $\begin{pmatrix}0.79\\-0.34\\0.64\end{pmatrix}$ | 0.018 0.010 0.013
0.010 0.007 0.009
0.013 0.009 0.016 | | |
| 21 | $\begin{pmatrix}-0.16\\0.85\\-0.45\end{pmatrix}$ | 0.006 0.007 0.007
0.007 0.015 0.010
0.007 0.010 0.008 | | | 46 | $\begin{pmatrix}0.81\\-0.86\\0.35\end{pmatrix}$ | 0.016 0.014 0.016
0.014 0.013 0.014
0.016 0.014 0.017 | | |
| 22 | $\begin{pmatrix}-0.13\\-0.95\\0.10\end{pmatrix}$ | 0.014 0.012 0.010
0.012 0.012 0.011
0.010 0.011 0.011 | | | 47 | $\begin{pmatrix}0.81\\-0.85\\-0.45\end{pmatrix}$ | 0.014 0.013 0.014
0.013 0.014 0.015
0.014 0.015 0.017 | | |
| 23 | $\begin{pmatrix}-0.11\\0.14\\0.82\end{pmatrix}$ | 0.017 0.012 0.009
0.012 0.015 0.011
0.009 0.011 0.008 | | | 48 | $\begin{pmatrix}0.91\\0.54\\-0.75\end{pmatrix}$ | 0.011 0.011 0.011
0.011 0.012 0.012
0.011 0.012 0.013 | | |
| 24 | $\begin{pmatrix}-0.03\\-0.50\\0.44\end{pmatrix}$ | 0.013 0.009 0.008
0.009 0.007 0.006
0.008 0.006 0.007 | | | 49 | $\begin{pmatrix}0.91\\-0.58\\0.66\end{pmatrix}$ | 0.018 0.015 0.005
0.015 0.014 0.004
0.005 0.004 0.002 | | |
| 25 | $\begin{pmatrix}-0.02\\-0.53\\-0.86\end{pmatrix}$ | 0.011 0.011 0.012
0.011 0.014 0.012
0.012 0.012 0.015 | | | 50 | $\begin{pmatrix}0.93\\0.09\\0.95\end{pmatrix}$ | 0.009 0.013 0.008
0.013 0.019 0.014
0.008 0.014 0.013 | | |

| Model | Weight ($w$) | Variance ($\sigma^2$) |
|-------|--------------|-----------------------|
| 1 | -6.7805 | 564.285 |
| 2 | -5.8464 | 625.412 |
| 3 | -2.7811 | 699.653 |
| 4 | -1.3851 | 710.190 |
| 5 | -0.0144 | 783.458 |
| 6 | 4.3483 | 846.372 |
| 7 | 6.3163 | 915.492 |
| 8 | 7.1061 | 922.537 |
| 9 | 8.5174 | 932.399 |
| 10 | 8.9350 | 978.001 |

*Table 3.* Summary of the 10 models used for linear regression. For each model, $x \sim \mathcal{U}(-10, 10)$ and $y = wx + e$ with $e \sim \mathcal{N}(0, \sigma^2)$.

$J \in \{10, \ldots, 100\}$. To evaluate the effectiveness of our procedure, we monitor the relative improvement of the worst-case objective and of the out-of-sample performance, as defined in Subsection 6.1.

Figure 8 illustrates the average improvement in both the worst-case objective and out-of-sample performance of the portfolio from before (left) to after (right) bilevel optimization. The displayed error bars represent bootstrapped confidence intervals, obtained by resampling the results to estimate the mean improvements. The results of this preliminary investigation suggest that the advantages of our approach may scale favorably with dimension, as the relative improvements are larger than in the lower-dimensional setting.

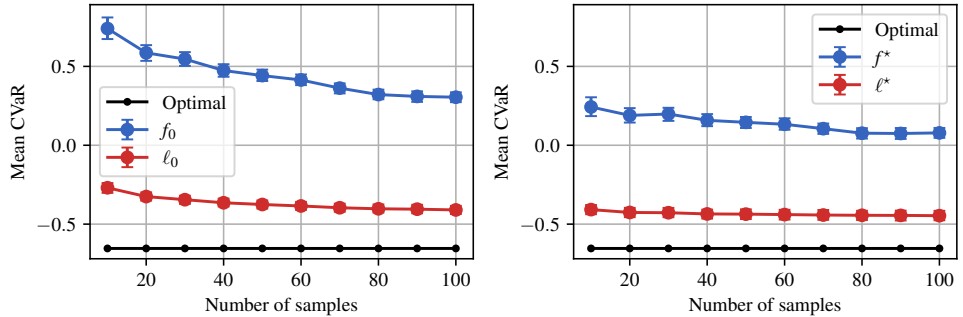

*Figure 8.* Average results for the higher-dimensional portfolio optimization example using a Gaussian reference distribution (worst-case objective in blue and out-of-sample performance in red) when considering multiple experiments with underlying true Gaussian distributions, before (left) and after (right) the bilevel optimization.

## F.2. Portfolio Optimization with Empirical Reference Distribution

We investigate the distributionally robust portfolio optimization task when employing the empirical distribution $\hat{\mathbb{P}} = \frac{1}{J} \sum_{j=1}^{J} \delta_{\hat{\xi}_j}$ as the center of the optimal transport ambiguity set. To assess the results, we again consider the same metrics described in Subsection 6.1.

### F.2.1. DISCRETE TRUE DISTRIBUTION

We generate 10 true discrete distributions defined as

$$\mathbb{P} = \sum_{i=1}^{10} p_i \delta_{x_i} \tag{62}$$

where the 10 different support points $x_i \in \mathbb{R}^k$ with $k = 3$ are uniformly sampled from the cube $[-1, 1]^3$, and the weights $p_i$ are drawn from a Dirichlet distribution. For each true distribution, we generate 10 independent datasets via resampling. This entire process is repeated for $J \in \{10, ..., 100\}$, keeping the underlying true distributions fixed. We use the type-1 Mahalanobis distance to parametrize the transportation cost.

Figure 9 illustrates the average improvement in both the worst-case objective and out-of-sample performance of the portfolio

from before (left) to after (right) bilevel optimization. The displayed error bars represent bootstrapped confidence intervals, obtained by resampling the results to estimate the mean improvements.

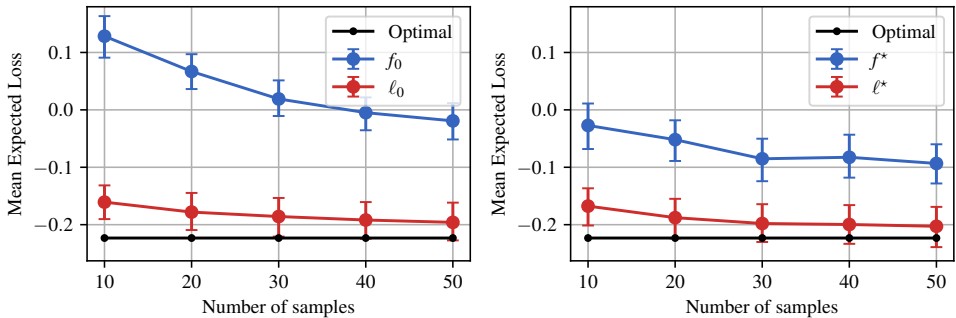

*Figure 9.* Average results (worst-case objective in blue and out-of-sample performance in red) when considering multiple experiments with underlying true discrete distributions, before (left) and after (right) the bilevel optimization.

### F.2.2. GAUSSIAN MIXTURE MODEL

We consider the case where the underlying true distribution is a Gaussian Mixture Model (GMM), defined as

$$\mathbb{P} = \sum_{i=1}^{3} \alpha_i \cdot \mathcal{N}(\mu_i, \Sigma_i), \tag{63}$$

where the means $\mu_i$ are uniformly sampled in the cube $[-1,1]^3$ and the covariances $\Sigma_i$ are generated following (61). The weights $\alpha_i$ are sampled from a Dirichlet distribution. Again, each distribution $\mathbb{P}$ is resampled 10 times to generate independent datasets per experiment. This procedure is repeated for sample sizes $J \in \{10, ..., 100\}$, using the same underlying true distributions.

Figure 10 illustrates the average improvement in both the worst-case objective and out-of-sample performance of the portfolio from before (left) to after (right) bilevel optimization.

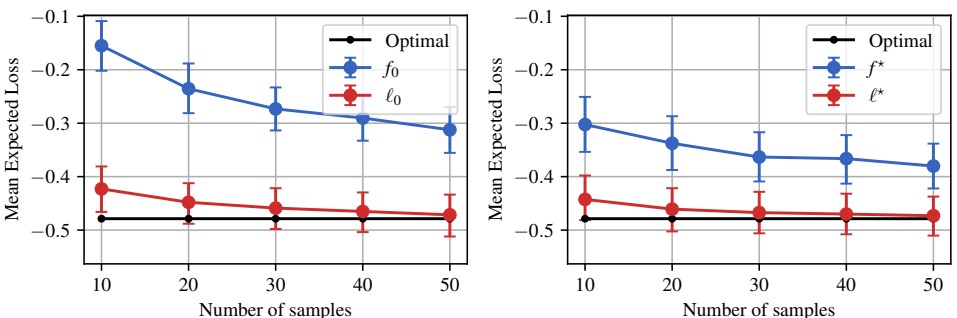

*Figure 10.* Average results (worst-case objective in blue and out-of-sample performance in red) when considering multiple experiments with underlying true GMM distributions, before (left) and after (right) the bilevel optimization.

### F.3. Linear Regression with Squared Error

We consider a linear regression task with squared loss function $\ell_2(w, \xi) = \left((-w, 1)^\top \xi\right)^2$, and we use the type-2 Mahalanobis distance as a parametrization for the transportation cost. As in the $\ell_1$ error case reported in Subsection 6.2, we first show the results of a single problem instance. We consider a linear model corrupted by zero-mean Gaussian noise

$$y = wx + \mathrm{e}, \text{ where } \mathrm{e} \sim \mathcal{N}(0, \sigma^2) \text{ and } x \sim \mathcal{U}(-10.0, 10.0). \tag{64}$$

The weight $w$ is deterministic and set to 1, and the standard deviation of the noise is $\sigma = 10$.

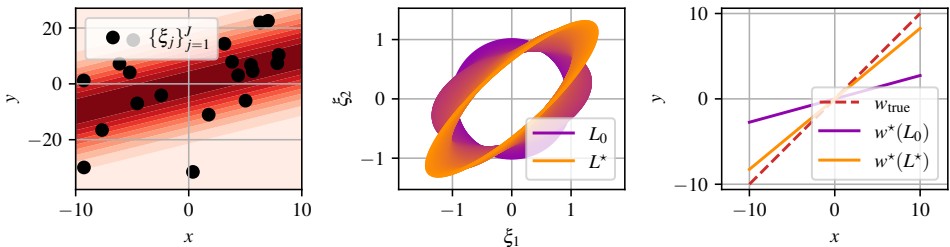

*Figure 11.* Underlying true distribution $\mathbb{P}$ (red contour) and samples (black), unit-cost ellipses defined by the matrices $L_i$ in the upper-level optimization, and optimal decisions before ($w^\star(L_0)$) and after ($w^\star(L^\star)$) optimization when using the squared error $\ell_2$. We set $n_b = 10$.

Figure 11 shows the true distribution of $\xi$ and the $J = 20$ samples from it (left), the unit transportation cost ellipses across iterations (center), and the effect of our procedure on the linear model (right).

Figure 12 (left) shows the expected squared error with respect to the worst-case distribution $\mathbb{Q}^\star(L)$, i.e., $e_{wc}(L) = \mathbb{E}_{\xi \sim \mathbb{Q}^\star(L)} (\ell_2(w^\star(L), \xi))$ which corresponds to the upper level objective, and the expected squared error on the true data generating process $\mathbb{P}$ approximated with $10^7$ samples, i.e., $e_{oos}(L) = \mathbb{E}_{\xi \sim \mathbb{P}} (\ell_2(w^\star(L), \xi))$. Both metrics decrease across iterations, indicating a reduction in conservatism of the decision. Additionally, we carry out a multi-experiment analysis by adopting the same models described in Appendix E.2. To evaluate the results, we again monitor the evolution of the relative improvement of the worst-case objective

$$f_0 = \mathbb{E}_{\xi \sim \mathbb{Q}^\star(L_0)} \left[ \left( \bar{w}^\star(L_0)^\top \xi \right)^2 \right] \text{ and } f^\star = \mathbb{E}_{\xi \sim \mathbb{Q}^\star(L^\star)} \left[ \left( \bar{w}^\star(L^\star)^\top \xi \right)^2 \right]$$

and of the out-of-sample performance

$$\ell_0 = \frac{1}{n_{oos}} \sum_{i=1}^{N_{oos}} \left( \bar{w}^\star(L_0)^\top \xi_i \right)^2 \text{ and } \ell^\star = \frac{1}{n_{oos}} \sum_{i=1}^{N_{oos}} \left( \bar{w}^\star(L^\star)^\top \xi_i \right)^2,$$

with $n_{oos} = 10^6$, across iterations. Figure 12 (right) and Figure 13 jointly indicate a reduction in the conservatism of the DRO solution induced by the proposed bilevel procedure, corroborating the results discussed in Subsection 6.2.

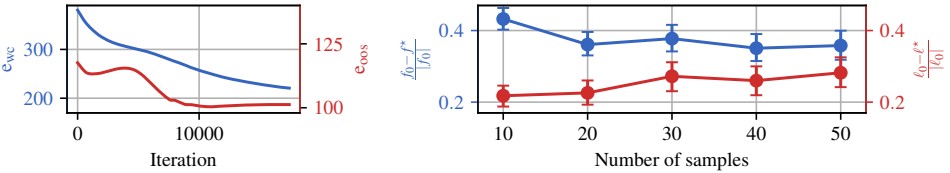

*Figure 12.* Expected squared error on the worst-case distribution $e_{wc}$ and expected squared error on the true data generating process $e_{oos}$ (left). Average improvement over multiple experiments w.r.t. the number of samples (right).

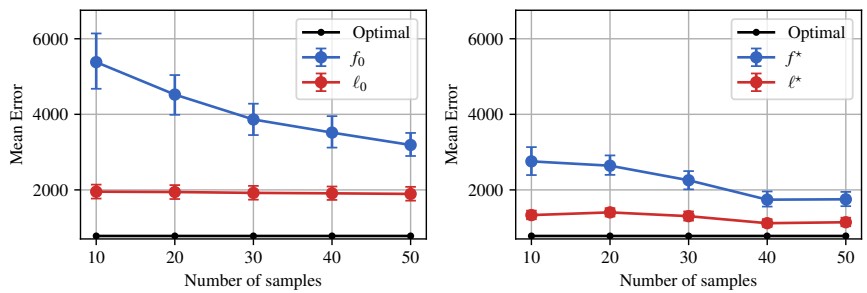

*Figure 13.* Average initial (left) and final (right) objectives (worst-case objective in blue and out-of-sample performance in red) of the distributionally robust linear regression model with squared error over multiple experiments. The black line shows the average best possible expected error over all experiments.

# G. Pedagogical Example: When Geometry Learning Helps

This appendix illustrates a regime in which the learned transport geometry is most useful. The example is designed so that some covariates have high variability but no direct relevance for the downstream prediction task. An isotropic transport cost treats these nuisance directions as equally important, whereas the learned geometry can assign them a lower transport cost and thereby reduce conservatism.

We provide a synthetic linear-regression experiment to illustrate this point. The experiment isolates the effect of covariates that have high variability but no direct relevance for the downstream prediction task. In this experiment, the covariates are $x = (x_0, x_1, x_2, x_3) \in \mathbb{R}^4$ and the response is $y \in \mathbb{R}$, with

$$y = x_0 + e, \qquad x_0 \sim \mathcal{N}(0, 1), \qquad e \sim \mathcal{N}(0, 1), \tag{65}$$

and nuisance coordinates

$$x_1, x_2, x_3 \overset{\text{ind}}{\sim} \mathcal{N}(0, \sigma_{\text{noise}}^2), \qquad \sigma_{\text{noise}} \in \{1, 5, 10, 15\}.$$

Each configuration uses $J = 20$ training samples, 3 independent problem seeds, and 3 datasets per problem seed, giving 9 replicated datasets per configuration. For each dataset, the radius $\epsilon$ is calibrated at the isotropic geometry $L_0 = I$ from $n_b = 10$ bootstrap resamples using the $(1 - \beta)$-quantile with $\beta = 0.10$. We then solve the isotropic OT-DRO problem, learn $L^\star$ with the bilevel procedure and the same initially calibrated radius, and report means over the 9 replicated datasets. All remaining solver and bilevel hyperparameters are the same as in the other experiments, using the type-2 Mahalanobis transportation cost and $10^6$ upper-level iterations.

For a fitted vector $w^\star(L)$, the out-of-sample loss is computed analytically under the true Gaussian data-generating law as

$$\ell(L) = \mathbb{E}\big[(w^\star(L)^\top x - y)^2\big] = \big(w_0^\star(L) - 1\big)^2 + \sigma_{\text{noise}}^2 \sum_{r=1}^{3} \big(w_r^\star(L)\big)^2 + 1. \tag{66}$$

We report the relative out-of-sample improvement

$$\Delta_{\text{oos}} = 100 \cdot \frac{\ell_0 - \ell^\star}{|\ell_0|}, \tag{67}$$

where $\ell_0 = \ell(L_0)$ and $\ell^\star = \ell(L^\star)$.

To interpret the learned geometry, let $e_0, e_1, e_2, e_3$ denote the coordinate directions corresponding to the four features; the response coordinate is included in the joint transport matrix but is not used in the feature-ratio diagnostics. The table reports the nuisance-to-informative coordinate-scale ratio

$$\rho_L = \frac{1}{3} \sum_{r=1}^{3} \frac{\|(L^\star)^\top e_r\|_2}{\|(L^\star)^\top e_0\|_2}. \tag{68}$$

For the isotropic baseline, this ratio equals one by construction. Values below one mean that a unit displacement in a nuisance direction has a smaller learned $L$-norm than a unit displacement in the informative direction, and hence incurs lower squared transport cost under the learned geometry.

Finally, we report the nuisance-to-informative coefficient ratio

$$\rho_w(L) = \frac{\|(w_1^\star(L), w_2^\star(L), w_3^\star(L))\|_2}{|w_0^\star(L)|}. \tag{69}$$

Lower values indicate that the fitted regressor relies less on nuisance coordinates relative to the informative coordinate.

The results support the intuition that the learned geometry is most useful when the isotropic OT cost allocates robustness budget to directions that are weakly relevant for the downstream decision. As the nuisance variance increases, the mean out-of-sample improvement over isotropic OT-DRO increases from $6.61\%$ at $\sigma_{\text{noise}} = 1$ to $22.01\%$ at $\sigma_{\text{noise}} = 15$. At the same time, the learned nuisance-to-informative coordinate-scale ratio decreases from $0.313$ to $0.126$, indicating that the learned geometry makes transport along nuisance directions relatively cheaper. This avoids treating high-variance nuisance directions as equally decision-relevant as the informative coordinate.

*Table 4.* Summary of the nuisance-feature linear regression experiment. The learned geometry becomes increasingly beneficial when nuisance directions have larger variability. The values are means over 9 replicated datasets per setting. For the isotropic baseline $L_0$, the out-of-sample improvement over itself is zero, and the coordinate-scale ratio between nuisance and informative directions equals one by construction.

| Quantity | Setting | Isotropic $L_0$ | Learned $L^\star$ |
|----------|---------|-----------------|-------------------|
| $\Delta_{\mathrm{oos}}$ | $\sigma_{\mathrm{noise}} = 1$ | 0.00% | 6.61% |
| $\Delta_{\mathrm{oos}}$ | $\sigma_{\mathrm{noise}} = 5$ | 0.00% | 18.88% |
| $\Delta_{\mathrm{oos}}$ | $\sigma_{\mathrm{noise}} = 10$ | 0.00% | 21.13% |
| $\Delta_{\mathrm{oos}}$ | $\sigma_{\mathrm{noise}} = 15$ | 0.00% | 22.01% |
| $\rho_L$ | $\sigma_{\mathrm{noise}} = 1$ | 1.000 | 0.313 |
| $\rho_L$ | $\sigma_{\mathrm{noise}} = 5$ | 1.000 | 0.129 |
| $\rho_L$ | $\sigma_{\mathrm{noise}} = 10$ | 1.000 | 0.125 |
| $\rho_L$ | $\sigma_{\mathrm{noise}} = 15$ | 1.000 | 0.126 |
| $\rho_w$ | $\sigma_{\mathrm{noise}} = 1$ | 0.492 | 0.596 |
| $\rho_w$ | $\sigma_{\mathrm{noise}} = 5$ | 0.913 | 0.253 |
| $\rho_w$ | $\sigma_{\mathrm{noise}} = 10$ | 0.686 | 0.112 |
| $\rho_w$ | $\sigma_{\mathrm{noise}} = 15$ | 0.780 | 0.101 |

The same effect is visible in the fitted regressor once the nuisance variance is moderate or large. For example, at $\sigma_{\mathrm{noise}} = 15$, the coefficient ratio decreases from $0.780$ for isotropic OT-DRO to $0.101$ for the learned geometry. At $\sigma_{\mathrm{noise}} = 1$, the nuisance coordinates have the same marginal scale as the informative coordinate, so the learned model improves out-of-sample loss mainly by estimating the informative coefficient $w_0$ closer to its true value $1$, even though the nuisance-to-informative coefficient ratio does not yet decrease. Thus, learning $L$ can stretch the ambiguity set in benign nuisance directions while keeping it tighter along directions that affect the downstream loss, thereby reducing conservatism without changing the initially calibrated radius.

# H. Computational Complexity and Sensitivity Analysis

## H.1. Computational Complexity of Differentiation Procedure

We analyze the complexity of the differentiation procedure described in Appendix D.

In general, computing the hypergradient requires the differentiation of two terms: i) the primal-dual solution map of a conic program, and ii) the penalty term. As we outlined in Appendix C, differentiating the solution of a conic program involves a set of matrix operations and the resolution of a linear system of equations (compare (50) for the forward derivative, and (52) for the adjoint), whose dimension equals $n + 2m$, where $n$ denotes the dimension of the primal variable $x$, and $m$ is the dimension of the dual vector $y$ and the slack variable $s$. Generally, solving the linear system is the most computationally intensive operation, scaling with the cube of the dimension (for example when the system is solved using Gaussian elimination).

The number $J$ of samples may affect the number of constraints in the convex reformulation of the DRO problem, and potentially also the dimension of the primal variable. This means that more samples lead to a linear system of larger dimension and potentially to a greater computational complexity. Specifically, the primal decision variable $x$ scales linearly with the number $J$ of samples for the type-1 Mahalanobis distance in the regression example, and similarly in the type-2 Mahalanobis distance in both the portfolio optimization example and in the regression example. The slack variable $s$ and the dual variable $y$ both scale linearly with $J$ in the regression example with type-2 Mahalanobis distance, where the dependency is increased to $2J$ on the same example if the chosen distance is type-1 Mahalanobis. Moreover, these variables scale with $(d + 2)J$ in the case of type-2 Mahalanobis distance on the portfolio example. In all other examples, the sample size does not affect the dimension of the variables.

Next, differentiating the penalty function requires the differentiation of $n_{\mathrm{b}}$ distances, computed on $n_{\mathrm{b}}$ different samples in each iteration. However, because of the $\max$ term in $\varphi_{\mathrm{p}}$, this is only required whenever the current design $\theta_i$ does not meet the required confidence level and thus $e(\theta_i) > 0$. Differentiating the Mahalanobis distance requires solving a linear program

whose dimension scales linearly with the number of samples (see Appendix D.1). Differentiating the Gelbrich distance can be done by solving a set of $d^2$ Lyapunov equations (compare Appendix D.2), each scaling linearly with dimension $d$ of $\theta$.

## H.2. Empirical wall-clock time

We complement the theoretical complexity discussion with empirical wall-clock measurements. All timings correspond to one upper-level iteration, including the lower-level solve and hypergradient computation, and were obtained on an HP ProBook G10 with an Intel i7-1355U CPU. The cost is incurred only during the offline learning phase for the geometry parameter $\theta$. After $L^\star$ has been learned, solving the downstream robust problem has the same form as standard OT-DRO with a fixed transportation cost.

*Table 5.* Average wall-clock time per upper-level iteration when varying the sample size $J$ with fixed dimension $k = 10$. Times are in milliseconds.

| Task | $J = 10$ | $J = 20$ | $J = 50$ | $J = 100$ |
|---|---|---|---|---|
| Portfolio, type-1 discrete from Section F.2.1 | $4.7 \pm 3.0$ | $6.4 \pm 3.0$ | $15 \pm 7.0$ | $22 \pm 10$ |
| Linear regression, type-2 from Section F.3 | $5.8 \pm 3.4$ | $9.8 \pm 5.2$ | $46 \pm 16$ | $220 \pm 64$ |

*Table 6.* Average wall-clock time per upper-level iteration when varying the dimension $k$ with fixed sample size $J = 50$. Times are in milliseconds.

| Task | $k = 2$ | $k = 5$ | $k = 10$ | $k = 20$ |
|---|---|---|---|---|
| Portfolio, type-1 discrete from Section F.2.1 | $7.8 \pm 4.0$ | $11 \pm 3.0$ | $18 \pm 7.0$ | $30 \pm 10$ |
| Linear regression, type-2 from Section F.3 | $10 \pm 5.0$ | $18 \pm 9.0$ | $51 \pm 19$ | $210 \pm 71$ |

## H.3. Sensitivity Analysis of Penalty Parameters

We examine the influence of the penalty parameters $\lambda_\mathrm{p}$ and $\eta_\mathrm{p}$ from (14) and (15) on both the achieved performance improvement and adherence to the coverage constraint (13). For this analysis, we focus on the portfolio optimization problem described in Section 6.1, setting the dimension to $k = 10$ and utilizing $J = 50$ samples. We use $n_b = 20$ bootstrapped distributions and choose $\beta = 0.1$ and $\gamma = 0.05$. We execute the optimization algorithm multiple times (10 experiments with different underlying distributions, each experiment repeated 10 times) across varying values of $\lambda_\mathrm{p}$ and $\eta_\mathrm{p}$.

As shown in Fig. 14, the mean relative improvement remains consistently high across a broad spectrum of $(\lambda_\mathrm{p}, \eta_\mathrm{p})$ configurations. This consistency suggests that the algorithm's performance is largely insensitive to the exact choice of penalty parameters, underscoring its practical robustness.

Furthermore, Fig. 15 shows the values of the coverage constraint (11b) violation after reformulation across different parameter configurations. We observe that the constraint is consistently satisfied for all tested combinations of $(\lambda_\mathrm{p}, \eta_\mathrm{p})$, with only minor violations occurring. While smaller values of the penalty parameter (e.g., $\lambda_\mathrm{p} = 1$) lead to slightly higher values of the constraint expression, they still remain within acceptable bounds, indicating that the penalization in (14) is sufficient even in those cases.

These findings suggest that the method is robust to a meaningful range of penalty-parameter choices, reducing the need for extensive hyperparameter tuning in the tested settings.

## H.4. Sensitivity to initialization

The upper-level geometry-learning problem is nonconvex in the transport matrix $L$, so a gradient-based bilevel method need not return the same endpoint from every initialization. In the main experiments, we initialize with $L_0 = I$, which corresponds to the standard isotropic Wasserstein geometry and gives an interpretable reference point for learning a task-adapted geometry. To assess how much this choice can matter, we run the bilevel optimization from multiple random lower-triangular initial matrices and compare both the optimization traces and the learned geometry matrices.

For each task, we use 20 random initializations. The strictly lower-triangular entries of each initial matrix were sampled independently from $\mathcal{N}(0, 0.5^2)$, the diagonal entries were sampled independently from $\mathrm{LogNormal}(0, 0.35^2)$, and the

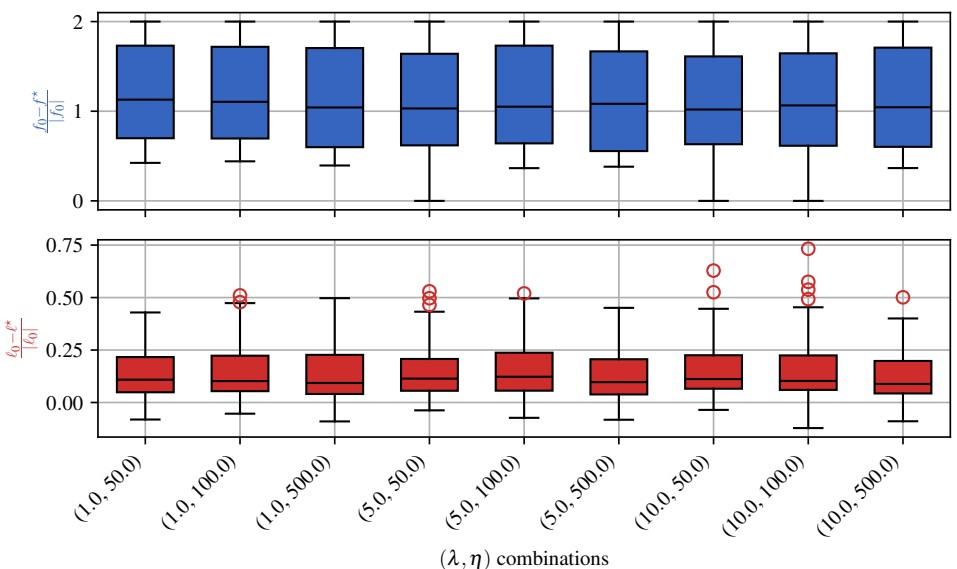

*Figure 14.* Mean relative improvement for different combinations of $(\lambda_{\mathrm{p}}, \eta_{\mathrm{p}})$.

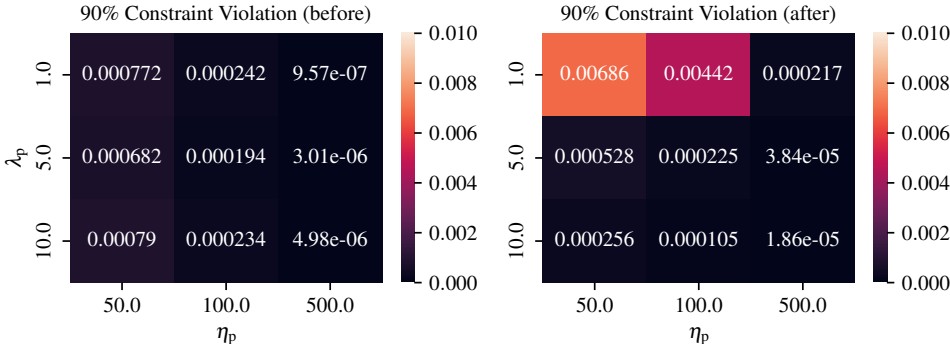

*Figure 15.* Coverage constraint (11b) violations (90th percentile) for different combinations of $(\lambda_{\mathrm{p}}, \eta_{\mathrm{p}})$ over the 100 experiments.

resulting lower-triangular matrix was stabilized and normalized to satisfy $\|L_0\|_{\mathrm{F}} = 1$. The ambiguity radius was held fixed across these runs. Specifically, we calibrated a single radius using the identity geometry, with the $1 - \beta = 0.9$ quantile of $n_{\mathrm{b}} = 10$ bootstrap distances, and then reused this same radius for all random starts. This isolates the effect of the upper-level initialization from the effect of recalibrating the radius for each geometry.

Figure 16 reports the initialization study for the Gaussian/CVaR portfolio example. The portfolio experiment uses $J = 20$ samples, $k = 2$, $\gamma = 0.05$, true mean $(0.5, -0.5)$, true covariance $\left( \begin{smallmatrix} 2.0 & -0.07 \\ -0.07 & 0.5 \end{smallmatrix} \right)$, and a Gaussian reference distribution fitted to the observed sample. Bootstrap distances were computed using the parameterized Gelbrich distance, which gives rise to the fixed radius $\varepsilon = 0.714456$. All 20 runs terminated by the tolerance criterion before the iteration budget was exhausted.

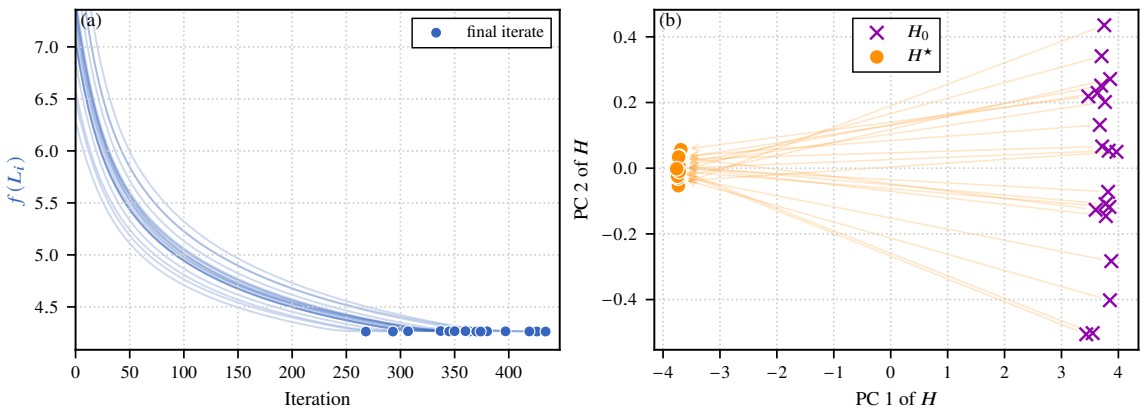

*Figure 16.* Initialization sensitivity for the Gaussian/CVaR portfolio experiment. Left: upper-level objective traces $f(L_i)$ over the bilevel iterations for the 20 random initializations; blue markers indicate the final stored iterate. Right: common PCA projection of the vectorized geometry matrices $H_0 = L_0 L_0^\top$ and $H^\star = L^\star (L^\star)^\top$; arrows match each initialization to its final learned geometry.

The portfolio instance shows little practical sensitivity to the random initialization. Although the initial upper-level objective values differ substantially, the objective traces in Figure 16 collapse to a narrow band. The final penalized upper-level objective has mean 4.2710, standard deviation 0.0015, and range $[4.2691, 4.2747]$. The final out-of-sample CVaR objective is similarly concentrated, with mean 1.6524, standard deviation 0.0013, and range $[1.6493, 1.6541]$. The right panel shows the same behavior geometrically: the random initial matrices occupy a broader region in the PCA projection, whereas the learned matrices form one final cluster. The median pairwise Frobenius distance between final SPD matrices $H^\star = L^\star (L^\star)^\top$ is 0.0465.

Figure 17 reports the same diagnostic for the linear-regression experiment. The data-generating model has $k = 2$ covariates and $J = 20$ observations, with $X_j \sim \mathcal{U}(-10, 10)^2$, $y_j = X_j^\top w_{\mathrm{true}} + \xi_j$, and $\xi_j \sim \mathcal{N}(0, \sigma_\xi^2)$. The problem instance is fixed with $w_{\mathrm{true}} = (-4.6043, -9.1805)$ and $\sigma_\xi^2 = 818.48$. The lower-level problem is the empirical type-2 Wasserstein linear-regression DRO problem. Since transport acts on the joint samples $(x, y)$, the learned lower-triangular matrix has dimension $k + 1 = 3$. The identity-calibrated bootstrap radius is $\varepsilon = 30.7113$. As in the portfolio study, all 20 runs terminated by the tolerance criterion before the iteration budget.

In contrast to the portfolio case, the linear-regression endpoint depends strongly on the initialization. The objective traces in Figure 17 separate into distinct final levels, and the PCA projection shows learned geometries that move to well-separated regions of the geometry space. Sorting the runs by final out-of-sample mean-squared error gives three apparent endpoint groups of sizes 10, 4, and 6, with final out-of-sample ranges $[832.0, 836.2]$, $[1854.2, 1878.8]$, and $[4164.0, 4334.5]$, respectively. Across all pairs of successful runs, the median Frobenius distance between the final SPD matrices is 1240.1, and the median pairwise out-of-sample gap is 1043.0. Thus, for this regression instance, the nonconvex upper-level problem admits different local endpoints, and multiple random initializations are informative.

### H.5. Sensitivity to the coverage level $\beta$

We include a sensitivity study for the coverage level $\beta$ used to calibrate the Wasserstein radius for true distribution coverage. The goal is to check whether the observed reduction in conservatism from the bilevel geometry-learning method depends

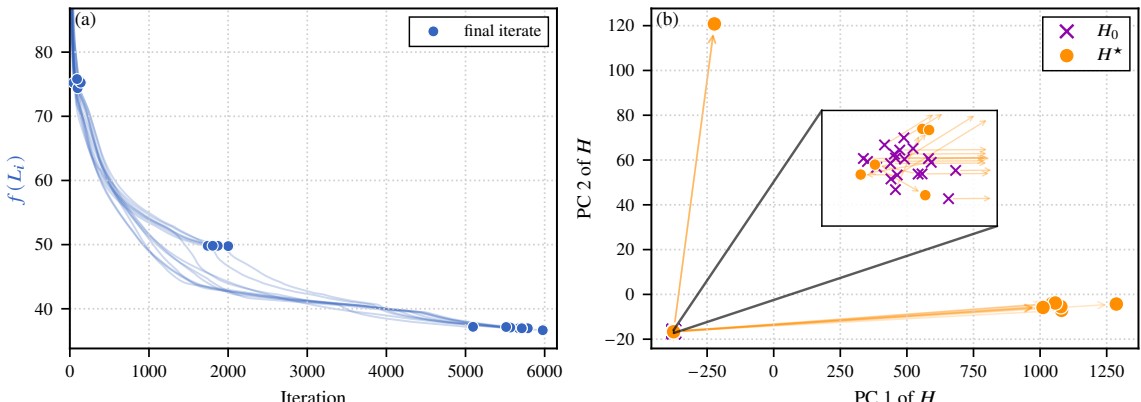

*Figure 17.* Initialization sensitivity for the linear-regression experiment. The left panel shows the upper-level objective traces $f(L_i)$ and their final stored iterates. The right panel uses the same PCA projection of $H_0 = L_0 L_0^\top$ and $H^\star = L^\star (L^\star)^\top$ as in Figure 16; the inset magnifies the initial $H_0$ cloud and nearby final points.

materially on this choice.

The experiment uses the squared-error linear-regression setting with type-2 Mahalanobis transportation cost. Each true problem has $k = 2$ covariates and a scalar response; transport acts on the joint samples $(x, y)$, so $L \in \mathbb{L}^3_{++}$. The data are generated as

$$x \sim \mathcal{U}(-10, 10)^2, \qquad y = x^\top w_{\text{true}} + e, \qquad e \sim \mathcal{N}(0, \sigma^2).$$

We generate 20 true models. For each model, the variance is drawn as $\sigma^2 \sim \mathcal{U}(500, 1000)$, followed by $w_{\text{true}} \sim \mathcal{U}(-10, 10)^2$. For each true model we draw 10 independent observed datasets with $J = 40$ samples, giving 200 problem-dataset pairs. For each pair and each $\beta \in \{0.05, 0.10, 0.15, 0.20\}$, the radius $\varepsilon$ is set to the $(1 - \beta)$-quantile of $n_{\text{b}} = 20$ bootstrap distances computed under the identity geometry $L_0 = I$. We then solve the isotropic OT-DRO problem and run the bilevel method from $L_0$ for max. $10^5$ upper-level iterations.

Figure 18 reports the resulting mean errors over different levels of $\beta$. In the left panel, $f_0$ denotes the square root of the worst-case squared-error objective under $L_0$, and $\ell_0$ denotes the root mean-squared error under the true data-generating model. In the right panel, $f^\star$ and $\ell^\star$ denote the corresponding quantities after learning $L^\star$. Error bars show one standard deviation over the 200 problem-dataset pairs.

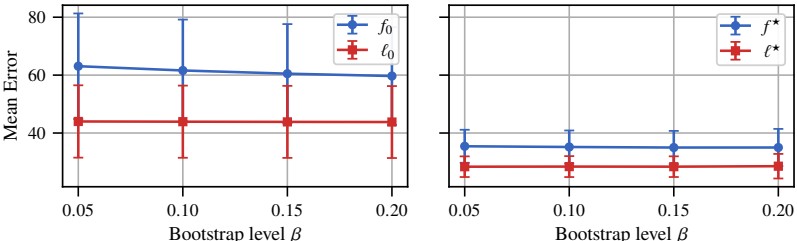

*Figure 18.* Sensitivity to the coverage level $\beta$ in the squared-error linear-regression experiment. Blue curves show the worst-case error $f_0$ and $f^\star$; red curves show the out-of-sample error $\ell_0$ and $\ell^\star$. The bilevel geometry-learning procedure reduces both quantities for all tested values of $\beta$, and the out-of-sample improvement is essentially unchanged across the sweep.

The out-of-sample root error decreases from 43.99 to 28.37 at $\beta = 0.05$, from 43.91 to 28.41 at $\beta = 0.10$, from 43.85 to 28.38 at $\beta = 0.15$, and from 43.79 to 28.54 at $\beta = 0.20$. These correspond to relative improvements of 35.5%, 35.3%, 35.3%, and 34.8%, respectively. Thus, within this range, the choice of $\beta$ has little influence on the out-of-sample improvement delivered by the learned geometry.

