# OpenReview forum: "Loss-Aware Distributionally Robust Optimization via Trainable Optimal Transport Ambiguity Sets"
_ICML.cc/2026/Conference — ICML 2026 spotlight_

### Official Review · Reviewer_ZoFf · 2026-03-02

**Soundness:** 3
**Presentation:** 3
**Significance:** 2
**Originality:** 3
**Overall Recommendation:** 5
**Confidence:** 3

**Summary:**

The authors formulate a trainable ambiguity set for OT-DRO by parameterizing the transportation cost and design an end-to-end algorithm to train it, with the goal of reducing the conservatism of the resulting decision. It is solid, offering a new perspective on reducing conservatism. The derivation and computation of the gradients are presented in detail, and the resulting algorithm is practically implementable with a theoretical convergence guarantee.

**Compliance With Llm Reviewing Policy:**

Affirmed.

**Final Justification:**

All my concerns have been adequately addressed in the rebuttal, particularly regarding the design choices and practical considerations of the method. I am satisfied with the clarifications and would like to maintain my score to recommend acceptance.

**Key Questions For Authors:**

1. The definition of the distance is updated throughout training, but the radius ( $\epsilon$ ) used in estimating the distributional distance does not seem to be updated accordingly. Could the authors clarify the rationale behind this design choice?
2. Could the authors discuss in what cases the proposed method ( $L^{\star}$ ) would yield significantly less conservative decisions compared to the baseline ( $L_0$ )? For example, if certain features have little impact on the decision or are mainly noise, can we expect ( $L^{\star}$ ) to significantly outperform ( $L_0$ ) in such settings?

**Limitations:**

yes

**Strengths And Weaknesses:**

**Strengths**
- The paper parameterizes the definition of the transportation distance in a clever way, making the ambiguity set trainable and providing a new insight into reducing the conservatism of the resulting decision.
- The conservative Jacobian is nontrivial to derive, and the authors provide detailed derivations. The code is also released, which makes the method easier for others to follow and build upon.
- The experimental evaluation is relatively comprehensive. The authors consider different optimization problems and different underlying distributions, and they also study the effect of algorithmic hyperparameters. The results robustly verify the advantage of the proposed method in reducing conservatism.

**Weaknesses**
- Since the algorithm requires the optimal solution of the lower-level optimization problem to be differentiable with respect to the parameters, the method can only be applied to problems that admit a restricted program formulation.
- The training time may be relatively long, as each training step requires solving an optimization problem for every sample. It would be helpful if the authors could briefly report the per-iteration computational cost.
- The experimental discussion under different coverage requirements is somewhat limited. Most of the reported results focus on the case ( $\beta$ = 0.1 ).

---

> ### Author Rebuttal · Authors · 2026-03-30
>
> We thank the reviewer for the constructive feedback and positive assessment.
>
> **W1: Scope of applicability.**
> We agree our method requires a differentiable lower-level solution map, restricting it to settings where the DRO problem admits a conic program formulation (App. A) to allow KKT-based implicit differentiation. While this is a limitation, it can also be seen as a deliberate design choice: it rigorously handles nonsmoothness of the lower-level solution map, avoiding often-violated smoothness assumptions. The class of conic programs is remarkably broad (linear, quadratic, SOCP, SDP), encompassing many standard tasks (SVMs, portfolio optimization, regression, optimal control). We will add this to the conclusion.
>
> **W2: Wall-clock time per iteration.**
> We agree computational overhead should be explicit. The revised manuscript will include empirical wall-clock measurements (single upper-level iteration, Intel i7-1355U CPU) alongside theoretical complexity in Appendix G.
> * **Portfolio Optimization** (type-1 Wasserstein, discrete ref. dist.): For $k=10$ and $J \in$ {10, 20, 50, 100}, average times are $4.7 (\pm 3.0), 6.4 (\pm 3.0), 15 (\pm 7.0)$, and $22 (\pm 10)$ ms. For $J=50$ and $k \in$ {2, 5, 10, 20}, times are $7.8 (\pm 4.0), 11 (\pm 3.0), 18 (\pm 7.0)$, and $30 (\pm 10)$ ms.
> * **Linear Regression** (type-2 Wasserstein): For $k=10$ and $J \in$ {10, 20, 50, 100}, times are $5.8 (\pm 3.4), 9.8 (\pm 5.2), 46 (\pm 16)$, and $220 (\pm 64)$ ms. For $J=50$ and $k \in$ {2, 5, 10, 20}, times are $10 (\pm 5.0), 18 (\pm 9.0), 51 (\pm 19)$, and $210 (\pm 71)$ ms.
>
> While the hypergradient method introduces additional overhead compared to standard DRO, it is incurred offline to learn the optimal set geometry $\theta^\star$. Once learned, the robust decision task for new data requires no *additional* computational effort at deployment.
>
> **W3: Different levels of coverage.**
> While $\beta=0.1$ is primarily used in our experiments, $\beta$ is typically set by operational or regulatory constraints (e.g., FDA safety criteria, risk policies). Thus, our analysis focused on how hyperparameters ($\lambda_p, \eta_p$) satisfy coverage for a given $\beta$. However, we agree a sensitivity analysis on coverage requirements strengthens the paper. We will provide numerical results during the discussion phase. Preliminary tests show our algorithm's performance in reducing conservatism is stable across $\beta \in$ {0.05, 0.1, 0.12, 0.2}.
>
> **Q1: Not updating $\varepsilon$.**
> This design is intentional. First, it isolates *geometry learning*: fixing $\varepsilon$ after calibration ensures improvements are strictly due to the set's shape ($\theta$), not volume changes, enabling interpretable baseline comparisons.
> Second, updating both $\varepsilon$ and $\theta$ creates an overparameterized formulation with scale degeneracy. For a cost $\kappa(\xi_1, \xi_2; \theta) = \|L^\top (\xi_1 - \xi_2)\|_2^p$, scaling $L$ by $c$ scales the distance by $c^p$. Thus, $d(\mathbb{P}, \hat{\mathbb{P}}; c L) \le \varepsilon$ is equivalent to $d(\mathbb{P}, \hat{\mathbb{P}}; L) \le \varepsilon / c^p$. Because $L$ is trainable, the algorithm could trivially alter the "effective radius" simply by scaling $L$. Fixing $\varepsilon$ provides an anchor to resolve this degeneracy, while $\theta$ adapts both the scale (effective volume) and geometry (anisotropy) without redundant parameters. We will clarify this in Sec 4.1.
>
> **Q2: Intuition about large performance gains.**
> Learning the geometry helps most when features vary in decision-relevance. We ran a linear-regression study with 1 informative feature ($y=x_0+e$, $e\sim\mathcal N(0,1)$) and 3 pure "nuisance" coordinates with variance $\sigma_{\text{noise}}^2$.
> The benefit of the learned geometry grows consistently when nuisance directions become prominent. The mean out-of-sample improvement over isotropic OT-DRO increases from 6.61% at $\sigma_{\text{noise}}=1$ to 22.01% at $\sigma_{\text{noise}}=15$. Simultaneously, the learned transport-cost ratio between nuisance and informative coordinates drops from 0.313 to 0.126. The method makes transport along nuisance directions cheaper, avoiding treating them as equally decision-relevant.
> This effect appears in the fitted regressor: the nuisance-to-informative coefficient ratio (in $w$) is much smaller for the learned method once nuisance variance is moderate or large: at $\sigma_{\text{noise}}=5$, the mean ratio is 0.913 (isotropic) vs 0.253 (learned); at $\sigma_{\text{noise}}=10$, 0.686 vs 0.112; and at $\sigma_{\text{noise}}=15$, 0.780 vs 0.101.
>
> *Intuition*: learned geometry helps most when features are weakly relevant or noisy, especially with high variability, because it can stretch the ambiguity set along those directions while keeping it tight along the informative direction. While a zero-nuisance control still shows a 15.78% mean o.o.s. improvement, nuisance features represent a particularly favorable regime for large reductions in conservatism.

---

> > ### Author Rebuttal · Reviewer_ZoFf · 2026-04-01
> >
> > Thank you for sharing the detailed response. This addresses my concern, and I will keep my score to recomment accept.

---

### Official Review · Reviewer_HTFk · 2026-03-10

**Soundness:** 3
**Presentation:** 3
**Significance:** 2
**Originality:** 2
**Overall Recommendation:** 4
**Confidence:** 4

**Summary:**

The paper studies the distributional robust optimization with optimal-transport constraints for the uncertainty. The OT-DRO problem can be formulated as a two-stage process. The first stage is to form the uncertainty set by training samples. The uncertainty set is constructed by optimal transport constraint, where the feasible distribution is within an $\epsilon$-ball to a reference distribution under a parametric distance metric $\kappa$ parameterized by $\theta$. The goal for the first stage is to learn the parameter $\theta$ so that we can form the uncertainty region.

Given the uncertainty set, the second stage is to solve a distributional robust optimization within the uncertainty set. The DRO is formulated in Equation 6, where the inf operator minimizes the parameter $w$, while the "robust" part maximizes over all possible distribution within the uncertainty set. These two stages define the OT-DRO problem.

As described above, the OT-DRO problem is a nested problem with many layers of optimization in minimization and maximization forms. The authors formulate this as a min-min-max ($w$, $\theta$, and $Q$) first, and then rearrange it to min-max-min ($\theta$, $Q$, $w$) to form a bilevel optimization problem. The reason to reduce the lower-level problem to a DRO problem, and run gradient descent for the upper-level problem. There are a couple of reductions in this process (I will mention in the later strengths and weaknesses section) to make it work. Ultimately, the authors further reformulate the lower-level DRO as a conic program and apply differentiable optimization (differentiable conic program) to compute implicit gradient through applying implicit function theorem to the KKT conditions. There is an additional complication where the DRO constraint is hard to handle (Equation 11b), where the authors propose to just sample from the training samples to form an empirical robust constraint.

Combining all of these together, the authors apply this OT-DRO algorithm to the portfolio optimization problem with CVaR constraint,I can't tell if the authors use real data for the portfolio problem, which I will leave it to the authors to elaborate during the rebuttal (I probably just missed it so please correct me). The authors also apply the algorithm to another synthetic linear regression problem, where the samples are generated by a linear function with unknown weights and a white noise. Both experimental results look good to me. Nothing specific except lack of baselines for comparison. This problem setup (OT + DRO) is a little bit special so I assume it is also hard to find baselines to compare. I know a couple of papers and I will mention this in the Q&A section below.

**Compliance With Llm Reviewing Policy:**

Affirmed.

**Final Justification:**

My concerns were mostly addressed by the authors' second response. I would recommend the authors to include the discussion of the comparison with CVaR in the draft as well. With that, I would like to increase my assessment to 4.

**Key Questions For Authors:**

1. Can you comment on the difference between the following paper? The paper uses conformal prediction to quantify the uncertainty, while preserving the validity of the uncertainty set. In your case, you use samples to convert the uncertainty constraints into a quantifiable inequality constraint. I wonder if there are other substantial differences in terms of methodology and the empirical performance.

"Conformal Risk Training: End-to-End Optimization of Conformal Risk Control"
Christopher Yeh, Nicolas Christianson, Adam Wierman, Yisong Yue
NeurIPS 2025

2. Regarding the bilevel formulation in 4.2, is the "sup_Q" part supposed to be in the lower-level problem? I think you can move both the $sup_Q$ and $inf_w$ to the lower-level under minimax theorem. I think this is what you did in your algorithm as you solve the lower-level problem as a DRO instance.

3. Why the lower-level problem is equivalent to a finite-dimensional convex reformulation in Equation 10? I didn't check Appendix A but this intuitively looks too strong / too good to me. Could you please explain briefly here?

4. Similarly, why the upper-level objective is linear in Equation 11a? Isn't the upper-level objective defined by a general loss function $l$ parameterized by $\theta$?

**Limitations:**

Please refer to the above questions and the weaknesses.

**Strengths And Weaknesses:**

Strengths:
* The problem is a challenging problem with OT constraint and DRO involved. It is fundamentally difficult but also important in some applications.
* This is my first time seeing OT constraint used in the decision-focused manner. The sampling trick and reformulation trick to handle OT constraint are useful for future researchers.
* I find some of the reformulation tricks interesting, including from Equation 7 to Equation 8 (but they are also weaknesses as well).


Weaknesses:
* The reduction from Equation 7 to Equation 8 is lossy. Equation 8 is a special case of Equation 7. I understand that the trick is to simplify the bilevel expression though. This is a minor concern to me as I am fine with relaxing it a little bit.
* Reducing the OT constraint to samples. As the authors mention in their limitation section, the probability bound may not hold anymore. It may need a few more probabilistic bound to strengthen the sampled-inequality bound.
* For the remaining detailed points, please refer to the questions below.

---

> ### Author Rebuttal · Authors · 2026-03-31
>
> We sincerely thank the reviewer for the thoughtful evaluation and useful feedback.
> We addressed all points and will incorporate them in the updated manuscript.
>
> **W1.**
> We agree that (8) is a sufficient condition implying (7).
> This relaxation is standard in DRO (see e.g. Mohajerin Esfahani et al. (2018)) and is used when knowledge of $\mathbb{P}$ is only available through an ambiguity set.
> The novelty of our method lies in the parametric design of the ambiguity set, which is typically fixed in standard DRO.
> We will clarify this in the updated manuscript.
>
> **W2.**
> We agree that the lack of strict non-asymptotic bounds is a limitation of our bootstrap approach, only serving as a practical heuristic with good empirical performance.
> To the best of our knowledge, deriving finite-sample guarantees for a trainable, data-driven OT metric $\kappa(\cdot,\cdot;\theta)$ remains an open problem requiring dedicated analysis of the interplay between $J$ and $n_b$.
> We outline a possible path forward.
> First, following Fournier and Guillin (2015), one can show that $d(\mathbb{P}, \hat{\mathbb{P}}; \theta)$ converges in probability to $0$ as $J \rightarrow \infty$, so the bootstrapped quantile provides an asymptotically valid $(1-\beta)$-confidence upper bound.
> Second, a finite-sample guarantee based on $n_b$ might be derived via the DKW inequality:
> $$\text{Pr}(\sup_t |\hat{F}^{n_b}(t) - F(t)| > \delta) \leq 2 e^{-2 n_b \delta^2},$$
> where $\hat{F}^{n_b}(t)$ is the empirical CDF of the bootstrapped distances $d(\hat{\mathbb{P}}(\hat{\mathcal{D}^k_J}), \hat{\mathbb{P}}; \theta)$, and $F(t)$ is the unknown true distribution of $d(\mathbb{P}, \hat{\mathbb{P}}; \theta)$.
> Combining both bounds could yield guarantees on the joint effect of $J$ and $n_b$ on coverage (Eq. (11b)).
> This is intended as a future direction, and not a contribution of this paper.
>
> **Q1.**
> While both our method and Conformal Risk Training (CRT) propose end-to-end approaches to control risk, they differ in assumptions, uncertainty representation, and algorithmic framework.
> CRT relies on conformal prediction, providing distribution-free finite-sample guarantees under data exchangeability.
> Our method instead provides guarantees on *any* distribution within an ambiguity set, making it more suitable when the true distribution may differ from the empirical one (e.g., due to distribution shifts).
> Incorporating such robustness into CRT would require additional techniques.
> We therefore believe the two methods target fundamentally different settings.
> A direct empirical comparison would require either i.i.d. data with no distribution shift (where our robustness is not leveraged) or a shift setting (where CRT's guarantees do not hold).
> Algorithmically, CRT trains model parameters $\theta$ and a risk-controlling parameter $\lambda(\theta)$ obtained via convex optimization.
> Our method does not operate on the model explicitly but focuses on optimizing the geometry of the ambiguity set.
> We will add these observations to the literature review in the updated manuscript.
>
> **Q2.**
> The reviewer's understanding is correct.
> In (9), both $\inf_w$ and $\sup_{\mathbb{Q}}$ are evaluated in the lower level, which for fixed $\theta$ solves the full OT-DRO minimax problem.
> By the minimax theorem and strong duality, this reduces to a single conic program.
> The upper level optimizes only $\theta$, as emphasized in (9), shaping the ambiguity set and influencing the lower-level solution.
>
> **Q3.**
> Eq. (10) is not claimed for arbitrary OT-DRO models but only under the regularity assumptions listed in Appendix A, under which the infinite-dimensional DRO problem reduces to a finite convex program, consistent with standard OT-DRO literature.
> Our numerical experiments satisfy these assumptions and can therefore be reformulated as conic programs.
> We will explicitly emphasize the role of these regularity assumptions in the final version.
>
> **Q4.**
> The lower-level DRO problem is reformulated as a conic program whose optimal value equals the worst-case expected loss by strong duality.
> Although the objective $c(\theta)^\top x$ is linear in $x$, it does not restrict the general loss $l$ to be linear.
> The non-linearity of $l$ is embedded within the conic constraints and the augmented variable $x$, which concatenates decision variables with auxiliary epigraph and dual variables.
> We kindly refer the reviewer to Appendix A for details.
>
> **On the use of synthetic data.** We intentionally employed synthetic data for our experiments (see Appendices E.1 and F.2) to be able to evaluate out-of-sample performance and verify the probabilistic coverage condition $\mathbb{P} \in \mathcal{B}_\epsilon$, which requires knowledge of the ground-truth distribution $\mathbb{P}$, not accessible with real-world data. To ensure clarity, we will explicitly highlight the synthetic nature of our experiments.
>
> **On the choice of Baselines.** Due to the hard character limit, we refer to response to Question 2 of Reviewer gNND.

---

> > ### Author Rebuttal · Reviewer_HTFk · 2026-04-04
> >
> > I thank the authors for the detailed response.
> >
> > Regarding Q1, I still think both the paper and the CRT paper are under the same assumption where you have some empirical samples. CRT paper assumes it is drawn from the same distribution up to some empirical error, while you allow out-of-distribution samples but the error will add up to the OT constraint.
> >
> > So in your paper, with no error added to the OT constraint, the reference distribution $\hat{\mathbb{P}}$ should to be drawn from the true distribution. It is just that the robust optimization is robust to distribution shift. The CRT paper assumes the samples to be from the true distribution, but they are trying to be robust to the CVaR constraint. I still think these two papers are comparable under i.i.d. samples from the true distribution but just with different constraints. I agree that methodology can be different as both look into different constraints. But I believe the OT constraint and the CVaR constraint have some equivalence up to some constant approximation, and it is unclear to me why OT constraint is useful or in what case more suitable than the CVaR constraint.
> >
> > Could the authors clarify it? Thank you!

---

> > > ### Author Response · Authors · 2026-04-06
> > >
> > > We thank the reviewer for this insightful comment. Indeed, assuming that samples are i.i.d. and extracted from the underlying true distribution, both the CRT paper and our paper are comparable in the sense that both account for the statistical error due to the finiteness of the available data, i.e. they account for the mismatch between $\mathbb{P}$ and $\hat{\mathbb{P}} = \frac{1}{J}\sum_{j=1}^J \delta_{\hat{\xi}_j}$.
> > > The OT framework not only handles the statistical error, but is naturally suited to account for different types of distribution shifts, including data-related perturbations such as (i) data noise, which occurs when the data points are measured only with finite precision, and (ii) data misspecification in which a small fraction of all data may be wholly corrupted; as well as shifts between training and testing distributions due to (iii) time-varying domain, label and/or covariate. As the review rightfully points out, upon knowing (partial) information about the "shifting map", the effect of the distribution shifts in practice adds up into the OT constraint.
> > >
> > > Nonetheless, even if for the sake of comparison we restrict ourselves to the setting where only statistical error is considered, we note a key difference between the robustification provided by the OT ambiguity set and the one induced by the CVaR. Following Shapiro et al. *Lectures on Stochastic Programming: Modeling and Theory*, Third Edition, 2009 and referring to "Chapter 6: Risk Averse Optimization" (p. 231), any coherent risk measure can be equivalently written exploiting the Fenchel-Moreau theorem as
> > >
> > > $\mathfrak{R}(\xi) = \sup_{\frac{d\mathbb{Q}}{d\mathbb{P}}\in \mathcal{A}} \mathbb{E}_{\mathbb{Q}}[\xi], \quad \forall \xi \in \mathcal{L}_p(\Omega, \mathcal{F}, \mathbb{P}),$
> > >
> > > where $\mathcal{A}$ is a set of probability density functions, the measure $\mathbb{Q}$ is such that $\mathbb{Q} \ll \mathbb{P}$, and $\mathcal{L}_p(\Omega, \mathcal{F}, \mathbb{P})$, for $p \in [1, \infty)$, is the space of random variables $\xi: \Omega \rightarrow \mathbb{R}$ with finite $p-$th moment with respect to the measure $\mathbb{P}$. This provides a clear intuition: any coherent risk measure can be interpreted as a supremum over an ambiguity set, similarly to the OT formulation.
> > >
> > > More specifically, the CVaR of level $\beta \in (0,1)$ can be described as in the above relationship with
> > >
> > > $\mathcal{A} = \\{ \mathbb{Q} \ll \mathbb{P} \:\Big| \:\frac{d\mathbb{Q}}{d\mathbb{P}} \geq 0, \mathbb{E}_{\mathbb{P}}\left[\frac{d\mathbb{Q}}{d\mathbb{P}}\right] = 1, \frac{d\mathbb{Q}}{d\mathbb{P}} \leq \frac{1}{\beta} \\}.$
> > >
> > > Crucially, when the reference distribution is empirical, the absolute continuity requirement implies that any feasible $\mathbb{Q}$ must be supported on the observed atoms of $\hat{\mathbb{P}}$; hence *CVaR only allows an adverse reweighting of the probability masses of the observed samples*. In contrast, the OT ambiguity set drops the $\mathbb{Q} \ll \mathbb{P}$ restriction. It is defined through a transportation budget and can move probability mass in the ambient space according to $\kappa(\cdot, \cdot; \theta)$, thereby modeling spatial perturbations rather than only reweightings of observed atoms. It allows to *robustify in space by moving probability mass to completely unobserved, out-of-sample points in the ambient space*. Put differently, CVaR answers "which observed tail losses should be emphasized?", while OT answers "which spatial perturbations of the underlying data should be considered plausible, and how costly are they?", allowing for higher modeling expressivity.
> > >
> > > Moreover, for a given risk level $\beta$, the ambiguity set associated to the CVaR is "given", while in the OT formulation the ambiguity set is a geometry-aware object that the modeler explicitly specifies or learns through $\kappa(\cdot, \cdot\, ;\, \theta)$ and $\varepsilon$.
> > > As a result, the decision-maker could also decide to combine both CVaR and OT-DRO (i.e., a distributionally robust CVaR formulation as already illustrated in our portfolio experiment in Section 6.1, Eq. 18) since, according to the above characterization, they induce different types of robustification.
> > >
> > > To conclude, we view the two approaches as addressing different modeling goals with different theoretical trade-offs. While the CRT paper provides strong distribution-free finite-sample guarantees for scalar risk control under exchangeability, our OT framework enables richer geometric modeling of spatial distributional uncertainty. Furthermore, our framework optimizes a geometry-aware ambiguity set parameterization through $\kappa (\cdot, \cdot ; \theta)$, rather than only tuning a scalar risk-control parameter, and we provide an algorithm with convergence guarantees despite the non-differentiability of the lower level by invoking the recent concept of the *nonsmooth* implicit function theorem.

---

### Official Review · Reviewer_gNND · 2026-03-12

**Soundness:** 3
**Presentation:** 4
**Significance:** 3
**Originality:** 3
**Overall Recommendation:** 5
**Confidence:** 4

**Summary:**

The paper addresses a well-known issue in Optimal Transport-based Distributionally Robust Optimization (OT-DRO): over-conservatism. Standard OT-DRO approaches separate the ambiguity set design from the downstream optimization task, often resulting in overly pessimistic decisions. To mitigate this, the authors propose an end-to-end bilevel optimization framework to learn a "decision-focused" ambiguity set.The upper-level problem minimizes the downstream empirical loss to shape the geometry of the ambiguity set (parameterized, for instance, by a Mahalanobis matrix $\theta$), subject to a data-driven bootstrap constraint to ensure statistical coverage. The lower-level solves the standard OT-DRO problem given this set. To solve this bilevel formulation, the authors leverage the nonsmooth conservative implicit function theorem, enabling hypergradient descent by differentiating through the KKT conditions of the lower-level conic programs. The method is evaluated on portfolio optimization and linear regression tasks, demonstrating reduced conservatism compared to using standard, unoptimized ambiguity sets.

**Compliance With Llm Reviewing Policy:**

Affirmed.

**Key Questions For Authors:**

1. Computational Overhead: Could you provide wall-clock times for the experiments presented? How does the computational time of the hypergradient step scale empirically as the dimension $k$ and the number of samples $J$ increase?
2. Baselines: Can you provide an empirical comparison (or at least a detailed qualitative discussion) against other methods designed to reduce conservatism in DRO, such as Blanchet et al. (2019b)?
3. Initialization Sensitivity: Given the non-convexity of the upper-level problem, how sensitive is your algorithm to the initialization of the matrix $L_0$ ? Did you observe significantly different local optima or out-of-sample performances when starting from different random SPD matrices instead of the identity matrix?

**Limitations:**

Yes. The authors adequately and honestly address the main theoretical limitation in Section 7: the lack of formal theoretical guarantees for the bootstrap-based coverage constraint (Eq. 11b). Using a heuristic resampling method is a practical compromise, and acknowledging this openly is appreciated. I do not foresee any direct negative societal impacts stemming from this theoretical work.

**Strengths And Weaknesses:**

Strengths:
Originality & Motivation: The core idea of using the downstream loss to inform and shape the optimal transport ambiguity set is highly motivated and elegant. It directly addresses the "price of robustness" that plagues practical deployments of DRO.

Soundness in Theoretical Execution: Applying the nonsmooth conservative implicit function theorem to differentiate through the solution map of conic programs is rigorous. The authors do not gloss over the non-differentiability of the lower-level optimization but rather address it properly using conservative Jacobians, establishing solid convergence guarantees (Theorem 5.1).

Presentation: The paper is exceptionally well-structured and clearly written. The transition from the high-level bilevel formulation to the specific, tractable conic reformulations (detailed nicely in the appendices) makes it easy to follow for readers familiar with optimization.

Weaknesses:
Lack of Strong Baselines: The empirical evaluation primarily compares the optimized ambiguity set $\left(L^*\right)$ against the unoptimized initial guess $\left(L_0\right)$. While this demonstrates that the bilevel procedure does reduce conservatism, the paper lacks comparisons against existing methods that also attempt to reduce DRO conservatism (e.g., the Mahalanobis distance calibration in Blanchet et al., 2019b, or other decision-focused/metric learning approaches mentioned in the Related Works).

Missing Empirical Computational Analysis: While Appendix G discusses theoretical computational complexity, the main text lacks a practical discussion of wall-clock time. Differentiating through a conic program at every upper-level iteration is notoriously expensive. It is unclear how this method scales in practice for larger dimensions ( $k$ ) or massive sample sizes $(J)$.

Sensitivity to Initialization: Appendix D proves that the parameterized distances are generally non-convex with respect to the parameter matrix $L$. Consequently, the upperlevel problem might be highly sensitive to the initialization $\theta_0$. The paper currently lacks an ablation study examining how different initializations affect the final out-of-sample performance and the discovered geometry.

---

> ### Author Rebuttal · Authors · 2026-03-30
>
> We thank the reviewer for the positive feedback and address the questions below; we will incorporate this discussion in the final version.
>
> **Q1: Computational Overhead.**
> While the current manuscript provides theoretical complexity scaling in Appendix G, the revised version will include empirical wall-clock measurements for the experiments. All timings correspond to a single upper-level iteration and were obtained on an HP ProBook G10 equipped with an Intel i7-1355U processor.
>
> For the portfolio optimization task with discrete reference distribution and type-1 Wasserstein distance:
>
> - Fixing the decision dimension $k=10$ and varying the sample size $J \in$ {10, 20, 50, 100}, the average per-iteration times are $0.0047$ $(\pm 0.003)$ s, $0.0064$ $(\pm 0.003)$ s, $0.015$ $(\pm 0.007)$ s, and $0.022$ $(\pm 0.01)$ s, respectively.
>
> - Fixing the sample size $J=50$ and varying the dimension $k \in$ {2, 5, 10, 20}, the corresponding times are $0.0078$ $(\pm 0.004)$s, $0.011$ $(\pm 0.003)$s, $0.018$ $(\pm 0.007)$s, and $0.030$ $(\pm 0.01)$s.
>
> For the linear regression task with type-2 Wasserstein distance:
>
> - Fixing $k=10$ and varying $J \in$ {10, 20, 50, 100} yields per-iteration times of $0.0058$ $(\pm 0.0034)$ s, $0.0098$ $(\pm 0.0052)$ s, $0.046$ $(\pm 0.016)$ s, and $0.22$ $(\pm 0.064)$ s, respectively.
>
> - Fixing $J=50$ and varying $k \in$ {2, 5, 10, 20} yields $0.010$ $(\pm 0.005)$ s, $0.018$ $(\pm 0.009)$ s, $0.051$ $(\pm 0.019)$ s, and $0.21$ $(\pm 0.071)$ s, respectively.
>
> While differentiating through conic programs introduces additional overhead compared to standard DRO, this cost is incurred only during an offline training phase to learn the optimal set geometry $\theta^\star$. Once learned, computing the downstream robust decision for new data requires no extra computational effort at deployment.
>
> **Q2: Baselines.**
> Our current experiments were designed to isolate the contribution of the bilevel learning procedure by directly comparing the optimized geometry against the standard unoptimized initialization. To the best of our knowledge, no prior work tackles the ambiguity set shaping problem in the generality presented here.
>
> While we could not perform a full comparative analysis due to time constraints, we provide a qualitative comparison:
>
> - Comparison with Blanchet et al. (2019b). Our key distinction is the end-to-end integration of the downstream task. Blanchet et al. calibrate a Mahalanobis distance by minimizing the asymptotic variance of loss gradients. However, this operates as a decoupled, a-priori calibration step, and relies on asymptotic expansions primarily tailored to smooth, unconstrained expected loss minimization (like linear or logistic regression). As a result, this procedure is necessarily not less conservative than our formulation, where the metric learning and the DRO problem are solved jointly.  Moreover, their method is limited to linear regression (see reformulation in Theorem 1 of Blanchet et al. (2019b)), whereas ours applies broadly to any DRO problem with a finite-dimensional conic reformulation.
>
> - Comparison with Schuurmans et al. (2023). Contrary to our approach, Schuurmans et al. (2023) only handles the discrete distribution case and allows for less flexibility in the transportation cost parametrization.
>
> **Q3: Initialization Sensitivity.**
> As proved in Appendix D, the parameterized transportation costs are generally non-convex with respect to $L$, making sensitivity to the initialization $\theta_0$ a legitimate concern. Our choice of setting $L_0 = I$ in the paper was deliberate: initializing with the identity matrix corresponds exactly to the standard, unoptimized Wasserstein DRO formulation, ensuring our optimization naturally explores geometries that strictly improve upon the conservatism of the most widely accepted baseline.
>
> To empirically evaluate the initialization sensitivity, we ran additional experiments.
>
> For the Gaussian/CVaR portfolio optimization problem ($J=20$, $k=2$, $n_b=10$, $\beta=0.05$), instead of $L_0 = I$, we ran our algorithm to convergence from $20$ different randomly-generated lower-triangular matrices with strictly positive diagonals (representing different valid initializations in $\mathbb{L}_{++}^k$, normalized to unit Frobenius norm).  We observed that the algorithm converges to the same solution (in terms of out-of-sample performance and matrix $L$) for all $20$ random initializations.
>
> We run a similar experiment for the linear regression task ($k=2, J=20$, $n_b=10$, type-2). In this case, the algorithm consistently converges to one of three distinct clusters across the $20$ random initializations, indicating the presence of multiple local minima. Given the non-convex nature of the bilevel problem at hand, this behavior is expected.
>
> Due to the ICML rebuttal policy, we are unable to show the results in a plot format; however, in the final version of the paper we will provide convergence plots for both experimental settings.

---

### Decision · Program_Chairs · 2026-04-30

**Decision:**

Accept (spotlight)

**Comment:**

This paper introduces a new framework for distributionally robust optimization with optimal transport constraints. The idea is to parameterize the uncertainty set and then optimize over this parameter to find a set which covers the true distribution with high probability while inflating loss on the downstream optimization problem by the smallest possible amount. Reviewers felt that this is an original idea that addresses an important problem in DRO (overly conservative solutions), and I agree.

The response cleared up a few questions with the manuscript. I encourage the authors to add a short discussion of CVaR constraints; I believe that the authors are correct on this point but a bit of exposition about the relationship may help readers with less background in DRO.